# Diel cycle of sea spray aerosol concentration

J. Michel Flores [1✉], Guillaume Bourdin [2,3], Alexander B. Kostinski [4], Orit Altaratz [1], Guy Dagan[5], Fabien Lombard [3], Nils Haëntjens [2], Emmanuel Boss [2], Matthew B. Sullivan[6], Gabriel Gorsky[3,7], Naama Lang-Yona [8,11], Miri Trainic[1], Sarah Romac [9], Christian R. Voolstra [10], Yinon Rudich [1], Assaf Vardi [8✉] & Ilan Koren [1✉]

Sea spray aerosol (SSA) formation have a major role in the climate system, but measurements at a global-scale of this micro-scale process are highly challenging. We measured high-resolution temporal patterns of SSA number concentration over the Atlantic Ocean, Caribbean Sea, and the Pacific Ocean covering over 42,000 km. We discovered a ubiquitous 24-hour rhythm to the SSA number concentration, with concentrations increasing after sunrise, remaining higher during the day, and returning to predawn values after sunset. The presence of dominating continental aerosol transport can mask the SSA cycle. We did not find significant links between the diel cycle of SSA number concentration and diel variations of surface winds, atmospheric physical properties, radiation, pollution, nor oceanic physical properties. However, the daily mean sea surface temperature positively correlated with the magnitude of the day-to-nighttime increase in SSA concentration. Parallel diel patterns in particle sizes were also detected in near-surface waters attributed to variations in the size of particles smaller than ~1 μm. These variations may point to microbial day-to-night modulation of bubble-bursting dynamics as a possible cause of the SSA cycle.

[1] Weizmann Institute of Science, Department of Earth and Planetary Sciences, Rehovot, Israel. [2] School of Marine Sciences, University of Maine, Orono, ME, USA. [3] Sorbonne Université, CNRS, Laboratoire d'Océanographie de Villefranche, Villefranche-sur-Mer, France. [4] Department of Physics, Michigan Technological University, Houghton, MI, USA. [5] Atmospheric, Oceanic and Planetary Physics, Department of Physics, University of Oxford, Oxford, UK. [6] Departments of Microbiology and Civil, Environmental and Geodetic Engineering, Ohio State University, Columbus, OH, USA. [7] Research Federation for the Study of Global Ocean Systems Ecology and Evolution, FR2022/Tara Oceans-GOSEE, Paris, France. [8] Weizmann Institute of Science, Department of Plant and Environmental Science, Rehovot, Israel. [9] Sorbonne Université, CNRS, Station Biologique de Roscoff, AD2M, UMR 7144, ECOMAP, Roscoff, France. [10] Department of Biology, University of Konstanz, Konstanz, Germany. [11]Present address: Civil and Environmental Engineering, Technion - Israel Institute of Technology, Haifa, Israel. ✉email: flores@weizmann.ac.il; assaf.vardi@weizmann.ac.il; ilan.koren@weizmann.ac.il

Marine aerosols (MA), defined as the aerosols present in the atmospheric marine boundary layer (AMBL), consist of a mixture of natural and anthropogenic components. MA are predominantly composed of four main sources: sea spray aerosols (SSA), aerosols transported from land (i.e., long-range transport), gas-to-particle conversion (homogeneous and heterogeneous nucleation, and condensation), and particles from ship exhausts. The total concentration of MA depends on the distance from its source in combination with transport and deposition processes. This leads to large spatial variability of MA concentrations, with higher concentrations near continents where the terrestrial contribution is larger, than in open ocean regions[1,2]. Moreover, the distance from the shore also acts as a size sorter as smaller particles are transported, in general, longer distances and have longer lifetimes than coarse particles[3]. SSA, the main component of MA globally, are generated at the ocean surface by wind-driven processes and have a major effect on climate as they impact atmospheric chemistry, Earth's radiative balance, cloud formation, and rain properties[4–8]. As ocean waves break, SSA forms when bubbles burst and create film and jet droplets[9]. At strong winds (above ~9 m s$^{-1}$) spume drops[10] can be torn off the wave crest. SSA production depends on the physical[11] and chemical[12,13] properties of the ocean surface as these, together with wind stress, determine the bubble's bursting dynamics and, therefore, the quantity, size, and composition of the emitted drops. The emitted drops transport, in turn, depends on the AMBL conditions such as the wind, profiles of relative humidity (RH), temperature, instability, and rain[4], and have an average (tropospheric) residence time of about half a day[6]. Understanding the processes that control SSA flux into the atmosphere is essential to better understand the climate system, weather, and accurate ocean-atmosphere models. However, global-scale coverage is generally restricted to remote sensing daytime measurements, and many of these processes, including whether a diel cycle in SSA production exists, have only been explored using artificial bubbling[13,14], where diel variability in mass and number fluxes was observed in biologically productive waters.

In this study, we explored and quantified diurnal patterns of the number concentration of marine aerosols ($N_{MA}$) in high spatial and temporal (every minute) resolution along 42,000 km over the Atlantic Ocean, the Caribbean Sea, and Pacific Ocean, measured aboard the schooner *Tara* during the *Tara* Pacific Expedition[1,15,16]. The expedition began on 28 May 2016 from Lorient, France, and finished the first year of the campaign on 17 June 2017, in Whangarei, New Zealand. Using an optical particle counter (OPC), aerosol size distributions (for optical diameters, $D_{op}$, between 0.25 and 32 μm at RH < 40%) were continuously measured at 30 m above sea level, along with the spectral particulate absorption and attenuation coefficients ($c_p$) at ~1.5 m water depth[17], as well as AMBL (temperature, RH, wind speed and direction, and photosynthetic active radiation) and oceanic (salinity, sea surface temperature, and chlorophyll a) variables. In addition, aerosols were collected on filters (see "Methods" for instrumentation details). *Tara*'s route combined sailing periods with days to week-long stops near islands. Here we report the discovery of a distinct 24-h pattern in the number concentration of marine aerosols, show that SSA number concentration ($N_{SSA}$) variations are responsible for this pattern, and suggest a possible mechanism.

## Results and discussion
**Main components of the marine aerosol number concentration**. We can define the total concentration of MA as the sum of four components:

$$N_{MA} = N_{SSA} + N_{LRT} + N_{SOA} + N_{SE} \qquad (1)$$

where $N_{LRT}$ is the number concentration of long-range continental transport, $N_{SOA}$ is the number concentration from aerosols formed from gas-to-particle conversion mechanisms, and $N_{SE}$ refers to the number concentration from ship emissions.

To distinguish between different aerosol sources, it is common to discuss the different aerosol types and properties according to their sizes (for dried particles or particle sizes measured at a certain RH), where different regimes of the aerosol size distribution belong to different aerosol types and processes[5]. For instance, the freshly produced marine secondary organic aerosol (SOA) is small in diameter (<0.1 μm); whereas, the typical size range of SSA is larger (>0.1 μm at an RH ~80%)[4]. The marine anthropogenic sources, such as engine pollution, yield sporadic large number concentrations of small submicron particles[1]. And, continental aerosols that are transported over the ocean have a variety of sources and therefore sizes; for example, the typical sizes of biomass burning and pollution aerosols are fine[18], whereas desert dust aerosols are usually coarser (>~0.6 μm[19]). Given we report $N_{MA}$ measurements for $D_{op} > 0.25$ μm, we can discard significant contributions to the total $N_{MA}$ from $N_{SE}$ and $N_{SOA}$, and we only have an interplay between the contribution of $N_{LRT}$ and $N_{SSA}$. To this end, given the surface winds range we encountered ($U_{30} < 16$ m s$^{-1}$), as a first approximation, the attribution between the local contribution of $N_{SSA}$ and $N_{LRT}$ can be scaled to the concentration. SSA is produced by the wind stress over the ocean surface, and its concentration has been shown to be (nonlinearly) propositional to the wind speed[4,20,21], with typical concentrations in the range of 10 s per cubic centimeter[21,22]. Whereas the contribution of $N_{LRT}$ is proportional to the yields of the continental sources (dust, pollution, smoke), the distance the aerosol traveled, and with aerosol lifetime in the atmosphere, in general, inversely proportional to the aerosol size[3,23,24], the quantity of smaller diameters will be greater. For example, being near the Saharan desert and tropical biomass burning areas (tropical Africa and the Amazon), the Atlantic ocean's atmosphere is much more polluted compare to the Pacific ocean[1,25]. Thus, low $N_{MA}$ values (i.e., ~<50 cm$^{-3}$ at $D_{op} > 0.25$ μm) indicate that $N_{SSA}$ is the primary contributor with low contribution from $N_{LRT}$, and an indicator of the contribution from $N_{LRT}$ will be seen as an increase in the $N_{MA}$.

**Diel pattern in marine aerosol number concentration**. We explored temporal patterns of the $N_{MA}$ with $D_{op} \geq 0.25$ μm and found there is a consistently higher number of aerosols during daytime compared to nighttime (Fig. 1A).

We calculated the day to nighttime ratios (DNR) by averaging from 07:00 to 17:00 MST (mean solar time; all data were converted from UTC to MST, see "Methods") for the day and from 19:00 to 05:00 MST for the night. The measurements between 05:00 to 07:00 and 17:00 to 19:00 were excluded to avoid strong fluctuations of the ratio. The DNR were calculated across *Tara*'s route for each size bin of the OPC and found it to be >1 for all sizes on the vast majority of the route (Fig. 1). DNR are shown to depend strongly on the aerosol diameter and the region (Figs. 1 and 2). For smaller aerosol diameters ($0.25 < D_{op} < 0.5$ μm) the DNR is closer to one (for the Atlantic and along the Keelung to Fiji leg in the Pacific Ocean, we could calculate the DNR for $D < 0.25$ μm and found it to be ~1 for all diameters down to ~0.03 μm; Supplementary Fig. 1), and it becomes larger as the diameters increase, reaching DNR > 10 for $D_{op} > 1.0$ μm. The DNR are larger for all diameters for the Pacific Ocean where total

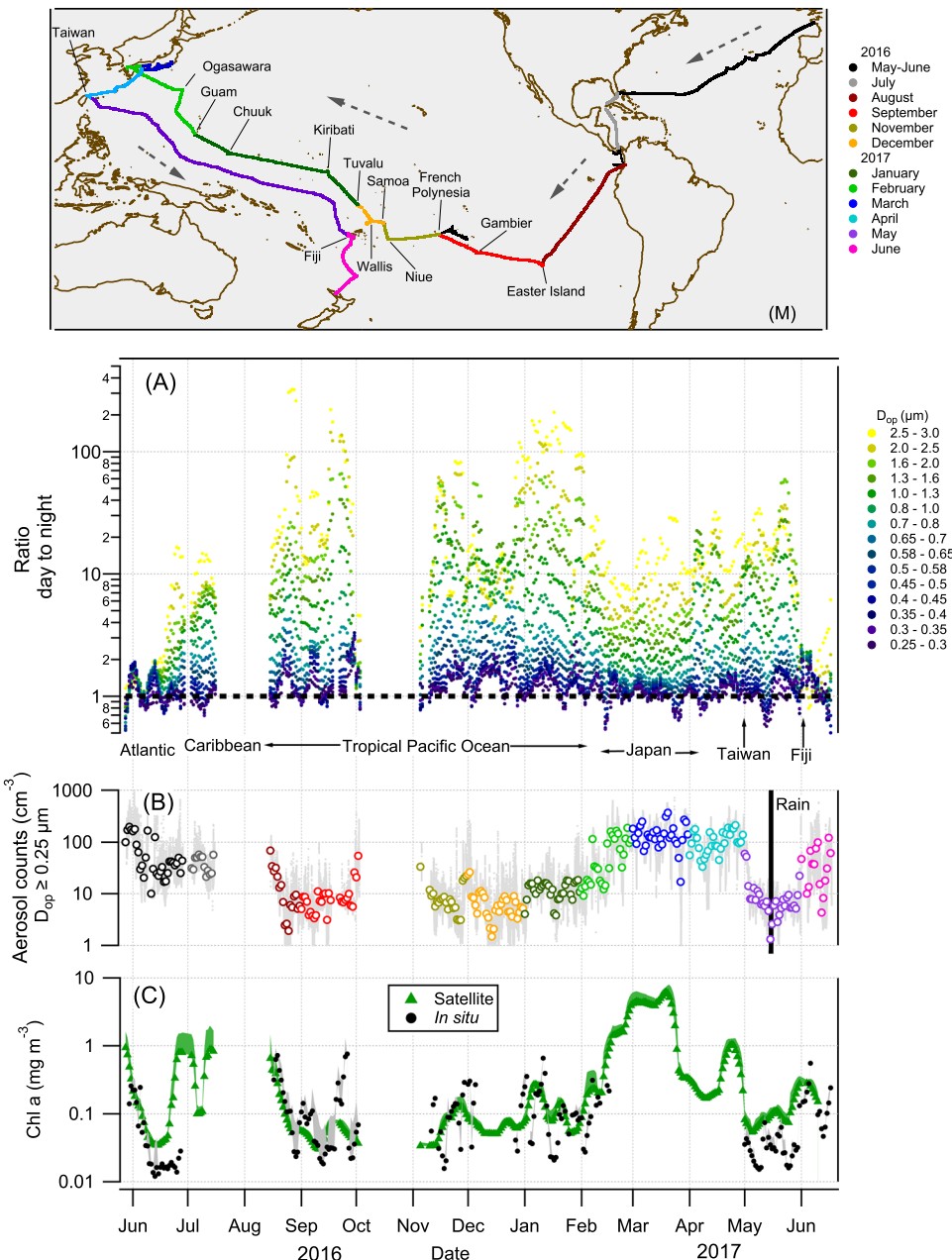

**Fig. 1 Day-to-night ratios for individual bins of the optical particle counter, total marine aerosol count for optical diameter ($D_{op}$) ≥ 0.25 μm, and chlorophyll-*a* concentration along R/V *Tara*'s route.** (M) *Tara*'s route, color-coded by month and dotted arrows showing the sailing direction. **A** The ratio (5-day running average) of day-to-night concentration of marine aerosol for each size bin of the optical particle counter. **B** The total aerosol count (daily mean) for $D_{op}$ ≥ 0.25 μm, the gray dots are the raw total counts per minute. The colors as defined in the map. **C** A 5-day running average of chlorophyll-*a* concentration measured with the AC-S aboard *Tara* (black circles) and calculated using satellite data (green triangles; see "Methods"), with the shaded area outlining one standard deviation. The day-to-night ratios are greater than one on the vast majority of the route and largest in clean areas with low chl-*a* concentration, i.e., in oligotrophic ("blue") waters.

aerosol concentrations are lowest ($N_{MA} \approx 10$ cm$^{-3}$), and are smaller in areas where the aerosol concentration is higher (Figs. 1A, B and 2); for example, the Atlantic Ocean, around Japan, near some of the Pacific islands, and on the leg from Fiji to New Zealand. Given the MA properties listed above, the increase in the concentration, to around $N_{MA} \approx 100$ cm$^{-3}$ over these areas, indicates larger contribution of $N_{LRT}$. The Atlantic Ocean has generally higher background aerosol concentration due to high mineral dust load[1,5,25], in the Japan and Taiwan legs *Tara* was near the coast, and in the Fiji to New Zealand the back trajectories

were coming directly from New Zealand (see next section and Fig. 3M). In all these areas the DNR was closer to one (Fig. 2), suggesting that a stronger influence of $N_{LRT}$ from the continents masks the local cycle.

In addition, aerosol lifetime and dry deposition velocities can also play a role in the size dependence of the DNR. The lifetime of fine mode aerosols can easily exceed a few days[6], therefore, any cycle that its characteristic periodicity is a day or longer will be averaged out for such aerosols. The larger the aerosols are, the shorter their lifetime and greater their dry deposition velocity[24],

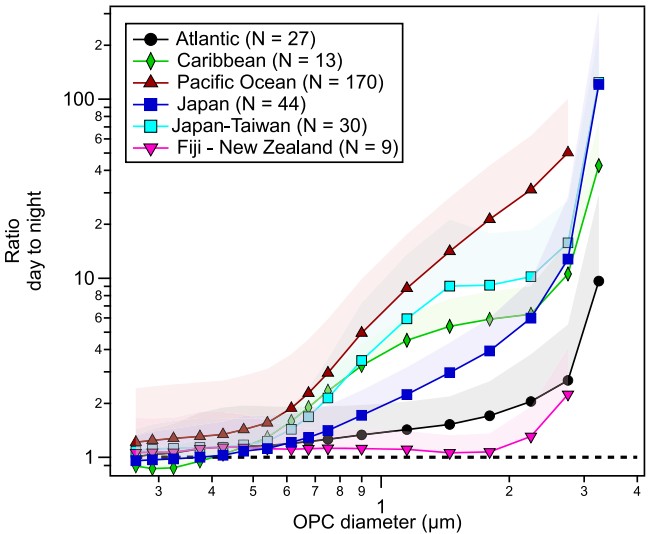

**Fig. 2 Mean day-to-night aerosol count ratio for each bin of the optical particle counter (OPC) for different regions across *Tara*'s route.** The shaded areas represent $1\sigma$, and only the top parts are shown for clarity. *N* refers to the number of days analyzed.

consequently, their concentration in the atmosphere will follow lifetime cycles with shorter periodicities. When we examined the coarse subset of the measured aerosols, a clear daily cycle emerges. The pattern becomes readily visible for particles with $D_{op} > \sim 0.58\ \mu m$ (Fig. 3A and Supplementary Fig. 2), where the diel changes follow a definite pattern: from midnight to dawn the number concentration is stable, it begins rising at $\sim$06:00 MST, within 30 min after sunrise and begins returning to pre-dawn values at $\sim$17:00 MST, with 18:00 MST concentration values remaining stable overnight. The $N_{MA}$ diel cycles were identified while sailing in oligotrophic oceanic regions (Fig. 3A), as well as in more biologically active water and island stopovers near coral reefs (Supplementary Fig. 3). This suggests that it is a widespread phenomenon across the oceans, and that islands (and *Tara*'s; see Supplementary Note 1) daily processes do not drive the cycles.

**SSA cause the marine aerosol diel cycles**. To further explore the interplay between the $N_{LRT}$ and $N_{SSA}$, we used back trajectory analysis and scanning electron microscopy with energy-disperse X-ray (SEM-EDX) analysis, and we determined the main contributor to the measured $N_{MA}$ cycles to be $N_{SSA}$ (Fig. 3).

The calculated back trajectories (using HYSPLIT[26,27]) show that the vast majority of the air masses spent at least 48 h over the ocean (Fig. 3M). For the tropical Pacific Ocean, besides the Japan–Taiwan region, a couple of days near South America, and near New Zealand, the air masses were coming from the middle of the ocean, thousands of kilometers from the nearest continent. The long residence of the air masses over the ocean will, on the one hand, minimize the influence of continental sources; for example, we see total aerosol concentrations, for $D_{op} \geq 0.25\ \mu m$, around $10\ cm^{-3}$ (Fig. 1B). On the other hand, it will buffer diurnal continental emission processes, therefore, the hour-by-hour concentration changes will be dominated by local processes and thus $N_{SSA}$.

Next, for the period where clear $N_{MA}$ diel patterns are observed in the western Pacific (Fig. 3A; see the orange-blue shaded region in the map), we obtained an approximate chemical signature of aerosols collected on filters with average geometrical diameters $D_{geo} > 0.3\ \mu m$. We used SEM-EDX analysis and a similar particle classification scheme as described in Laskin et al.[28] (Fig. 3B; a total of 15.5 days were analyzed between 3 and 17 May, 2017, but for

clarity only 11 days are shown. See Supplementary Table 1 for the other days and "Methods" for the classification scheme). We found that over 94% of the particles analyzed contained Na, of which *sea-salt* particles comprised between 45 and 98% of the total particles by number. Up to 1000 km (May 9; orange-shaded area in the map of Fig. 3) away from Keelung, Taiwan, we identified a noticeable depletion of chloride (Supplementary Fig. 4), together with a lower *sea-salt* fraction (45–75%) and an increase in the presence of other metals (e.g., Al, Si, K, Ca, S) and sulfate. The chlorine depletion (seen as a second mode in the chlorine mass distribution with Cl mass % <0.1; Supplementary Fig. 4b) together with the increase in sulfates, suggests that anthropogenic pollutants (e.g., $H_2SO_4$, $SO_2$) were present in the AMBL, as SSA are known to react with them[28]. After this period, the *sea-salt* fraction comprised 81–99% of the total particles by number. *Sulfates* with no sodium and *Other* species were <6% for the whole period. The $N_{MA}$ diel cycles were also revealed in the filter counts; we counted a total of 5701 sea salt particles (out of 6901) in 14 daytime filters and 4447 sea salt particles (out of 5894) in 15 nighttime filters (see Supplementary Table 1). We have not, however, observed significant day-to-night aerosol-class differences.

This 2-week SEM-EDX analysis where clear diel patterns were seen confirms the primary type of aerosol exhibiting diel cycles to be sea salt. The diameters measured in the filters using the SEM are the dry geometrical diameters, which differ from the OPC diameters that depend on the particle's shape and refractive index (RI) compared to that of the particles used for the OPC calibration (typically polystyrene latex spheres with an RI = 1.59[29]). The difference between the measured particle RI and the RI of the particles used for calibration causes the discrepancy between $D_{geo}$ and $D_{op}$. For instance, the geometrical diameters of dry sea salt are generally 4–30% larger than the dry optical diameter[29] (e.g., $D_{geo} = 0.3\ \mu m$ is $\sim D_{op} = 0.28\ \mu m$; see Supplementary Fig. 2). The OPC measured the aerosol at an RH < 40%, which is below the efflorescent point of NaCl[30], though some SSA may still contain water bound in the form of hydrates[31–33]. Therefore, most aerosols measured by the OPC during the diel cycles will be sized smaller than their geometrical size. Consequently, the vast majority of the particles of all diameters measured by the OPC showing the diel pattern can be considered dry or slightly humidified sea salt. For particles with $D_{geo}$ larger than about 0.5 μm, previous measurements in the AMBL have also shown that the dominant component of SSA in the marine aerosol mass size distribution is inorganic sea salt[21,22,34,35], consistent with our findings.

Thus, we can conclude that the diel cycle of $N_{MA}$ is primarily controlled by $N_{SSA}$ and that in areas where long-range transport is significant, the $N_{LRT}$ can mask the day-to-night differences.

**Atmospheric and oceanic environmental factors that may control SSA production and transport**. Given that we found that the diel cycle of $N_{MA}$ is primarily controlled by the $N_{SSA}$, we explore the factors that can affect SSA production and transport. Since the $N_{SSA}$ diel cycle is clearly visible for $D_{op} \geq 0.58\ \mu m$ (Fig. 3A), we use the $N_{SSA}$ for $D_{op} \geq 0.58\ \mu m$ ($N_{SSA \geq 0.58\ \mu m}$) and the DNR of $N_{SSA \geq 0.58\ \mu m}$ as proxies for the diel cycle to help understand the impact of atmospheric and oceanic variables on the $N_{SSA}$ diel cycle, as any effect caused by atmospheric or oceanic variables will be more easily visible.

The size distribution and number concentration of the bubbles and SSA created by a breaking wave are controlled mainly by the wind speed[4]. First, we found that $N_{SSA \geq 0.58\ \mu m}$ increases with wind speed for both the Pacific and Atlantic Oceans (Fig. 4A, B), consistent with previous studies[4,5,21]. However, the Pacific Ocean daytime $N_{SSA \geq 0.58\ \mu m}$ values consistently exceeded the nighttime

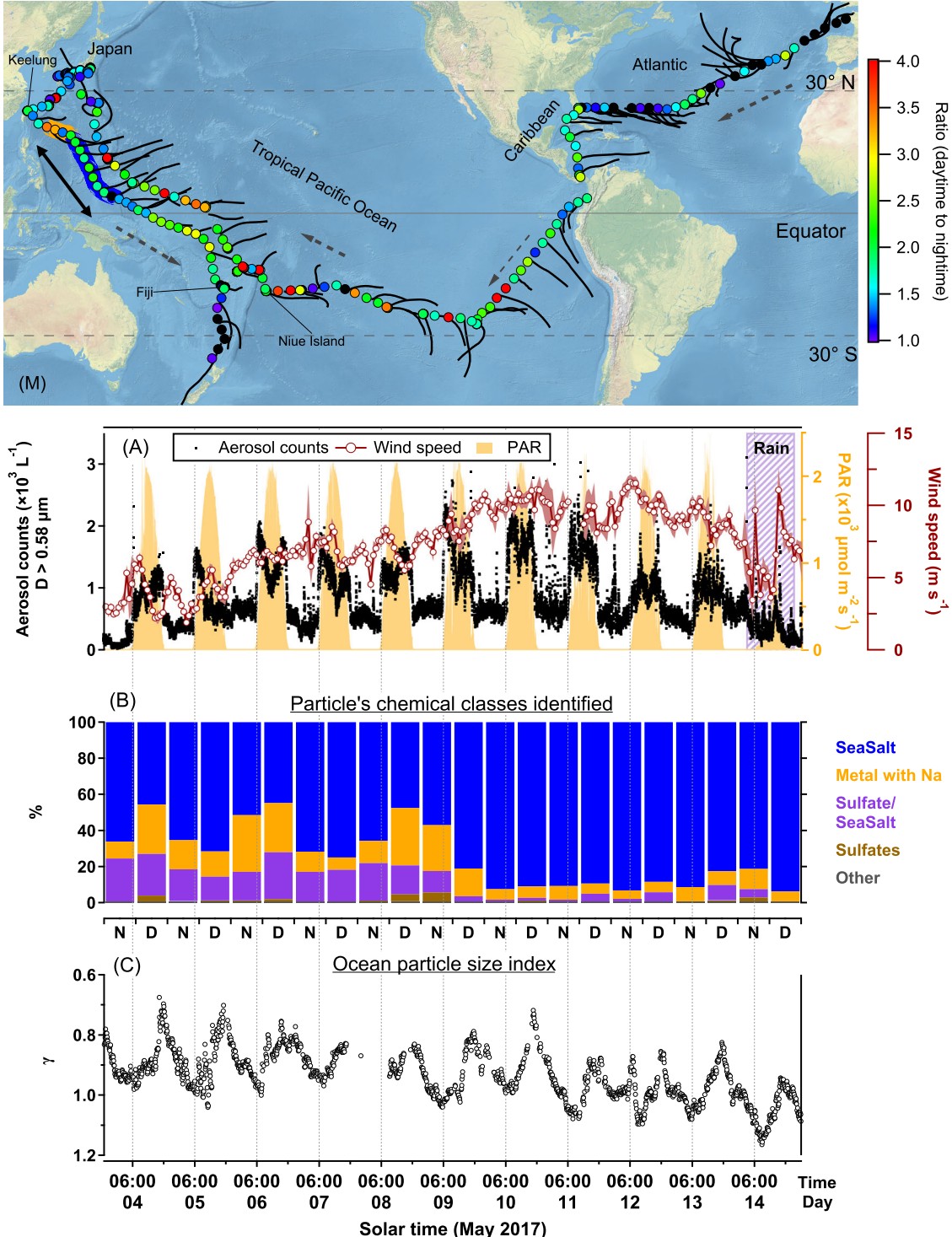

**Fig. 3 Twenty-four cycle of aerosol concentration and marine particle size index.** (M) Map of R/V *Tara*'s route, with dotted arrows along the sailing direction and solid black lines along the 48-h back-trajectories. Filled circles on the route are colored by the value of the day-to-night concentration ratio. The data in panels (**A**) through (**C**) are from the orange and blue-shaded transect in the western Pacific between Keelung and Fiji (next to the double-ended arrow). The orange-shaded region represents anthropogenic polluted conditions, and the blue-shaded refers to clean ones. **A** Main observation. Aerosol concentration per liter (optical diameter ($D_{op}$) > 0.58 µm, collected 30 m above sea surface), superimposed on the 24-h beat of incoming solar flux as represented by the photo-synthetically active radiation (PAR). Time series are punctuated by abrupt spikes at dawn and drops at dusk. The diel rhythm (away from land) is evident, ubiquitous, and persists on cloudy days. Pollution origin of this cycle is ruled out by the 48-h back-trajectories. **B** Aerosol composition determined by SEM-EDX for geometrical diameters ($D_{geo}$) > 0.3 µm. N and D denote night and day, respectively. This is compelling evidence for the marine origin of the aerosols. The collection filters were replaced at about 08:00–09:30 and 20:00–21:30 (see Supplementary Table 1 in the SI for timing details). **C** Twenty-hour signal of marine particle size index $\gamma$ (vertical axis inverted), where the mean particle size increases during the day and decreases during the night. Data collected at 0.5–3 m below the sea surface.

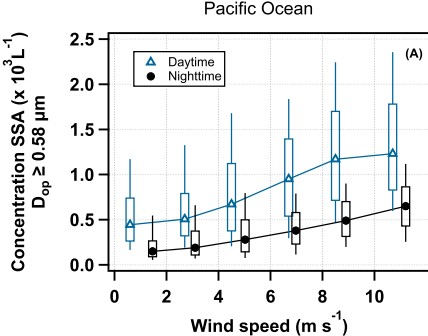
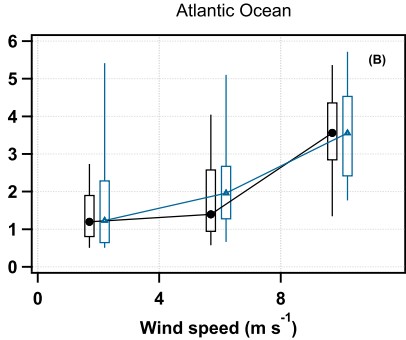

**Fig. 4 Dependence of the sea spray aerosol number concentration with optical diameters ($D_{op}$) ≥ 0.58 μm ($N_{SSA ≥ 0.58 μm}$) on wind speed.** Box plots of the $N_{SSA ≥ 0.58 μm}$ vs. wind speed, binned at 2 ms⁻¹ for the Pacific Ocean (**A**) and by 4 ms⁻¹ for the Atlantic Ocean (**B**) where data collected further than 100 km away from land was used. The day and night data are offset for clarity. The y-axis scale is different for the two panels. While the expected increase of aerosol concentration with wind speed is indeed observed, no relation between the 24-h cycle and wind speed is found. The box plots show the median, and the 10th, 25th, 75th, and 90th percentiles. For the daytime (nighttime) Pacific Ocean, N = 3612 (4057), 11,638 (14,833), 20,134 (19,506), 19,988 (23,136), 15,701 (16,097), and 8129 (9034) samples where used for the 0–2, 2–4, 4–6, 6–8, 8–10, and >10 m s⁻¹ bins, respectively. For the daytime (nighttime) Atlantic Ocean, N = 1089 (949), 4768 (5056), and 1082 (1707) samples were used for the 0–4, 4–8, and >8 m s⁻¹ bins, respectively.

ones, compared at similar wind speeds (Fig. 4A). This shows that the wind is a key player in the process of SSA emission, but is not driving the observed diel cycles.

To evaluate the interplay between the diel cycle, the wind speed, and the influence of long-range continental transport, we explored the relationship between the DNR of $N_{SSA ≥ 0.58 μm}$, and the total aerosol concentration (taken as the concentration for $D_{op} ≥ 0.25 μm$ during daytime) at different wind speed regimes (Fig. 5). As discussed above, higher concentrations imply greater contribution of nonlocal aerosols that mask the cycle. Figure 5 shows that for the cases with concentration >100 # cm⁻³, the DNR approaches unity regardless of the wind speed and location. In the cleaner conditions of the Pacific Ocean ($N_{MA}$ < 50 cm⁻³), we found high variability of the DNR of $N_{SSA ≥ 0.58 μm}$ and no correlation with the mean day-to-night wind speed differences (Fig. 5B and Supplementary Fig. 5). This shows that the magnitude of the DNR of $N_{SSA ≥ 0.58 μm}$ is not controlled by the wind speed, but by a parallel mechanism.

Other atmospheric environmental factors (e.g., rain, RH, air temperature, and atmospheric stability[1]) are known to affect the production and number of SSA, but we found no evidence that they are the main drivers of the observed diel cycle. First, rain suppressed the cycle (Fig. 3A), as it is a known washout mechanism of aerosols. Second, links between the diel changes of RH and air temperature to the $N_{SSA}$ diel cycle were explored. If RH or the air temperature were the main drivers of the diel cycle, by affecting bubble and droplet evaporation, or SSA sizes, we expect to have a clear relationship between the DNR of $N_{SSA ≥ 0.58 μm}$ and the mean day-to-night differences of RH and air temperature, but we found no correlation between them (Supplementary Fig. 5). Therefore, while RH and air temperature can affect bubble evaporation, and in consequence, the production rate of SSA, they are most likely not driving the diel cycles in $N_{SSA ≥ 0.58 μm}$.

The atmospheric stability that influences the transport of aerosols from the ocean surface upward does not fully explain the diurnal cycle either. First, under most atmospheric conditions, concentrations of SSA with $D_{dry}$ < 10 μm are well mixed in the marine boundary layer, showing little variation with height[4]; hence, assuming a constant production of SSA, a change in atmospheric stability will most likely not cause a change in $N_{SSA}$. In addition, during the Taiwan – Fiji transect we took photos of the sky and we could see that the cycle appeared in three distinct atmospheric states: with clear skies at low wind speeds, with overcast conditions, and with trade cumulus throughout the day (see Supplementary Fig. 6). Especially, that a cycle is observed even if the morning is overcast (Supplementary Fig. 6B), and that

cycles are evident also when there is no air temperature variability (Supplementary Fig. 5). These observations suggest that atmospheric stability does not play a significant role in driving the diel cycles. Changes in AMBL height do not fully explain the diel cycles in $N_{SSA}$ either. If height variations within the AMBL were driving the diel pattern in $N_{SSA ≥ 0.58 μm}$, given there are no changes in the production between day and night, the diurnal concentration changes should follow those seen in the AMBL. For the period with clear diel cycles in $N_{SSA ≥ 0.58 μm}$ shown in Fig. 3A, we did not find AMBL height (using ERA5 reanalysis data[36]; see "Methods") diurnal changes that follow those we measured in $N_{SSA ≥ 0.58 μm}$, and the changes we found of the AMBL height were <40 m for a mean height of 3735 m (i.e., ~1%; Supplementary Fig. 7). We cannot exclude a possible influence, but it is unlikely that atmospheric stability or changes in the AMBL height are driving the diel cycle.

Finally, we did not find a significant linear relationship between the intensity of photosynthetically active radiation and the DNR of $N_{SSA ≥ 0.58 μm}$ (Supplementary Fig. 8). While we cannot discard condensational growth after sunrise, the growth rates are too slow (<6 nm h⁻¹)[37,38] to account for the increase in $N_{MA}$ within 30 min after sunrise (Fig. 3A). Furthermore, condensational growth cannot explain the decrease in concentration at sunset. Thus, atmospheric variables, while they can affect the production, size, and transport of SSA, do not seem likely to drive the observed $N_{SSA ≥ 0.58 μm}$ diel cycles.

Within the ocean surface, changes in salinity and sea surface temperature (SST) can also affect SSA production[20,21]. However, salinity in the tropics has typical diurnal anomalies of only 0.005 psu[39], and this anomalies will occur gradually throughout the day with increasing evaporation and not within 30 min after sunrise or sunset. We also did not find strong diurnal salinity changes nor a correlation with the DNR of $N_{SSA ≥ 0.58 μm}$ ($R^2$ = 0.01; Supplementary Fig. 9). Similarly, although SST diurnal variations are known[40], the rate of temperature change is much slower and can be observed later in the day (ref. [40] and Supplementary Fig. 10) compared to the observed increase in $N_{SSA ≥ 0.58 μm}$. In addition, we observed the $N_{SSA ≥ 0.58 μm}$ cycle in conditions of wind speeds above 10 m s⁻¹ (Fig. 3A and Supplementary Fig. 6), where SST diurnal changes are expected to be <0.25 K[40], and in morning overcast conditions (Supplementary Fig. 6B), when no significant changes in SST are expected. We also did not find a correlation with the DNR of $N_{SSA ≥ 0.58 μm}$ ($R^2$ = 0.02; Supplementary Fig. 9). We did find a positive correlation of the DNR of $N_{SSA ≥ 0.58 μm}$ to the daily mean

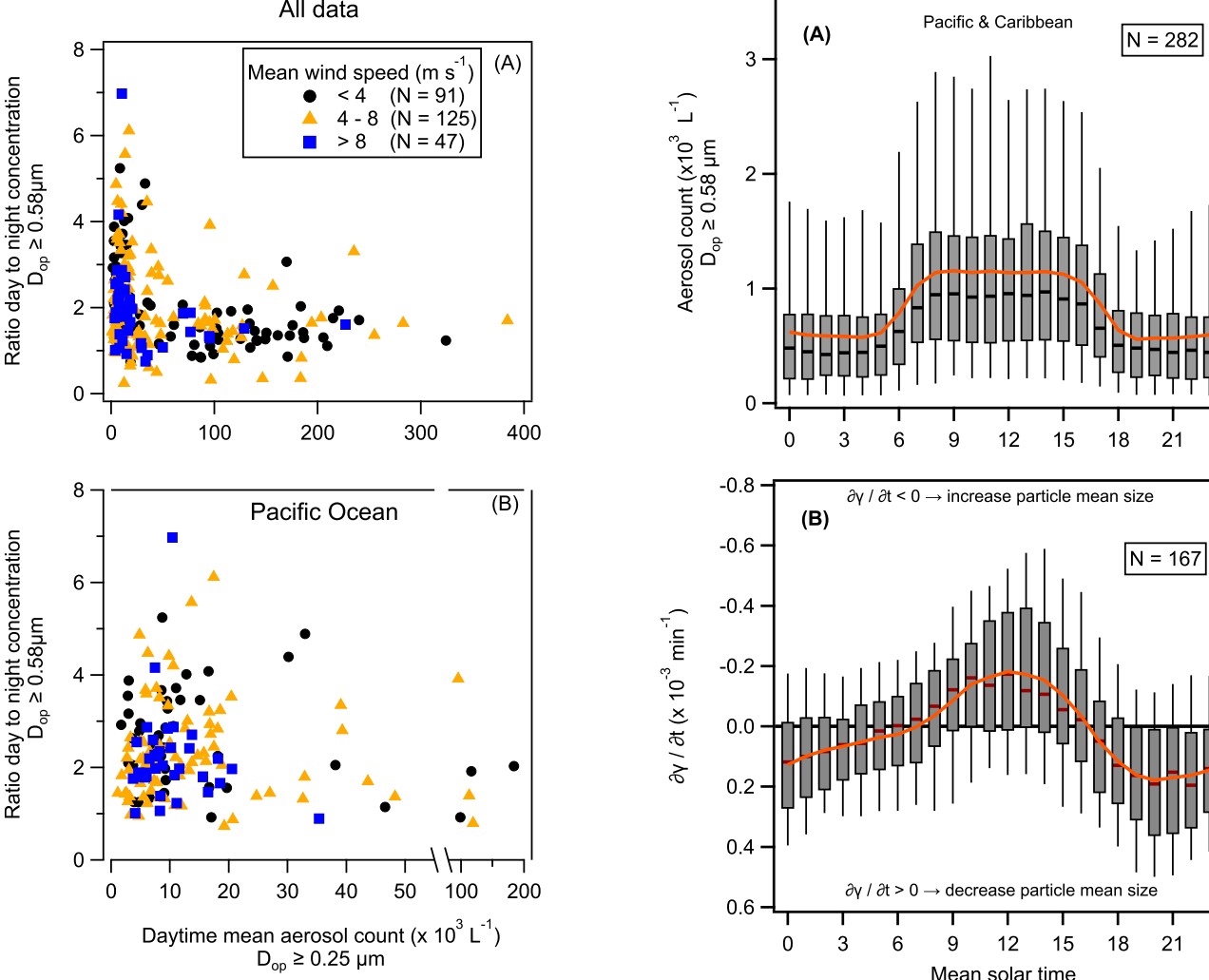

**Fig. 5 Dependence of the day-to-nighttime ratio (DNR) of the sea spray aerosol number concentration with an optical diameter ($D_{op}$) ≥ 0.58 μm ($N_{SSA ≥ 0.58 μm}$) on total daytime aerosol count for three different wind speed regimes.** For daily mean wind speeds below 4 ms$^{-1}$ (black circles), between 4 and 8 ms$^{-1}$ (orange triangles), and above 8 ms$^{-1}$ (blue squares) for **A** all the data, and **B** a clean atmosphere (i.e., only in the Pacific Ocean at least 100 km away from the continents). *N* refers to the number of days analyzed. While there is no clear relationship between the DNR of $N_{SSA ≥ 0.58 μm}$ and the wind speed, the DNR tending to one as the total aerosol count increases shows that long-range transport of aerosols can mask the diel cycle.

**Fig. 6 Day-to-night changes of the sea spray aerosol number concentration with optical diameters ($D_{op}$) ≥ 0.58 μm ($N_{SSA ≥ 0.58 μm}$) and the rate of change of γ ($∂γ/∂t$ (hr$^{-1}$)). A** Box plot analysis of the 282 days with the 24-h cycle in the Pacific Ocean shows the compelling statistical significance of the day and night counts. **B** Box plot analysis of the γ rate of change (vertical axis inverted) for the 167 days (near the Pacific islands, Japan, and Taiwan there is no data). Statistically significant day to night variation is readily discernible. The box plots show the median, and the 5th, 25th, 75th, and 95th percentiles; the orange line is the mean. *N* refers to the number of days analyzed.

SST ($R^2 = 0.89$; Supplementary Fig. 11), consistent with the recent report showing that higher SST enhances SSA generation[21]. This suggests that SST is not likely to drive the $N_{SSA ≥ 0.58 μm}$ diel cycles, but it does play a role in modulating the intensity of the DNR.

These findings suggest that neither atmospheric nor oceanic environmental factors can fully explain the $N_{SSA ≥ 0.58 μm}$ diel cycles, and consequently the greater amount of aerosol we measured in all the other diameters. While the wind speed and SST affect the diel cycles, they do not drive them. Furthermore, the ubiquitous occurrence of the diel cycle, whether in the presence of anthropogenic pollutants (Fig. 3A, B and Supplementary Fig. 4) or in clean conditions (with $N_{MA} ≈ 10$ cm$^{-3}$), implies that anthropogenic and continental sources are also not the cause. Together with the high variability of the DNR of $N_{SSA ≥ 0.58 μm}$ in clean conditions of the Pacific Ocean (Fig. 5B), it

suggests that the diel cycle may be triggered by changes in the ocean surface itself.

**Diel patterns of SSA are correlated to the size of seawater particles.** The identified $N_{SSA}$ diel cycles in the lower atmosphere were accompanied by distinct diel cycles of the near-surface ocean light attenuation wavelength dependence of $c_p$ (expressed as particle size index γ; Fig. 3C). $c_p$ is an inherent optical property that depends, theoretically, on all the particles present in the water column (i.e., autotrophic and heterotrophic micro-organisms, as well as detritus and mineral particles). Given $c_p$ is wavelength-dependent, γ is found via a power-law fit[41,42]:

$$c_p = c_p(\lambda_0)\left(\frac{\lambda}{\lambda_0}\right)^{-\gamma} \quad (2)$$

where $c_p(\lambda_0)$ is $c_p$ at a reference wavelength $\lambda_0$ and $\gamma$ is the spectral slope of $c_p$. Variability of $\gamma$ indicates changes in the median particle size and is most sensitive to particles in the range of 0.22–20 µm[41,43], with smaller $\gamma$ associated with larger median particle size (similar to the Ångström exponent for aerosols). Figure 3C shows that there is a congruency between the $N_{SSA \geq 0.58\,\mu m}$ diel cycle and the daily changes of $\gamma$. Note that $\gamma$ reached a maximum value (i.e., minimum mean diameter) right after sunrise when the $N_{SSA \geq 0.58\,\mu m}$ began to increase, and a minimum value (i.e., maximum mean diameter) before sunset when $N_{SSA \geq 0.58\,\mu m}$ decreased (Fig. 3C). We, therefore, calculated the rate of change of $\gamma$ ($\partial\gamma/\partial t$ (hr$^{-1}$); see "Methods" for concurrent days (167 days in total) where the $N_{SSA \geq 0.58\,\mu m}$ was also measured (Fig. 6A, B). We found a parallel behavior, with a continuous decrease in the mean particle diameter at nighttime ($\partial\gamma/\partial t$ (hr$^{-1}$) > 0) when $N_{SSA \geq 0.58\,\mu m}$ were lowest, and a continuous increase ($\partial\gamma/\partial t$ (hr$^{-1}$) < 0) between 07:00 and 17:00, when $N_{SSA \geq 0.58\,\mu m}$ were highest. We also found an inverse relationship between $N_{SSA \geq 0.58\,\mu m}$ and $\partial\gamma/\partial t$ (hr$^{-1}$) (Supplementary Fig. 12; $R^2 = 0.51$), with higher $N_{SSA \geq 0.58\,\mu m}$ when $\partial\gamma/\partial t$ (hr$^{-1}$) is negative.

In the Pacific Ocean, most $\gamma$ values were above 0.8 (Supplementary Fig. 13A) reaching 1.4 with a strong latitudinal dependence (Supplementary Fig. 13B). In the Atlantic Ocean (Supplementary Fig. 13A), its values were below 0.8, indicating a larger mean particle size and thereby suggesting the presence of larger planktonic species than in the Pacific Ocean. This indicates that it is not the size of the planktonic species which is the significant factor but $\partial\gamma/\partial t$.

Similar diel cycles of $\gamma$ in the open ocean have been previously documented[42–44]. Changes in $\gamma$ can be driven by different reasons; for example, cell growth, division, and aggregation, selective changes in particle size or concentration due to a balance between primary production and cell loss due to grazing and viral pressure, or changes in the refractive index of the cell population, which is related to their carbon content. In the Equatorial Pacific, phytoplankton cells were reported to contribute from 50 to 83% of the total particulate organic carbon concentration[45], and the changes in $c_p$ and $\gamma$ throughout the day, were principally driven by phytoplankton growth and division. During daytime, cells photosynthesize, fix inorganic carbon, and accumulate carbohydrates and lipids which are respired during the night, subsequently cells generally divide. We found that the daytime increase in $\gamma$ is associated with cell growth or aggregation of pico-phytoplanktonic populations, as the $\gamma$ changes measured during $N_{SSA \geq 0.58\,\mu m}$ diel cycles (in the western Pacific) can be attributed to variations in size of particles smaller than ~1 µm (see "Methods" and Supplementary Fig. 14). We also observed a daily increase in particulate organic carbon (Supplementary Fig. 15). The decrease in mean size at night can be due to selective grazing, cell division, or virus-induced lysis[46,47].

**Microbial processes might control SSA production**. The $N_{SSA}$ diel cycles were observed across high and low chlorophyll $a$ (chl-$a$) concentrations regions (Fig. 1), suggesting that processes in oligotrophic waters can be the dominant factors driving the cycles. In oligotrophic oceanic regions, the dominating marine phytoplankton are picocyanobacteria[48,49], in which growth and division have been shown to be well synchronized with daylight[50] and to contribute a large proportion of the daily variations of $c_p$[51]. During photosynthesis, which begins within tens of seconds after light exposure, and carbon fixation within minutes[52], eukaryotic phytoplankton and cyanobacteria can secrete extracellular polymeric substances (EPS). EPS are a diverse array of large molecules which form a major component of the dissolved organic carbon pool in the ocean, and have been implicated in the formation of

biofilms and marine snow[53]. Recently, in a different context, EPS released by bacteria were found to increase bubble lifetime, thereby dramatically decreasing their film thickness, and yielding more numerous and transportable droplets at burst than those produced by clean bubbles (see Fig. 1 in ref. [54]). This observations can be directly linked to SSA formation since it is directly related to film drops formed by the fragmentation of the thin fluid cap film[55].

Consequently, we conjecture that the diel changes in $N_{SSA}$ are controlled by microbial processes in the ocean surface. Such processes whether at the near-surface water-AMBL interface, or in the upper several meters of the ocean through bubble scavenging of the excreted EPS from bacteria, may affect bubble bursting dynamics, changing the number and size of the emitted droplets and therefore $N_{SSA}$. The AMBL typically mixes within an hour. A characteristic time is in the order of 20 min for a 500 m AMBL[4]. Hence, significant changes in the production of droplets can be expected to happen within a similar timeframe in which we observed the transient increase and decrease of $N_{SSA}$. Although we do not pinpoint a direct role of marine bacteria affecting bubble-bursting dynamics in the open ocean surface, recent studies have shown a possible role for bacteria modulating ocean surface properties[53,56–59] and bubble-bursting dynamics[54,60,61].

Biological activity near the ocean surface has been shown to affect SSA abundance and chemical composition[13,14,62–64]. In particular, Long et al.[13] and Keene et al.[14] explored differences in SSA production between day and night, generating SSA artificially from seawater on ships at sea. Differently from our study, they reported an increase in the number and mass of SSA during the day in biologically active waters and not in oligotrophic ones. They attributed the increased number concentration of SSA following sunrise to biogenic surfactants emitted either from biota or produced photochemically. Other studies focused on links between biological activity and SSA production have shown contradicting results. In the laboratory, on the one hand, SSA production increased when diatomaceous exudate[62] or actively growing bacteria and phytoplankton[65] were present in the seawater; and on the other hand, a decrease in production occurred when surfactants were present[12], or when there was an increase in algal biomass[66]. A satellite observation study[67], reported a negative correlation of chl-$a$ concentration to coarse-mode aerosol optical depth in the south Pacific Gyre on a seasonal time scale and suggested that high biological activity suppresses particle production.

We presented in situ evidence that the number concentration of SSA with $D_{op} \geq 0.25$ µm (at RH < 40%; for dry NaCl particles it is equivalent to $D_{geo} \geq 0.26$ µm, see Supplementary Fig. 2) have a distinct 24-h cycle (clearly visible for $N_{SSA \geq 0.58\,\mu m}$), with higher number concentrations during the day than at night over vast areas of the Pacific Ocean, the Caribbean Sea, and the Atlantic Ocean. Our results show an inverse correlation between the ambient aerosol concentration and the magnitude of the observed cycle (Fig. 5), as the ambient aerosol concentration increases the day-to-night ratio tends to one, suggesting that over areas with high aerosol concentration (e.g., with a strong contribution from long-range transport of continental aerosol) the cycle is present but masked. In addition, the increase of the magnitude of the diel cycle for larger diameters (Figs. 1 and 2) suggests that the longer lifetimes of smaller aerosols (the lifetime of aerosols, excluding rain events, is inversely proportional to their size) also contributes to masking the cycle, particularly for the small sizes. While it is known that SSA formation originates from the interaction of wind and waves, with bulk oceanic properties (i.e., SST, salinity, chl-$a$) affecting it[5,21], here we show that there is a concomitant daily mechanism that modulates aerosol concentration in a 24-h rhythm with the magnitude of the daytime concentration increase

positively correlated with SST. While we do not provide (or possess) direct measurements of near-surface water microbial processes, the parallel increase of the mean particle size within the ocean surface during the day, driven by photosynthetic growth and secretion of extracellular polymeric substances, points towards a possible link between microbial processes at the ocean surface and the $N_{SSA}$ cycle.

Although the SSA number fraction (for $D > \sim0.5\,\mu m$) in the AMBL, by number, is <20%[1,22], small concentration changes of big aerosols can drive relatively large changes in cloud sizes, lifetime, and rain yields[68,69], especially over pristine areas where clouds are aerosol-limited[70]. Consequently, the discovery of the diel cycle in $N_{SSA}$ opens many new questions for future studies to elucidate the mechanism underlying this phenomenon and the direct impact of marine biological processes on the physical properties of the surface ocean, and the link to aerosol fluxes and properties. Moreover, on a larger scale, the connection to cloud properties and consequently energy fluxes and climate.

## Methods

**Schooner _Tara_**. Measurements were conducted aboard the R/V _Tara_ during the first year of the _Tara_ Pacific expedition[1,15,16]. The R/V _Tara_ is a 36-m long, 10-m wide aluminum hull schooner with two 27-m masts, equipped with a meteorological station (Station Bathos II, Météo France) measuring air temperature, relative humidity, and pressure. The station is located on the stern around 7 m above sea level, the wind speed and direction are measured at the top of the mast, ~30 m above sea level (asl), and a thermo-salinometer (Sea-Bird Electronics SBE45 MicroTSG) measures sea surface temperature (SST) and salinity with its main water entrance located about 0.5–3 m under the sea surface (depending on ocean conditions). The intensity of photosynthetically active radiation (PAR; wavelengths between 400 and 700 nm) was measured next to the meteorological station by a QCR-2150 (Biospherical Instruments Inc.). The meteorological station recorded frequencies are listed in Supplementary Table 2. The SST and salinity were measured at 0.1 Hz and processed to 1 min averages. The PAR is analyzed to 1 min average of 1 Hz measurements.

_Continuous aerosol instrumentation and inlet_. A detailed description of the aerosol instrumentation during the expedition can be found in Flores et al.[1]. In short, an optical particle counter (OPC; EDM-180 GRIMM Aerosol Technik Ainring GmbH & Co. KG, Ainring, Germany), for continuous aerosol size distribution measurements (from 0.25 to 32 μm, sorted into 31 bins), and a custom-made aerosol filter system consisting of four 47 mm filter holders and one vacuum pump (Diaphragm pump ME 16 NT, VACUUBRAND BmbH & Co KG, Wertheim, Germany) were installed aboard R/V _Tara_. Two separate inlets, located next to each other, were constructed out of conductive tubing of 1.9 cm inner diameter and a funnel (allowing the collection of all diameters) and mounted on the rear backstay of _Tara_. For the Atlantic Ocean measurements, from Lorient, France to Miami, USA, the inlet was installed half way up the backstay (~15 m asl) and after Miami, the inlet was relocated to the top of the backstay (~30 m asl).

The OPC measures single particles at 683 nm and it was calibrated at the refractive index of polystyrene latex spheres. It collects the scattered light using a wide-angle collector optic at a mean scattering angle of 90°; the optical design smoothes out Mie scattering resonances and reduces the sensitivity to particle shape. A Nafion dryer was installed before the OPC, which reduced the sampled air relative humidity to below 40%[1]. The flow through the OPC was 1.2 L/min and it produced a particle size distribution every 60 s. Particle loss corrections to the OPC were applied taking into account the inlet's length and the tube's internal diameter using the particle loss calculator[71]. During the Atlantic crossing and the Keelung to Fiji leg, a scanning mobility particle sizer was functioning and the overlap region with the OPC was used to check and correct it if necessary; see Flores et al.[1] for details.

The filters from the custom-made system were changed, in general, twice a day, collecting aerosols for periods of at least 12 h. The filter holder for the analysis presented here contained 0.8 μm polycarbonate filters (ATTP04700, Millipore) that were stored at room temperature in PetriSlide dishes preloaded with absorbent pads (Millipore, PDMA04700) to keep the filters dry while stored. The flow through the filter was 30 L/min for the filters analyzed here. Blank filters were collected every few days by placing filters on the filter holders, closing the system for a few seconds, reopening the holders, and storing them in a PetriSlide dish.

**Continuous water measurements**. The R/V _Tara_ was equipped with an ocean surface flow-through autonomous sampling system, similar to the one installed during the _Tara_ Oceans Expeditions, to measure sea surface physical and bio-optical properties as described in ref. [17]. The inline system consisted of a Sea-Bird Electronics SBE45 MicroTSG for measurements of sea surface temperature (SST) and salinity and an AC-S spectrophotometer (WET Labs, Inc.) measuring

hyperspectral particulate absorption ($a_p$) and particulate attenuation ($c_p$) with a ~4 nm resolution, and an ECO-BB3 (WetLabs Inc.) set in a BB-box of ~4.5 L measuring particulate backscattering at three wavelength (470, 532, and 650 nm), altogether mounted in an autonomous setup described in Dall'Olmo et al.[72] and Slade et al.[44]. The size range of the measured particles is >0.2 μm, but contribution of particles >20 μm is assumed negligible. Particulate organic carbon (POC) concentrations were computed from $c_p$[73] and chlorophyll-_a_ concentrations were estimated from the particulate absorption line height[17]. In addition, a particle size index ($\gamma$), an estimate of the mean particle size in the ocean near-surface waters, was calculated using the wavelength-dependency of $c_p$ and that its spectral shape can be approximated as a power law (see main text).

**Chlorophyll-_a_ along _Tara_'s route**. The chlorophyll-_a_ concentration along _Tara_'s route was calculated using the AC-S[17] and to approximate the chl-_a_ concentrations when the AC-S was not functioning, we used the level 3 SNPP-VIIRS satellite monthly data maps. For each month, we used _Tara_'s hourly location to first extract a $0.2 \times 0.2$ degree area for each point, then this area was averaged to get a corresponding chl-_a_ concentration at each point. Finally, a 24-h average was taken along _Tara_'s route. Figure 1C shows the satellite calculated chl-_a_ concentration, and the in situ chl-_a_ inferred from AC-S measurements.

**Diel pattern per diameter**. To understand the diurnal changes driving the greater amount of SSA during the day, we analyzed each bin from the OPC. We only included data in the Pacific Ocean and when _Tara_ was at least 100 km away from land (islands included). The OPC has 31 bins for measurements between 0.25 to 32 μm, in Supplementary Fig. 2 we show the box plot analyses for 16 bins from the OPC, up to 3.0 μm. Supplementary Fig. 2 shows that the diurnal behavior becomes evident around $D_{op} \geq 0.58\,\mu m$.

**Diurnal cycle near Niue Island when _Tara_ was anchored**. Similar to Fig. 3 in the main text, a diel cycle of $N_{SSA \geq 0.58\,\mu m}$ was detected while _Tara_ was anchored near Niue Island (19°03′14″S 169°55′12″W; Supplementary Fig. 3). Supplementary Fig. 3 also shows a diurnal cycle of $\gamma$, increasing during nighttime (smaller particle mean diameter) and decreasing during daytime (bigger particle mean diameter).

**Air mass back trajectory analysis**. The presented 48 h back trajectories in Fig. 3M were calculated using the NOAA's HYSPLIT atmospheric transport and dispersion model[26,27]. They represent the average trajectories of the 'Ensemble option' that were calculated based on an endpoint at 250 m height. We chose the 'Ensemble option' to have a better representation of where the air masses were coming from. We did not use a lower starting height as the minimum height for the optimal configuration of the ensemble is 250 m.

**Scanning electron microscope with energy disperse X-ray analysis**. Using scanning electron microscopy with energy-disperse X-ray analysis (SEM-EDX) and a similar particle classification scheme as described in Laskin et al.[28], we classified each particle into one of five major classes of aerosols: (i) _Sea salt_: [Na] greater than all other elements detected (except Cl); (ii) _Metals with Na_: [Na] present but [Na] < [Al, Si, K, Ca, S]; (iii) _Sulfate/SeaSalt_: [Na] > [Al, Si, K, Ca] but [Na] < [S]; (iv) _Sulfates_: [Na]=0 and [S] >0; and (v) _Other_: all remaining particles.

To perform the SEM-EDS analysis, we used a Zeiss Sigma500 SEM with a Bruker XFlash®-6|60 Quantax EDS detector, and the Bruker ESPRIT feature software package for automatic particle detection and chemical classification in EDS.

The SEM was set at a working distance of about 7.5 mm (±0.1), an accelerating voltage of 8.0 kV, an aperture size of 60 μm, and a magnification of 2000. The backscatter detector was used to acquire the images. For each filter four images, covering a total of 2471 μm² surface area, were taken and each particle above a minimum area of 0.08 μm² was counted and an EDS spectrum acquired. After the acquisition of the images and EDS spectra, we took only the particles that had an average diameter >0.3 μm and for each of their corresponding EDS spectra, the method described in ref. [74] was used to calculate the mass percent of each detected element. We excluded C from the mass percent calculation since the filters were made of polycarbonate. Following the mass calculation, the method described by Laskin et al. was applied. First, particles containing sodium above 0.01 mass percent ([Na] > 0) were first separated from those without sodium. The Na containing particles with more sodium than any other detected element (besides Cl) were denoted _Sea-salt_. The rest of Na containing particles were subdivided into two classes: _Metals with Na_ if [Na] < [Al], [Ca], [K], [Si], and mixed _SeaSalt/Sulfate_ —if [Na] > [Al], [Ca], [K], [Si] [Na] but [Na] < [S]. The sodium-free particles were assigned to two classes: _Sulfate_ if [S] >0 and _Other_ for the remaining particles. A total of 14,339 particles, where 13,405 have average $D \geq 0.3\,\mu m$, were analyzed. In the 14 daytime filters (for the period between 4 May and 17 May, 2017; see Supplementary Table 1) we counted a total of 7247 particles and 6901 with $D \geq 0.3\,\mu m$. In the 15 nighttime ones we counted a total of 5894 particles with all having $D \geq 0.3\,\mu m$. We had between 80 and 781 particles per filter.

Supplementary Fig. 4A shows the $SSA_{0.58\,\mu m}$ counts per litre calculated using the SEM images (particle count and area imaged) and the total air sampled.

Supplementary Fig. 4B shows histograms of the chlorine mass percentage found in the particles per filter.

**Daytime and nighttime SSA$_{0.58\,\mu m}$ concentration vs wind speed.** In order to understand the role of wind speed in the N$_{SSA \geq 0.58\,\mu m}$ cycle, we separated the Pacific data (for days when a cycle was detected) into daytime (07:00–17:00) and nighttime (19:00–05:00) periods, and binned the total aerosol counts of $D \geq 0.58\,\mu m$ into 2 m s$^{-1}$ bins (Fig. 4a; data within 100 km from continental coasts and Japan was not used to avoid pollution artifacts). There were between 3612 and 23,136 events per bin used. The Atlantic Ocean data were binned into 4 m s$^{-1}$ bins for comparison. There were between 949 and 5056 events per bin used.

**Day to nighttime ratio vs. aerosol concentration at different mean wind speeds regimes.** To explore the relationship between the diel cycles and the aerosol concentration at different mean wind speeds, we quantified the day-to-nighttime concentration ratio for $D \geq 0.58\,\mu m$ vs. the total (using all the bins from the OPC) daytime aerosol concentration and separated them into three mean wind speed regimes: U$_{30m} < 4$, $4 \leq$ U$_{30m} < 8$, and U$_{30m} \geq 8$. After converting the data to mean solar time and taking every 24-h period as independent, we first averaged the total daytime concentration (from 19:00 to 05:00), next we took the DNR for $D \geq 0.58\,\mu m$, and finally took the mean wind speed of the whole 24-h period for the corresponding day. Figure 5 shows the inverse relationship between the day to nighttime concentration ratio and the aerosol loading for all data (Fig. 5A) and clean conditions in the Pacific Ocean (Fig. 5B; data within 100 km from continental coasts and Japan was not used).

**Relationship of atmospheric variables and the SSA diurnal cycle.** To explore the relationship between the diurnal changes is SSA and the air temperature, relative humidity, and wind speed, we calculated the day-to-nighttime ratio for N$_{SSA \geq 0.58\,\mu m}$ and, for each atmospheric variable, the difference between the mean daytime value and the mean nighttime value. The scatter plots are shown in Supplementary Fig. 5. We found no correlation with any of the three atmospheric variables, with $R^2 < 0.002$.

To explore the effect of atmospheric stability that will influence the transport of SSA from the ocean surface upwards, we looked for distinct atmospheric states along *Tara*'s route. Supplementary Fig. 6 shows three representative pictures of the ocean and the atmospheric states during the Keelung – Fiji leg: with trade cumulus (Supplementary Fig. 6A), with morning overcast conditions (Supplementary Fig. 6B), and with clear skies at low wind speeds (Supplementary Fig. 6C). Alongside each picture, we show their respective N$_{SSA \geq 0.58\,\mu m}$, wind speed, and photosynthetic available radiation. The diel cycle in N$_{SSA \geq 0.58\,\mu m}$ occurred irrespective of the atmospheric conditions.

**Photosynthetically active radiation (PAR) vs daytime to nighttime ratio of concentration.** By definition, solar radiation drives diurnal cycles. Therefore, we explored links between the intensity of solar radiation, measured by the average daytime photosynthetically available radiation (PAR), and the DNR of SSA$_{0.58\,\mu m}$ to determine if the intensity of solar radiation has a measurable effect on the total amount of SSA$_{0.58\,\mu m}$. Supplementary Fig. 8 shows the scatter plot of the DNR of N$_{SSA \geq 0.58\,\mu m}$ vs. the average PAR. No significant linear relation between PAR and N$_{SSA \geq 0.58\,\mu m}$ was found. Furthermore, examples of days with similar PAR that showed different daytime N$_{SSA \geq 0.58\,\mu m}$ can be seen in Fig. 3B and Supplementary Fig. 6.

**Atmospheric marine boundary layer height.** The atmospheric marine boundary layer height (ABMLH), in meters, was derived from ERA5 reanalysis data[36]. The ERA5 reanalysis data is provided by the European Centre for Medium-Range Weather Forecasts (ECMWF) with 30-km horizontal resolution and 137 vertical hybrid levels with the high vertical resolution (10–200 m intervals) in the lowermost 3 km above surface. The ERA5 AMBLH data have a 0.25° spatial and a 1-h temporal resolution. We used the latitude and longitude along *Tara*'s route to obtain the closest AMBLH point from ERA5. Supplementary Fig. 7 shows the AMBLH along the orange and blue-shaded transect in the western Pacific between Keelung and Fiji (next to the double-ended arrow) shown in Fig. 3M. We also did not find a correlation between the AMBLH and the measured N$_{SSA \geq 0.58\,\mu m}$.

**Relationship of the surface ocean variables and the diel cyle in SSA.** Similar to the atmospheric variables, we calculated the day-to-nighttime ratio for N$_{SSA \geq 0.58\,\mu m}$ and the mean day-to-night differences for sea surface salinity (SSS), sea surface temperature (SST), and chl-*a*. We found no correlation between any of the variables with $R^2 < 0.02$. Supplementary Fig. 9 shows the scatter plots and the linear fit between the DNR of N$_{SSA \geq 0.58\,\mu m}$ and the three oceanic variables.

**Diurnal anomalies of SST and particulate organic carbon.** We calculated the diurnal anomalies of the sea surface temperature (Supplementary Fig. 10) across *Tara*'s route and of the rate of change for particulate organic carbon ($\partial POC/\partial t$ (hr$^{-1}$); Supplementary Fig. 15) across the Keelung to Fiji leg. The average between midnight and 05:00 is the baseline for each variable. To avoid continental influence on the SST anomaly, this analysis was done only in the open ocean and near the Pacific islands

except Japan and Fiji. To calculate the anomalies, first we converted the data to mean solar time and took every 24-h period as independent, then we took the mean between midnight and 05:00 and subtracted the mean value at each hour, and finally took the mean at each hour.

**Rate of change of $\gamma$ ($\partial\gamma/\partial t$).** As mentioned above, $\gamma$ is an indicator of the size distribution among particles (<20 μm in diameter) in the ocean surface. From Fig. 3 we see $\gamma$ decrease at daytime (i.e., the sizes of the plankton increase) and increase over nighttime. To quantify the intensity and timing of this change over a full day, we calculated the rate of change of $\gamma$. First, to fill in data gaps that correspond to periods when the AC-S was measuring filtered seawater for calibration purposes (normally shorter than 30 min), we did a linear interpolation. Data gaps larger than 30 min were not interpolated. Then, each continuous segment was smoothed applying a low-pass digital filter with a pass band frequency of 18 h. Then the rate of change $\partial\gamma/\partial t$ (hr$^{-1}$) was calculated. Finally, Fig. 6B shows a box plot analysis for 167 days across *Tara*'s route and there was at least 23 h of the AC-S data, and Supplementary Fig. 12 shows a scatter plot of N$_{SSA \geq 0.58\,\mu m}$ vs. $\partial\gamma/\partial t$ (hr$^{-1}$) for the Pacific Ocean data.

**Particle size index $\gamma$ in different parts of the ocean.** To understand the differences in $\gamma$ across *Tara*'s route and how it might be related to SSA$_{0.58\,\mu m}$ production, we calculated the average $\gamma$ in 24 h cycles. Supplementary Fig. 13 shows two different scenarios: Supplementary Fig. 13A shows the average $\gamma$ measured in the Atlantic Ocean and in the Pacific Ocean. Supplementary Fig. 13B shows the average $\gamma$ measured during the Keelung – Fiji leg separated by different latitude ranges and the average $\gamma$ while *Tara* was anchored near Niue Island.

**Contribution of small, ~<1 μm particles, to $\gamma$ variations.** In order to determine the causes of the daytime $\gamma$ changes, we calculated the contribution of small particles to the size changes observed in $\gamma$. To do so, we calculated the backscattering ($b_{bp}$) to total particulate scattering ($b_p$) ratio ($b_{bp}$:$b_p$) at $\lambda = 532$ nm.

$$b_{bp}{:}b_p(\lambda) = \frac{b_{bp}(\lambda)}{c_p(\lambda) - a_p(\lambda)} \qquad (3)$$

where $c_p$ is the particulate attenuation and $a_p$ the particulate absorption. Since smaller particles have a higher $b_{bp}$:$b_p$, the $b_{bp}$:$b_p$ can serve as a proxy for the contribution of small (~<1 μm particles) to the bulk particle size index $\gamma$ variation. $b_{bp}$:$b_p$ and $\gamma$ can be taken as two independent variables since the former is the ratio of the amplitudes of the backscattering and the total scattering at a single wavelength (here 532 nm), while the latter refers to the spectral shape of $c_p$.

Since the $b_{bp}$ measurements with the ECO-BB3 sensor have low signal-to-noise ratio due to its sensitivity to bubble in the water line and accumulation of particles in the sensor, the $b_{bp}$:$b_p$ was first averaged for 1-h periods and then smoothen with a 5 h moving average. Supplementary Fig. 14 shows the co-occurrence of a diel cycle in $\gamma$ and the $b_{bp}$:$b_p$ in the western Pacific Ocean between 15°N 137°E and 17.6°S 177.4°E, where diel cycles in N$_{SSA \geq 0.58\,\mu m}$ were also observed. Twardowski et al.[75] showed that for $\gamma < 0.8$ the backscattering ratio is mostly affected by changes in refractive index of particles, for $\gamma > 0.8$ the size of particles become a major contributor to the backscattering ratio. Therefore, the observation of a diel cycle of the backscattering ratio synchronized with $\gamma$ and daylight in these oligotrophic waters, suggests that the diel cycle in ocean particle size is mainly due to changes in the size of pico-phytoplanktonic populations. Pronounced diel cycles of pico-phytoplanktonic populations (e.g., cyanobacteria) in the Equatorial Pacific, have been previously shown[50,51,76].

**Computer code.** Codes were written in the software program IGOR Pro. v7.08 to analyzed the data. The codes are available on this link: https://doi.org/10.34933/wis.000392.

**Reporting summary.** Further information on research design is available in the Nature Research Reporting Summary linked to this article.

## Data availability

All data for this article have been deposited in the open access Weizmann Institute's institutional repository and are available in this link: https://doi.org/10.34933/wis.000392.

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

## Acknowledgements

Special thanks of gratitude to the R/V *Tara* crew. The authors would like to thank the *Tara* Pacific Coordinators for their valuable comments on the manuscript. This research was supported by a research grant from Scott Jordan and Gina Valdez, the De Botton for Marine Science, the Yeda-Sela center for Basic research, and a research grant from the Yotam Project. Special thanks to the Tara Ocean Foundation, the R/V Tara crew, and the Tara Pacific Expedition Participants (https://doi.org/10.5281/zenodo.3777760). We are keen to thank the commitment of the following institutions for their financial and scientific support that made this unique Tara Pacific Expedition possible: CNRS, PSL, CSM, EPHE, Genoscope, CEA, Inserm, Université Côte d'Azur, ANR, agnès b., UNESCO-IOC, the Veolia Foundation, the Prince Albert II de Monaco Foundation, Région Bretagne, Billerudkorsnas, AmerisourceBergen Company, Lorient Agglomération, Oceans by Disney, L'Oréal, Biotherm, France Collectivités, Fonds Français pour l'Environnement Mondial (FFEM), Etienne Bourgois, and the Tara Ocean Foundation teams. Tara Pacific would not exist without the continuous support of the participating institutes. The authors also particularly thank Serge Planes, Denis Allemand, and the Tara Pacific consortium. A.K. was supported by NSF AGS-1639868. This study has been conducted using E.U. Copernicus Marine Service Information and Mercator Ocean products. G.D. is supported by the European Research Council (ERC) project conStRaining the EffeCts of Aerosols on Precipitation (RECAP) under the European Union's Horizon 2020 research and innovation program with grant agreement No. 724602. F.L. is supported by Sorbonne Université, Institut Universitaire de France and the Fondation CA-PCA. The in-line and atmospheric optics dataset was collected and analyzed with support from NASA Ocean Biology and Biogeochemistry program under grants NNX13AE58G and NNX15AC08G to the University of Maine. M.S.B. is supported by NSF OCE1829831. N.L.Y. acknowledges support from the Women Bridging position and the Sustainability and Energy Research Initiative (SAERI), Weizmann Institute of Science. This is publication number # 13 of the *Tara* Pacific Consortium. Permits to access the areas sampled were given by Ministerio de Ambiente, Republica de Panama on 13-06-2016, No. SE/AP-18-16; Parques Nacionales Naturales de Colombia on 22-02-2016, Codigo: AMSPNN_FO_16, No: 2016230420500002E; Armada de Chile Servicio Hidrografico y Oceanografico on 29-08-2016, No. 13270/24/457/Vrs.; Convention sur le commerce international des espèces de flore et de faune sauvages menacées d'extinction, Polynésie Française (France), on 03/11/2016 for No. FR1698700198-E and on 21/11/2016 for No. FR16987002189-E; Cook Island Research Committee, Cook Islands, on 12/09/2016, file ref: 510.3; Ministry of Natural Resources and Environment, Samoa, on 29/11/2016, No. SAMC16012; Administration superieure des Iles Wallis et Futuna, Terrirtoire des Iles Wallis et Futuna, on 24/11/2016, No. 2016-527; Ministry of Foreign Affairs, Trade, Tourism, Environment and Labor, Government of Tuvalu, on 19/10/2016, No. 2016/753527; Environment and Conservation Division, Republic of Kiribati, on 24/11/2016, No. 015/16; Department of Resources & Development, Federated States of Micronesia, on 19/01/17, No. CFM17-01-01; Department of Agriculture, Guam, on 4/2/2017, No. SC-17-004; Ministry of Agriculture, Forestry and Fisheries, Japan, on 18/1/2017, No. 019; Ministry of Foreign Affairs, Republic of Fiji, on 11/07/2017, No. 456/2017.

## Author contributions

Conceptualization, J.M.F., O.A., A.K., M.T., E.B., F.L., G.G., Y.R., A.V., and I.K.; data curation, J.M.F., G.B., N.H., E.B., and F.L.; formal analysis, J.M.F. lead, G.B., G.D., N.H., and F.L. supporting; funding acquisition, E.B., F.L., G.G., A.V., and I.K.; investigation, J.M.F., G.B. N.H., N.L.-Y., and F.L.; project administration, J.M.F., G.B., M.T., E.B., F.L., G.G., A.V., and I.K.; supervision, J.M.F., E.B., F.L., G.G., Y.R., A.V., and I.K.; visualization, J.M.F., A.V., and I.K.; writing—original draft, J.M.F.; writing—review & editing, J.M.F., A.K., O.A., I.K., and A.V. lead, G.B., G.D., N.H., M.S.B., N.L.-Y., M.T., E.B., F.L., G.G., S.R., C.R.V., and Y.R. supporting.

## Competing interests

The authors declare no competing interests.
