## [Peer Review File · Nature Communications]

REVIEWER COMMENTS

Reviewer #1 (Remarks to the Author):

Flores et al. present a set of aerosol concentration measurements obtained in the marine boundary layer during a multi-leg research cruise across the Atlantic Ocean, Caribbean Sea and Pacific Ocean. The main finding by the authors is that the aerosol concentration measured exhibits a diel cycle over the vast oligotrophic regions of the tropical Pacific, with maximum aerosol concentrations during the day and lower concentrations at night. Although this is the first time such an observation has been made in the ambient marine boundary layer it is likely that the processes behind this have been captured previously (e.g. Long et al., 2014 and Keene et al., 2017). I certainly think that this data is interesting and worthy of publication since it builds on these previous studies, moving our understanding from the laboratory to the field. As such, I would be supportive of this paper being published in Nature Communications. However, there are some issues I would like to see addressed first.

Major Comments

My first issue with the manuscript is the context (or lack of) that the authors place their work in. In 2014 Long et al., published a paper centered on observations of diel variability in nascent sea spray aerosol number fluxes in biologically productive waters. These authors attribute this diel cycle in sea spray aerosol number production to interactions between bubble plumes and biologically or photochemically derived surfactants. The observations made by Long et al. (2014) received further support from the expanded data provided in Keene et al. (2017). The authors of the manuscript under consideration in this review are clearly aware of these two studies since they are cited in the introduction. However, essentially no comment as to the findings of these studies and the results, which are clearly critically relevant to the manuscript in question, have been made by the authors in either their introduction or their discussion. This must be rectified.

My second issue is with the terminology used by the authors. Sea spray aerosol, which the authors use in the title and throughout the manuscript are aerosols emitted directly into the atmosphere following production at the ocean surface either through the mechanism of bubble bursting and the ejection of droplets or when water is ripped directly off of wave crests by the wind. As such, sea spray aerosol has a very clearly defined origin. Marine aerosol, which the authors have chosen not to use in this article, is the background aerosol in the boundary layer over the oceans. It contains numerous particle classes including sea spray aerosol, secondary aerosols following the oxidation of dimethyl sulphide or other aerosol precursors, local anthropogenic aerosols (e.g. ship emissions), aerosols of terrestrial origin which may be either natural (e.g. mineral dust emissions) or anthropogenic (e.g. biomass burning and industrial emissions) etc. The point is that this is a mixture of numerous aerosol sources and most critically this is what the authors are quantifying when they make their measurements in the ambient marine boundary layer. They have determined a diel cycle in marine aerosols over large parts of the Pacific Ocean. The authors have done a commendable job in trying to ascertain which of the particle classes present in marine aerosol might be responsible for their observations and it may well be a diel cycle in sea spray aerosol that is driving their observations. However, they cannot be sure since the data they present is limited to aerosol size distribution measurements in the marine boundary layer. The repercussions of this are that the title needs to be amended – something along the lines of “Diel cycle of marine aerosol concentration over vast areas of the tropical Pacific Ocean and the Caribbean Sea” would be a fair representation of the manuscript – and all references to direct measurements of sea spray aerosol in the manuscript should be edited to marine aerosol (e.g. SSA (N_{SSA}) and $N_{SSA_{0.58\mu m}}$ should be renamed MA (N_{MA}) and $N_{MA_{0.58\mu m}}$).

My third point follows on from the second. The authors have used SEM-EDX to classify the marine aerosol they have sampled onto filters into some of the classes found within it, such as sea salt (the inorganic fraction of sea spray aerosol). While the authors state that “diel cycles were also revealed in the filter counts” on line 122, which I assume refers to the total number of particles on the filters and as such “marine aerosol”, I could not find any mention of whether the authors

observed a diel cycle in the sea salt class specifically. This would be interesting to see because it would give strong support to the author's hypothesis that it was diel cycles in sea spray aerosol that was driving their observations of a diel cycle in marine aerosols. If possible I would like to see this analysis included in the manuscript since it would strengthen the hypothesis.

Specific comments

Line 34 – I would omit "...with spikes at dawn and drops at dusk" since I do not think this accurately represents the results.

Line 34 – I would rephrase to "...and Caribbean Sea and the SSA number concentration more than doubling during the day compared to at night. No correlations between the SSA number concentration and surface winds,..."

Line 45 – Should read "...SSA forms..."

Line 53 – I would omit "Yet, these are not well understood etc." and instead say "However, many of these processes, including whether a diel cycle in SSA production exists have only been explored using artificial bubbling^{10,11}."

Line 69 – Instead of "at daytime" this should read "during daytime".

Line 88 - Presumably these are the averages of the 143 days with clear day night differences. It would be good to include here the averages for the other 125 days as well. Also average can mean a number of different things to different people so the authors should be specific here. Do they mean "mean" or "median" or something else?

Line 120 - Here the referencing format seems different e.g. "(15)".

Line 132 - This would read better as "in the process of marine aerosol emission but is not driving the observed diel cycles."

Line 179 - "In the Atlantic Ocean, though there were diel cycles in γ ..." There is a leap in understanding here. This statement suggests that the majority of the particles measured using this technique are plankton. Is that the case? I would like to see an introductory sentence at the beginning of this section clarifying what type of particles this approach is sensitive to given that the approach will be unfamiliar to many aerosol scientists and this is an interdisciplinary journal.

Line 190 – Should read "We found that the daytime increase in γ is associated..."

Line 236 – I miss discussion relating to the studies by Long et al., (2014) and Keene et al. (2017) here.

Line 277 – There is no mention of any field blanks being conducted. These should be mentioned here.

Line 407 – Incorrect reference form "method described in...44..."

Line 419 – Should read "counts per litre".

Line 503 – This would read better as "In order to determine the cause of the daytime γ changes, we calculated..."

Line 514 – "...first averaged for one hour periods and then smoothed using a five hour moving average."

Figure 1 – While the caption correctly uses "Aerosol concentration per litre" the figure does not and the "SSA" in the figure should be replaced with "Aerosol counts".

Figure 1 caption, line 699 - I would be careful using the term "grow" since it implies processes which you have not quantified (e.g. hygroscopic growth or condensation of matter onto existing particles). Better to say that the particles were consistently larger during the day than during the night.

Figure 3 - Again, since the measurement technique has not differentiated between SSA and any other aerosol I would say simply " $N > 0.58 \mu\text{m}$ " rather than " $N_{\text{SAA}} > 0.58 \mu\text{m}$ ".

Figure 4 caption, line 731 - Here I am confused. In the previous sentence and in the main text you state that there were 143 days where a clear diel cycle was observed. However here you say 94 days. Which is it? Or was there another criteria for these 94 days? If yes, this should be stated here.

Figure 4 caption, line 734 - Average is meaningless. Presumably the authors mean "mean"?

Matt Salter

Reviewer #2 (Remarks to the Author):

Review of "Diel cycle of sea spray aerosol concentration over vast areas of the tropical Pacific Ocean and the Caribbean Sea by Flores et al."

The manuscript by Flores et al. addresses an important and timely problem, namely the lack of knowledge on sea spray aerosol. Sea spray aerosol constitute a major fraction of the global mass of aerosols and influence cloud formation and cloud properties. Many aspects of sea spray aerosol are however not understood, in particular the relation between properties of the surface water (e.g. chemical composition, biological activity, temperature), properties of the marine atmosphere and the source strength and properties of the sea spray aerosol.

The manuscript by Flores et al. presents an interesting and high quality dataset from ship based measurements in Atlantic Ocean, Caribbean Sea, and the Pacific Ocean over a year in 2016-2017. The authors used an optical instrument to measure time resolved size distributions in the size range 250 nm-32 micrometer. They assign these particles to be of marine origin based on back trajectory analysis supported by filter analysis (limited number of days).

They demonstrate a clear difference between particle number concentrations during night and day over most of the cruise. The key result is that the authors demonstrate the presence of a diurnal cycle in the concentration of sea spray aerosols with diameter larger than 0.58 micrometer in several cases in the Pacific Ocean and the Caribbean Sea. The authors also measured parameters related to the state of the atmosphere (air temperature, RH and wind speed, PAR) and the surface water (temperature, salinity, Chl-a, particulate organic carbon, particle size index). The authors come to the conclusion, that the 24-hour cycle in airborne particle number (> 0.58 micrometer) is unrelated to any of the measured parameters except variations in the size of particles (< 1 micrometer) in the surface water.

The authors present an impressive set of field data. The findings of the authors in relation to the diurnal cycle in particle number concentration is very interesting and clearly merits publication in Nature Communications. The authors also discuss an interesting hypothesis that the cycle is linked to particle sizes in the surface water from which the sea spray aerosol derive and Figure 4 provides a nice argument for this connection.

I thus find the findings of the authors novel and of interest to others in the community and in the wider field.

I have some points (detailed below) which should be addressed before publication. In particular: 1) it seems that the authors use somewhat different criteria and baselines for deciding whether the different variables are related to the observed diurnal cycle, 2) the particles are dried to $RH < 40\%$. This is close to the efflorescence point of NaCl - is there a chance that phase state may affect the classification into size bins? 3) looking at gamma versus time it seems to be correlated with sea surface temperature with a clear delay in the day compared to the diurnal cycle of the sea spray aerosol, perhaps this would be worth looking into.

Below some detailed comments and suggestions, which I hope, the authors will find useful.

Detailed comments:

There is a clear criterium defined by the authors for the presence of a diurnal cycle in sea spray number concentration. The same criterium is not applied to the parameters the authors investigate as being correlated to the cycle - what would the result be if the same criterium was applied to the variables investigated (eg. RH, SST etc)?

The authors use different base lines for the different variables - it is not clear to me why that is? I.e. for sea spray aerosol the criterium is: $\mu_{\text{morning}} > \mu_{\text{dawn}} + \sigma_{\text{dawn}}$ and $\mu_{\text{night}} < \mu_{\text{afternoon}} - \sigma_{\text{afternoon}}$, where dawn is 00-00.05, morning is 07.00-11:59, afternoon is 12.00-17.00 and night is 19.00 - 23.59. For wind anomalies it is concluded that these "showed no clear diurnal pattern (see Fig. S6)." - but as far as I can see it is not defined what a diurnal cycle is in this case, and the baseline used is the average of 00.00-00.05 (dawn). For the rate of change of organic carbon, chlorophyll, sea surface temperature and salinity the baseline used is from mid-night to two in the morning.

Line 36: the abstract says "no correlation with surface winds, atmospheric radiation, pollution nor oceanic physical properties were found". It would be informative to give correlation coefficients or perhaps this sentence should be rephrased.

The authors use "optical diameter" throughout the manuscript. Were any corrections for shape applied?

The optical diameters were measured at $RH < 40\%$ - this is close to the efflorescence RH of NaCl. The authors should comment on the phase state of the particles they measure and the importance for the correct assignment into size bins. I miss a discussion about the efflorescence and deliquescence of sea salt and particle phase state and potential implications if both solid and liquid states are present.

Line 68: "we explore how atmospheric and oceanic environmental factors" - I would suggest to write "selected atmospheric and oceanic environmental factors" - and perhaps explain how the factors were selected.

Line 69: "show they are not responsible for the diel cycle;" quite a strong statement - see other comments about the criteria and statistical basis. Can it be entirely excluded that these factors are unrelated to the presence of a cycle?

Line 80: It would be nice if the authors gave this average ratio for the different sections of the route.

Line 104: we also found an inverse relationship between the day-to-night ratio of the NSSA_0.58 μm and the background aerosol concentration (taken as the nighttime concentration; see Fig. S4). - What did this look like in the clean atmosphere? I think the authors should also

show the corresponding plot for a case of clean background.

Line 117: why are there two modes in the chlorine mass distribution – is this an indication of external mixture? (until May 9)?

Line 132: “Further, wind speed anomalies, for the days the diel cycles were detected, showed no clear diurnal pattern (see Fig. S6).” What was the criterion for a diurnal cycle used to make this conclusion?

Line 134-135: “Other atmospheric and oceanic environmental factors are known to affect the production and number of SSA” – which ones do the authors think of here – could be stated with references.

Figure S9 – “No link between the intensity of photosynthetically active radiation and the daytime NSSA_{0.58μm} was found (Fig. S9).” - I agree it does not look like it from the plot. It would however be nice to see the slope of a linear fit to the data and the correlation coefficient. This could be provided as background information, for example in the figure caption.

Line 140: “We also discarded secondary organic aerosol production as the cause of the diel cycle since the produced aerosol by this mechanism is much smaller in diameter (< 0.1 μm).” – what about growth of the aerosols due to condensation – can that be ruled out – in particular in polluted regions?

Line 143: “While there is a diurnal signal in the ABL height, it has less than 30% variations and it cannot account for the variations we see in NSSA_{0.58μm}.” - do I understand correctly that this is based on literature and ABL height was not measured during the cruise? Why is less than 30% variation an argument that it cannot account for the variations seen in NSSA_{0.58μm}? there are other variations discussed which seem to be within 30% (e.g. gamma figure 1C)?

Figure 10 in the reference (19) shows a clear diurnal variation in height of the marine boundary layer – just by looking by eye it seems that it would be very consistent with variation in the time intervals defined by the authors for the sea spray particle cycle. How can a link be entirely excluded? See also Davis et al. 2020.

Figure S9: I suggest that the corresponding figure for days without diurnal variation is shown as a panel b.

It is not clear what the particle index looks like for non-cycle days. The authors could make a figure similar to figure S10 for the particle index showing the diurnal variation in gamma during cycle and non-cycle days for the different parts of the cruise.

In general: N (the number of days of which the average is taken) should be given in all figures.

Line 169: Figure 1C shows that the NSSA_{0.58μm} diel cycle correlates well with the values of γ. – here NSSA₀₅₈ could be plotted versus gamma and the correlation coefficient given to quantify “well”.

By number the majority of sea spray aerosol are smaller than 0.58 micrometer in dry diameter - what are the reflections on that?

Line 449: the authors state that one would expect to see diurnal variations in the smaller particles if RH is an important parameter – but since such variation in the small particles is not observed RH is not important – but elsewhere in the text the authors say that they expect that there is a diurnal variation in the smaller particle sizes but that it is masked by the long lifetime of these particles (Line 228) – how can the effect of RH then be ruled out?

Line 472: “No clear correlation between PAR and NSSA_{0.58µm} was found” – the authors should show a linear least squares fit to the data and give the correlation coefficient.

Figure S8 shows that the strength of the cycle is stronger with stronger windspeed – for the three selected days shown on the photos this seems to be a clear trend? Would that be interesting to look further into?

Figure S11: it is not quite clear – are the averages from days where the cycle is present only or do the averages include data where there is not cycle in the airborne particles? How many days were involved in each of the averages shown?

Were there some days where a cycle in gamma was observed but not in the airborne particles? or was the cycle in gamma always and only occurring when there was a cycle in the airborne particles (larger than 0.58 micrometer)?

Minor.

Line 56: “in high spatial and temporal resolution”: it should be stated what the spatial and temporal resolution were and also the time it took to cover the 42000 km should be given. It would be nice to state also when the expedition took place even if it is given in the references.

Line 60: “AMBL and oceanic (salinity, sea surface temperature, and chlorophyll a) variables. Only oceanic variables are listed, the variables measured for AMBL should also be listed.

Line 103: I do not see that reference 11 shows that the regions near Japan and New Zealand were heavily polluted during the Tara cruise – is it the correct reference?

Figure 1:

I suggest to write the names of the oceans on Figure 1 (they are given in figure 2 but used before that). In line 80 it says that the average ratio is 2.3 over the tropical Pacific Ocean. In figure S1 it is averages over the Pacific ocean. Is it the same time periods, that are referred to in figure S1 and in Figure 2? Line 86 it says “pristine Pacific Ocean” – what is the definition of pristine here- is it the 100 km from the coast?

Line 156: “Furthermore, the presence of the NSSA_{0.58µm} diel cycle in the presence of anthropogenic pollutants and in clean conditions (Fig. 1A,B and Fig. S5), implies anthropogenic and continental sources are also not the cause.” On one of the maps it could be shown where the authors consider conditions clean and where they consider them polluted.

The authors write “litter” as unit in several places - I think it should be “Liter “

Figure S1: It should be stated for how long time (days/hours) was averaged over (the time more than 100 km in the Pacific ocean) This should be stated in the Figure caption.

Figure S2: In the figure caption it says concentration per hour ? on the plots the y-axes just give concentration? It should be stated in the Figure caption what "N" is.

Figure S10: In the caption it says gamma – but gamma is not shown

Figure S10 –What is shown on the y-axis – is it an average of how many days – all days in the Pacific or were some days left out? This should be stated,

In general it should say what the shaded areas in the figures are – is it always one standard deviation of the average?

References

Effect of Clouds on the Diurnal Evolution of the Atmospheric Boundary-Layer Height Over a Tropical Coastal Station, Edwin V. Davis, K. Rajeev & Manoj Kumar Mishra, Boundary-Layer Meteorology volume 175, pages135–152(2020)

Reviewer #3 (Remarks to the Author):

This paper reports a field aerosol measurement over Pacific Ocean and the Caribbean Sea. It reports an interesting and novel finding that the number concentrations of large marine aerosols ($D_p > 580 \text{ nm}$) would show a diel cycle. This paper proposes that the diel cycle might relate to biological activity in seawater. I have some major concerns for this study.

(1) The aerosol number concentrations that this paper reports seem to be very low. A typical marine aerosol concentration is at least several hundred particles/cm³ in pristine environment [Lewis & Schwartz 2004]. The number concentration for particles larger than 0.25 μm would contribute at least ~30% of the total concentration. However, the OPC measurement (Fig.S1) shows that the total aerosol concentration seems to be fewer than 10 particles/cm³. It does not look quite right to me. The OPC is not a good instrument to measurement submicron particles. I think the authors need to provide a comparison of particle size distribution measurement between the OPC and a SMPS.

(2) Another concern is that most of SSA are smaller than 0.58 μm . N_{SSA_0.58 μm} only accounts for a smaller fraction of total SSA number concentration. Thus, this study only investigated a small fraction of SSA, rather than total SSA. The atmospheric implication for this small SSA fraction should be discussed specifically.

(3) The authors argue that the RH, air temperature, and atmospheric stability can be ruled out as the cause of the diel patten. However, their explanation is weak, as the RH, air temperature, and atmospheric stability clearly showed a diel pattern. It is possible that they might have a delayed effect on SSA production. For example, air temperature needs some time to change the temperature of air-sea interface, which is known to affect SSA production.

(4) The authors use Figure S9 to argue that “no link between the intensity of photosynthetically active radiation and the daytime NSSA_{0.58μm} was found”. However, I think Figure 9 can only suggest that there is no significant linear relation between the intensity of PAR and the daytime NSSA_{0.58μm}.

(5) The authors states that “we also discarded secondary organic aerosol production as the cause of the diel cycle since the produced aerosol by this mechanism is much smaller in diameter (< 0.1 μm)”, which is an absolute false statement. Secondary organic aerosol (SOA) matter can form and condense on the existing particles. It is totally possible that smaller SSA were coated by SOA during daytime and their diameters became larger than 0.58μm, thereby causing the diel pattern.

Specific comments:

Line 31: “global scale assessment of this micro-scale process is highly challenging”: this sentence is very vague. Please be specific. Which assessment is challenging?

Line 33: “42,000 km of open ocean waters”: sounds strange. Please revise.

Line 36: “No correlation” should be changed to “no significant correlation

Line 65: “NSSA_{0.58μm}” is quite misleading. I would use something like “NSSA_{>0.58μm}”.

Line 108: I would never use “prove” in any non-mathematical scientific paper.

Line 138: please explain in the main text why the RH, air temperature, and atmospheric stability can be ruled out as the cause of the diel patten. This is very important.

Line 161: what is “seawater particles”?

Line 183: documented in what?

Line 183: “Changes in γ can be attributed to several factors: cell growth, division, and aggregation, selective changes in particle size or concentration due to a balance between primary production and loss due to grazing and viral pressure, or changes in the refractive index of the cell population, which is related to their carbon content.”:

how can the authors contribute the change of aerosol particle size to cell growth, division, and aggregation etc? Or they are actually POC in seawater. It is confusing.

Line 207: Poulain & Bourouiba (literature 34) does not show any results of bubble bursting aerosol flux. The production rate of film drop is an extremely complicated problem. How organic surfactant affect this rate is still largely unknown.

Line 209: Again, SSA flux is an extremely complicated problem. In our laboratory, we studied the effect of biology in seawater on SSA flux but got inconsistent, sometimes contradictory results. I am not convinced that the hypothesis proposed by this paper is well supported by existing evidence.

Line 228: “the lifetime of aerosols is inversely proportional to their size”: this statement is not quite correct. For submicron marine aerosols, the main way to remove them from the air is through wet deposition, which is insensitive to the aerosol sizes. Only for larger aerosol particles, when dry deposition is their main scavenge mechanism, then their lifetime is inversely proportional to their size.

Line 245: how the ship engine exhaust was avoided during the aerosol sampling?

Line 449: “Hence, if RH variations were the cause, we expect to see the diurnal patterns in all sizes, and especially at smaller diameters”: I do not understand the logic here. The particles were dried before measurement, right? The RH could affect bubble evaporation, and it is possible that

the production rate of bubble bursting aerosol is affected.

Line 465: Again, Figure 9 can only suggest that there is no significant linear relation between the intensity of PAR and the daytime NSSA_0.58 μ m.

REVIEWER COMMENTS

We are grateful for the time and effort the reviewers invested in our work, and highly appreciate all of the constructive comments that helped us considerably improve the paper. Thanks to the reviewers comments we were able to generalize our message, make it clearer, and significantly more comprehensive. We will start our reply with few general points and then address all the reviewers' comments point by point. First, we describe here shortly the main modifications done in the manuscript:

- 1) We no longer use a definition to find or define a diel cycle, instead we now calculate a simple ratio. This new analysis clearly shows that **there are more aerosols during the day for all the diameters** measured ($D > 0.25 \mu\text{m}$) and throughout the vast majority of the route (i.e. over the Atlantic Ocean, Caribbean Sea, and the Pacific Ocean). **Suggesting the diel cycle is a widespread phenomenon across the oceans.**
- 2) With further analysis and the simpler definition, we show that air temperature and relative humidity and the atmospheric marine boundary layer height (using the new ERA5 reanalysis data) are most likely not driving the diel cycle. In contrast, we show that the **daily mean sea surface temperature positively correlates with the magnitude of the day-to-nighttime increase in SSA concentration**, and that the SSA concentration is inversely correlated to the rate of change of marine particles sizes (i.e., γ). This further supports the hypothesis that the diel cycle is controlled by processes within the ocean surface.
- 3) Due to these changes, **we added two new figures** and restructured one. One figure shows **the regional and size dependency of the day-to-night ratio (DNR)**. The other figure shows how the DNR varies with total aerosol concentration (for $D > 0.25 \mu\text{m}$) at different wind speed regimes, showing that **long-range transport can mask the diel cycle** and that the wind speed does not control the magnitude of the DNR. The restructured figure has **the total aerosol concentration along Tara's route** and the DNR per diameter, allowing the reader to view the spatial differences of our findings. Moreover, it allows us to show **the interplay between the contribution of long-range continental transport and the local production** to the total marine aerosol concentration, and shows, together with back-trajectory and SEM-EDX analysis, that the main component of the diel cycle comes from sea spray aerosols (SSA).
- 4) Given these changes, we changed **the title of the manuscript to "Diel cycle of sea spray aerosol concentration"**. This highlights the broader nature of our findings. The revised manuscript is now structured as follows: 1) we discuss the main components of the marine aerosol number concentration; 2) we show measurements of consistently more marine aerosols during the day on the vast majority of the route and its dependence on size and region; 3) we show that **SSA causes the marine aerosol diel cycles**; 4) we discuss that atmospheric and oceanic environmental factors are not likely to be the main drivers of the diel cycle, that higher contribution of long-range transport can mask the diel cycle, and that **the magnitude of the daytime increase in aerosol concentration is positive correlated to the SST**; 5) Finally, we show that the identified aerosol diel cycles were accompanied by parallel diel patterns of the size of seawater particles. In addition, we conjecture that microbial processes at the ocean surface that influence bubble-bursting dynamics are the most probable cause of the observed diel cycles in SSA number concentration.

Reviewer #1 (Remarks to the Author):

Flores et al. present a set of aerosol concentration measurements obtained in the marine boundary layer during a multi-leg research cruise across the Atlantic Ocean, Caribbean Sea and Pacific Ocean. The main finding by the authors is that the aerosol concentration measured exhibits a diel cycle over the vast oligotrophic regions of the tropical pacific, with maximum aerosol concentrations during the day and lower concentrations at night. Although this is the first time such an observation has been made in the ambient marine boundary layer it is likely that the processes behind this have been captured previously (e.g. Long et al., 2014 and Keene et al., 2017). I certainly think that this data is interesting and worthy of publication since it builds on these previous studies, moving our understanding from the laboratory to the field. As such, I would be supportive of this paper being published in Nature Communications. However, there are some issues I would like to see addressed first.

Dear Dr. M. Salter, we would like to thank you for the time and effort spent thoroughly reading and commenting our manuscript. We appreciate all the constructive comments that helped us improved the paper. We have changed the introduction and discussion to describe better those previous works. Please see below detailed answers to this point and to all of your other comments in a point-by-point manner (our answers are marked in blue).

Major Comments

1) My first issue with the manuscript is the context (or lack of) that the authors place their work in. In 2014 Long et al., published a paper centered on observations of diel variability in nascent sea spray aerosol number fluxes in biologically productive waters. These authors attribute this diel cycle in sea spray aerosol number production to interactions between bubble plumes and biologically or photochemically derived surfactants. The observations made by Long et al. (2014) received further support from the expanded data provided in Keene et al. (2017). The authors of the manuscript under consideration in this review are clearly aware of these two studies since they are cited in the introduction. However, essentially no comment as to the findings of these studies and the results, which are clearly critically relevant to the manuscript in question, have been made by the authors in either their introduction or their discussion. This must be rectified.

Answer: Thank you for this comment. We have updated the introduction and the discussion to refer to these works in this context.

The following paragraph was added to the revised discussion:

“Biological activity near the ocean surface has been shown to affect SSA abundance and chemical composition (Facchini et al., 2008; Fuentes et al., 2010; Keene et al., 2017; Long et al., 2014; Quinn et al., 2015). In particular, Long et al.(Long et al., 2014) and Keene et al.(Keene et al., 2017) explored differences in SSA production between day and night, generating SSA artificially from seawater on ships at sea. Differently from our study, they reported an increase in the number and mass of SSA during the day in biologically active waters and not in oligotrophic ones. They attributed the increased number concentration of SSA following sunrise to biogenic surfactants emitted either from biota or produced photochemically.

Other studies focused on links between biological activity and SSA production have shown contradicting results. In the laboratory, on the one hand, SSA production increased when diatomaceous exudate (Fuentes et al., 2010) or actively growing bacteria and phytoplankton (Alpert et al., 2015) were present in the seawater; and on the other hand, a decrease in production occurred when surfactants were present (Modini et al., 2013) or when there was an increase in algal biomass (Christiansen et al., 2019). A satellite observation study (Dror et al., 2018), reported a negative correlation of chl-a concentration to coarse-mode aerosol optical depth in the south Pacific Gyre on a seasonal time scale, and suggested that high biological activity suppresses particle production.

We are referring to these references also in the introduction:

“However, global-scale coverage is generally restricted to remote sensing daytime measurements, and many of these processes, including whether a diel cycle in SSA production exists, have only been explored using artificial bubbling (Keene et al., 2017; Long et al., 2014), where diel variability in mass and number fluxes was observed in biologically productive waters.”

- Long, M. S. *et al.* Light-enhanced primary marine aerosol production from biologically productive seawater. *Geophys. Res. Lett.* **41**, 2661–2670 (2014).
- Fuentes, E., Coe, H., Green, D., Leeuw, G. de & McFiggans, G. On the impacts of phytoplankton-derived organic matter on the properties of the primary marine aerosol—Part 1: Source fluxes. *Atmos. Chem. Phys.* **10**, 9295–9317 (2010).
- Facchini, M. C. *et al.* Primary submicron marine aerosol dominated by insoluble organic colloids and aggregates. *Geophys. Res. Lett.* **35**, (2008).
- Quinn, P. K., Collins, D. B., Grassian, V. H., Prather, K. A. & Bates, T. S. Chemistry and related properties of freshly emitted sea spray aerosol. *Chem. Rev.* **115**, 4383–4399 (2015).
- Keene, W. C. *et al.* Factors that modulate properties of primary marine aerosol generated from ambient seawater on ships at sea. *J. Geophys. Res. Atmos.* **122**, 11,961–11,990 (2017).
- Alpert, P. A. *et al.* The influence of marine microbial activities on aerosol production: A laboratory mesocosm study. *J. Geophys. Res. Atmos.* **120**, 8841–8860 (2015).
- Modini, R. L., Russell, L. M., Deane, G. B. & Stokes, M. D. Effect of soluble surfactant on bubble persistence and bubble-produced aerosol particles. *J. Geophys. Res. Atmos.* **118**, 1388–1400 (2013).
- Christiansen, S., Salter, M. E., Gorokhova, E., Nguyen, Q. T. & Bilde, M. Sea spray aerosol formation: laboratory results on the role of air entrainment, water temperature, and phytoplankton biomass. *Environ. Sci. Technol.* **53**, 13107–13116 (2019).
- Dror, T., Lehahn, Y., Altaratz, O. & Koren, I. Temporal-Scale Analysis of Environmental Controls on Sea Spray Aerosol Production Over the South Pacific Gyre. *Geophys. Res. Lett.* (2018). doi:10.1029/2018GL078707

2) My second issue is with the terminology used by the authors. Sea spray aerosol, which the authors use in the title and throughout the manuscript are aerosols emitted directly into the atmosphere following production at the ocean surface either through the mechanism of bubble bursting and the ejection of droplets or when water is ripped directly off of wave crests by the wind. As such, sea spray aerosol has a very clearly defined origin. Marine aerosol, which the authors have chosen not to use in this article, is the background aerosol in the boundary layer over the oceans. It contains numerous particle classes including sea spray aerosol, secondary aerosols following the oxidation of dimethyl sulphide or other aerosol precursors, local anthropogenic aerosols (e.g. ship emissions), aerosols of terrestrial origin which may be either natural (e.g. mineral dust emissions) or anthropogenic (e.g. biomass burning and industrial emissions) etc. The point is that this is a mixture of numerous aerosol sources and most critically this is what the authors are quantifying when they make their measurements in the ambient marine boundary layer. They have determined a diel cycle in marine aerosols over large parts of the Pacific Ocean. The authors have done a commendable job in trying to ascertain which of the particle classes present in marine aerosol might be responsible for their observations and it may well be a diel cycle in sea spray aerosol that is driving their observations. However, they cannot be sure since the data they present is limited to aerosol size distribution measurements in the

marine boundary layer. The repercussions of this are that the title needs to be amended – something along the lines of “Diel cycle of marine aerosol concentration over vast areas of the tropical Pacific Ocean and the Caribbean Sea” would be a fair representation of the manuscript – and all references to direct measurements of sea spray aerosol in the manuscript should be edited to marine aerosol (e.g. SSA (N_{SSA}) and $N_{SSA_{0.58\mu m}}$ should be renamed MA (N_{MA}) and $N_{MA_{0.58\mu m}}$).

3) My third point follows on from the second. The authors have used SEM-EDX to classify the marine aerosol they have sampled onto filters into some of the classes found within it, such as sea salt (the inorganic fraction of sea spray aerosol). While the authors state that “diel cycles were also revealed in the filter counts” on line 122, which I assume refers to the total number of particles on the filters and as such “marine aerosol”, I could not find any mention of whether the authors observed a diel cycle in the sea salt class specifically. This would be interesting to see because it would give strong support to the author’s hypothesis that it was diel cycles in sea spray aerosol that was driving their observations of a diel cycle in marine aerosols. If possible I would like to see this analysis included in the manuscript since it would strengthen the hypothesis.

Answer: As the second and third comments are tightly connected, we will answer them together. Thank you for these comments, they helped us expand and clarify our findings.

We agree with Dr. Salter that we are measuring marine aerosols. We have revised the manuscript in the context of measurements of marine aerosols. Since we believe we have compelling evidence that SSA is the oscillating component of the measured marine aerosol (as explained below), we changed the notation at the beginning of the manuscript and kept the older notation after we show we are measuring SSA.

- First, the revised main text now starts with the following paragraph: “Marine aerosols (MA), defined as the aerosols present in the atmospheric marine boundary layer (AMBL), consist of a mixture of natural and anthropogenic components. MA are predominantly composed of four main sources: sea spray aerosols (SSA), aerosols transported from land (i.e., long-range transport), gas-to-particle conversion (homogeneous and heterogeneous nucleation, and condensation), and particles from ship exhausts. The total concentration of MA depends on the distance from its source in combination with transport and deposition processes. This leads to large spatial variability of MA concentrations, with higher concentrations near continents where the terrestrial contribution is larger, than in open ocean regions (Flores et al., 2020; Mayol et al., 2017). Moreover, the distance from the shore also acts as a size sorter as smaller particles are transported, in general, longer distances and have longer lifetimes than coarse particles (Gillett and Morales, 1979). SSA, the main component of MA globally, are...”

- Flores, J. M. *et al.* Tara Pacific Expedition’s Atmospheric Measurements of Marine Aerosols across the Atlantic and Pacific Oceans: Overview and Preliminary Results. *Bulletin of the American Meteorological Society* **101**, E536–E554 (2020).
- Mayol, E. *et al.* Long-range transport of airborne microbes over the global tropical and subtropical ocean. *Nat. Commun.* **8**, 201 (2017).
- Gillett, D. & Morales, C. Environmental factors affecting dust emission by wind erosion. *Saharan dust* 71–94 (1979).

- The revised second paragraph of the main text now reads:

1) “In this study, we explored and quantified diurnal patterns of the *number concentration of marine aerosols* (N_{MA}) in high spatial and temporal (every minute) resolution along 42,000 km over the

Atlantic Ocean, the Caribbean Sea, and Pacific Ocean, measured aboard the schooner *Tara* during the *Tara* Pacific Expedition...”

2) “Here we report the discovery of *a distinct 24-hour pattern in the number concentration of marine aerosols*, show that SSA number concentration (N_{SSA}) variations are responsible for this pattern, and suggest a possible mechanism.”

-We also added a new section to the revised manuscript where we discuss the main components of the marine aerosol number concentrations:

“Main components of the marine aerosol number concentration

We can define the total concentration of MA as the sum of four components: $N_{MA} = N_{SSA} + N_{LRT} + N_{SOA} + N_{SE}$; where N_{LRT} is the number concentration of long-range continental transport, N_{SOA} is the number concentration from aerosols formed from gas-to-particle conversion mechanisms; and N_{SE} refers to the number concentration from ship emissions.

To distinguish between different aerosol sources, it is common to discuss the different aerosol types and properties according to their sizes, where different regimes of the aerosol size distribution belong to different aerosol types and processes (de Leeuw et al., 2014). For instance, the freshly produced marine secondary organic aerosol (SOA) are small in diameter ($< 0.1 \mu\text{m}$); whereas, the typical size range of SSA is larger ($> 0.1 \mu\text{m}$) (Lewis and Schwartz, 2004). The marine anthropogenic sources, such as engine pollution, yield sporadic large number concentrations of small submicron particles (Flores et al., 2020). And, continental aerosols that are transported over the ocean have a variety of sources and therefore sizes; for example, the typical sizes of biomass burning and pollution aerosols are fine (Lighty et al., 2000), whereas desert dust aerosols are usually coarser ($> \sim 0.6 \mu\text{m}$) (Maring et al., 2000). Given we report N_{MA} measurements for $D_{op} > 0.25 \mu\text{m}$, we can discard significant contributions to the total N_{MA} from N_{SE} and N_{SOA} , and we only have an interplay between the contribution of N_{LRT} and N_{SSA} . To this end, given the surface winds range we encountered ($U_{27} < 16 \text{ m s}^{-1}$), as a first approximation, the attribution between the local contribution of N_{SSA} and N_{LRT} can be scaled to the concentration. SSA is produced by the wind stress over the ocean surface, and its concentration has been shown to be (nonlinearly) proportional to the wind speed (de Leeuw et al., 2011; Lewis and Schwartz, 2004; Liu et al., 2021), with typical concentrations in the range of 10s per cubic centimeter (Liu et al., 2021; Quinn et al., 2017). Whereas the contribution of N_{LRT} is proportional to the yields of the continental sources (dust, pollution, smoke), the distance the aerosol traveled, and with aerosol lifetime in the atmosphere, in general, inversely proportional to the aerosol size (Gillett and Morales, 1979; Maring, 2003; Witek et al., 2011), the quantity of smaller diameters will be greater. For example, being near the Saharan desert and tropical biomass burning areas (tropical Africa and the Amazon), the Atlantic ocean’s atmosphere is much more polluted compare to the Pacific ocean (Ben-Ami et al., 2012; Flores et al., 2020). Thus, low N_{MA} values (i.e., $\sim < 50 \text{ cm}^{-3}$ at $D_{op} > 0.25 \mu\text{m}$) indicates that N_{SSA} is the primary contributor with low contribution from N_{LRT} , and an indicator of the contribution from N_{LRT} will be seen as an increase in the N_{MA} .”

- de Leeuw, G. et al. in *Ocean-Atmosphere Interactions of Gases and Particles* (eds. Liss, P. S. & Johnson, M. T.) 171–246 (Springer Berlin Heidelberg, 2014). doi:10.1007/978-3-642-25643-1_4.
- Lewis, R. & Schwartz, E. *Sea salt aerosol production: mechanisms, methods, measurements and models—a critical review*. **152**, (American Geophysical Union, 2004).
- Flores, J. M. et al. *Tara Pacific Expedition’s Atmospheric Measurements of Marine Aerosols across the Atlantic and Pacific Oceans: Overview and Preliminary Results*. *Bulletin of the American Meteorological Society* **101**, E536–E554 (2020).
- Lighty, J. S., Veranth, J. M. & Sarofim, A. F. Combustion aerosols: factors governing their size and composition and implications to human health. *J Air Waste Manag Assoc* **50**, 1565–618; discussion 1619 (2000).

- Maring, H. *et al.* Aerosol physical and optical properties and their relationship to aerosol composition in the free troposphere at Izaña, Tenerife, Canary Islands, during July 1995. *J. Geophys. Res.* **105**, 14677–14700 (2000).
- de Leeuw, G. *et al.* Production flux of sea spray aerosol. *Rev. Geophys.* **49**, (2011).
- Liu, S. *et al.* Sea spray aerosol concentration modulated by sea surface temperature. *Proc. Natl. Acad. Sci. USA* **118**, (2021).
- Quinn, P. K., Coffman, D. J., Johnson, J. E., Upchurch, L. M. & Bates, T. S. Small fraction of marine cloud condensation nuclei made up of sea spray aerosol. *Nat. Geosci.* **10**, 674–679 (2017).
- Gillett, D. & Morales, C. Environmental factors affecting dust emission by wind erosion. *Saharan dust* 71–94 (1979).
- Maring, H. Mineral dust aerosol size distribution change during atmospheric transport. *J. Geophys. Res.* **108**, 8592 (2003).
- Witek, M. L., Flatau, P. J., Teixeira, J. & Markowicz, K. M. Numerical investigation of sea salt aerosol size bin partitioning in global transport models: implications for mass budget and optical depth. *Aerosol Science and Technology* **45**, 401–414 (2011).
- Ben-Ami, Y., Koren, I., Altaratz, O., Kostinski, A. & Lehahn, Y. Discernible rhythm in the spatio/temporal distributions of transatlantic dust. *Atmos. Chem. Phys.* **12**, 2253–2262 (2012).

- We changed the title of the section “Diel pattern in SSA number concentration” to “***Diel pattern in marine aerosol number concentration***”

And the opening phrase of this section now reads: “We explored **temporal patterns of the N_{MA}** with $D_{op} \geq 0.25 \mu\text{m}$ and found there is a consistently higher number of aerosols during daytime compared to nighttime (Fig. 1A).”.

-We changed the title of the section “Origin of the diel cycles” to “***SSA cause the marine aerosol diel cycles***”

And the opening phrase of this section now reads: “To further explored the interplay between the N_{LRT} and N_{SSA} , we used back trajectory analysis and scanning electron microscopy with energy-disperse X-ray (SEM-EDX) analysis, and we determined **the main contributor to the measured N_{MA} cycles** to be N_{SSA} (Fig. 3).”

As we mentioned above, we believe we have compelling evidence that the vast majority of the marine aerosols we measured are sea spray aerosol. Specifically, for three main reasons (which are explained in the manuscript):

- 1) The back trajectory analysis shown in the map of Fig.1 (now Figure 3).
- 2) The total marine aerosol concentration for $D_{op} > 0.25\mu\text{m}$, which was added to the revised Figure 3, now is Figure 1.
- 3) The SEM-EDX analysis, which we expanded to geometrical diameters $> 0.3\mu\text{m}$

Taking this three points, the revised discussion in the section ‘SSA cause the marine aerosol diel cycles’ now reads:

“The calculated back trajectories (using HYSPLIT(Rolph et al., 2017; Stein et al., 2015)) show that the vast majority of **the air masses spent at least 48 hours over the ocean** (Fig. 3). For the tropical Pacific Ocean, besides the Japan – Taiwan region, a couple of days near south America, and near New Zealand, the air masses were coming from the middle of the ocean, thousands of kilometers from the nearest continent. The long residence of the air masses over the ocean will, on the one hand, minimize the influence of continental sources; for example, **we see total aerosol concentrations, for $D_{op} \geq 0.25 \mu\text{m}$, around 10 cm^{-3}** (Fig. 1B). On the other hand, it will buffer diurnal continental emission processes, therefore, the hour-by-hour concentration changes will be dominated by local processes and thus N_{SSA} .

Next, for the period where clear N_{MA} diel patterns are observed in the western Pacific (Fig. 3A; see the orange-blue shaded region in the map), we obtained an approximate chemical signature of aerosols collected on filters with average geometrical diameters $D_{geo} > 0.3 \mu\text{m}$. We used SEM-EDX analysis and a similar particle classification scheme as described in Laskin et al. (Laskin et al., 2012) (Fig. 3B; a total of 15.5 days were analyzed between 3 – 17 May, 2017, but for clarity only 11 days are shown. See Table S1 for the other days and Methods for the classification scheme). **We found that over 94% of the particles analyzed contained Na**, of which “sea-salt” particles comprised between 45 – 98% of the total particles by number. Up to 1000 km (May 9; orange-shaded area in the map of Fig. 3) away from Keelung, Taiwan, we identified a noticeable depletion of chloride (Fig. S3), together with a lower “sea-salt” fraction (45 – 75%) and an increase in the presence of other metals (e.g., Al, Si, K, Ca, S) and sulfate. The chlorine depletion (seen as a second mode in the chlorine mass distribution with Cl mass % < 0.1; Fig. S3b) together with the increase in sulfates, suggests that anthropogenic pollutants (e.g., H_2SO_4 , SO_2) were present in the AMBL, as SSA are known to react with them (Laskin et al., 2012). After this period, the “sea-salt” fraction comprised 81–99% of the total particles by number. “Sulfates” with no sodium and “Other” species were < 6% for the whole period. **The N_{MA} diel cycles were also revealed in the filter counts; we counted a total of 5701 sea salt particles (out of 6901) in 14 daytime filters and 4447 sea salt particles (out of 5894) in 15 nighttime filters (see Table S1). We have not, however, observed significant day-to-night aerosol-class differences.**

This two-week SEM-EDX analysis where clear diel patterns were seen, confirms the primarily type of aerosol exhibiting diel cycles to be sea salt. While the diameters measured in the filters using the SEM are the dry geometrical diameters, the OPC diameters will depend on the refractive index and shape of the particles. Given that the aerosol was dried to an $\text{RH} < 40\%$, which is below the efflorescent point of NaCl (Gupta et al., 2015), we can consider the measured sea salt by the OPC to be dry sea salt. The geometrical diameters of dry sea salt are in general 4 – 30 % larger than the dry optical diameter (Flores et al., 2009) (e.g., $D_{geo} = 0.3 \mu\text{m}$ is approximately $D_{op} = 0.28 \mu\text{m}$; see Fig. S1), therefore, the vast majority of the particles of all diameters measured by the OPC showing the diel pattern, can be considered to be dry sea salt. For particles with D_{geo} larger than about $0.5 \mu\text{m}$, previous measurements in the AMBL have also shown that the dominant component of SSA in the marine aerosol mass size distribution is inorganic sea salt (Liu et al., 2021; McInnes et al., 1996; Quinn and Coffman, 1998; Quinn et al., 2017), consistent with our findings.

Thus, we can conclude that the diel cycle of N_{MA} is primarily controlled by N_{SSA} and that in areas where long-range transport is significant, the N_{LRT} can mask the day-to-night differences. ”

-In line 122 as the reviewer points out, we mentioned that we also see evidence of greater amount of SSA during daytime. This analysis observed a diel cycle specifically in the sea salt class, we corrected the sentence (we bolded the phrase above, but for the reviewer’s convenience we repeat it here): **“The N_{MA} diel cycles were also revealed in the filter counts; we counted a total of 5701 sea salt particles (out of 6901) in 14 daytime filters and 4447 sea salt particles (out of 5894) in 15 nighttime filters”**.

- Stein, A. F. *et al.* Noaa’s HYSPLIT atmospheric transport and dispersion modeling system. *Bull. Amer. Meteor. Soc.* **96**, 2059–2077 (2015).
- Rolph, G., Stein, A. & Stunder, B. Real-time Environmental Applications and Display sYstem: READY. *Environ. Model. Softw.* **95**, 210–228 (2017).
- Laskin, A. *et al.* Tropospheric chemistry of internally mixed sea salt and organic particles: Surprising reactivity of NaCl with weak organic acids. *J. Geophys. Res.* **117**, (2012).
- Gupta, D. *et al.* Hygroscopic properties of NaCl and NaNO_3 mixture particles as reacted inorganic sea-salt aerosol surrogates. *Atmos. Chem. Phys.* **15**, 3379–3393 (2015).

- Flores, J. M., Trainic, M., Borrmann, S. & Rudich, Y. Effective broadband refractive index retrieval by a white light optical particle counter. *Phys. Chem. Chem. Phys.* **11**, 7943–7950 (2009).
- McInnes, L. M., Quinn, P. K., Covert, D. S. & Anderson, T. L. Gravimetric analysis, ionic composition, and associated water mass of the marine aerosol. *Atmos. Environ.* **30**, 869–884 (1996).
- Quinn, P. K. & Coffman, D. J. Local closure during the First Aerosol Characterization Experiment (ACE 1): Aerosol mass concentration and scattering and backscattering coefficients. *J. Geophys. Res. Atmos.* (1998).
- Quinn, P. K., Coffman, D. J., Johnson, J. E., Upchurch, L. M. & Bates, T. S. Small fraction of marine cloud condensation nuclei made up of sea spray aerosol. *Nat. Geosci.* **10**, 674–679 (2017).
- Liu, S. *et al.* Sea spray aerosol concentration modulated by sea surface temperature. *Proc. Natl. Acad. Sci. USA* **118**, (2021).

Specific comments

4) Line 34 – I would omit “...with spikes at dawn and drops at dusk” since I do not think this accurately represents the results.

Line 34 – I would rephrase to “...and Caribbean Sea and the SSA number concentration more than doubling during the day compared to at night. No correlations between the SSA number concentration and surface winds...”

Answer: Due to the generalization of our findings, we rephrased the abstract. This part of the abstract now reads: “We discovered a ubiquitous 24-hour rhythm to the SSA number concentration, with concentrations increasing after sunrise, remaining higher during the day, and returning to predawn values after sunset. We found continental aerosol transport can mask the SSA cycle. We did not find significant links between the diel cycle of SSA number concentration and diel variations of surface winds, atmospheric physical properties, radiation, pollution, nor oceanic physical properties.”

5) Line 45 – Should read “...SSA forms...”

Answer: Thank you. Corrected

6) Line 53 – I would omit “Yet, these are not well understood etc.” and instead say “However, many of these processes, including whether a diel cycle in SSA production exists have only been explored using artificial bubbling^{10,11}.”

Answer: We rephrased according to the suggestion. The phrase now reads: “***However, global-scale coverage is generally restricted to remote sensing daytime measurements, and many of these processes, including whether a diel cycle in SSA production exists, have only been explored using artificial bubbling(Keene et al., 2017; Long et al., 2014), where diel variability in mass and number fluxes was observed in biologically productive waters.***”

- Long, M. S. *et al.* Light-enhanced primary marine aerosol production from biologically productive seawater. *Geophys. Res. Lett.* **41**, 2661–2670 (2014).
- Keene, W. C. *et al.* Factors that modulate properties of primary marine aerosol generated from ambient seawater on ships at sea. *J. Geophys. Res. Atmos.* **122**, 11,961–11,990 (2017).

7) Line 69 – Instead of “at daytime” this should read “during daytime”.

Answer: Thank you. We deleted this section

8) Line 88 - Presumably these are the averages of the 143 days with clear day night differences. It would be good to include here the averages for the other 125 days as well. Also average can mean a number of

different things to different people so the authors should be specific here. Do they mean “mean” or “median” or something else?

Answer: We do not use a specific definition to fine a diel cycle, therefore, we deleted this section.

9) Line 120 - Here the referencing format seems different e.g. “(15)”.

Answer: Thanks for noticing it, we fixed it.

10) Line 132 - This would read better as “in the process of marine aerosol emission but is not driving the observed diel cycles.”

Answer: Thank you. The text was changed to “in the process of SSA emission, but is not driving the observed diel cycles” Note: we changed to SSA since above we show SSA is driving the diel cycles.

11) Line 179 - “In the Atlantic Ocean, though there were diel cycles in γ ...” There is a leap in understanding here. This statement suggests that the majority of the particles measured using this technique are plankton. Is that the case? I would like to see an introductory sentence at the beginning of this section clarifying what type of particles this approach is sensitive to given that the approach will be unfamiliar to many aerosol scientists and this is an interdisciplinary journal.

Answer: We updated the first paragraph of this section to include a small introduction about this issue, the first paragraph now reads:

“The identified N_{SSA} diel cycles in the lower atmosphere were accompanied by distinct diel cycles of the near-surface ocean light attenuation wavelength dependence of c_p (expressed as particle size index γ ; Fig. 1C). c_p is an inherent optical property that depends, theoretically, on all the particles present in the water column (i.e., autotrophic and heterotrophic micro-organisms, as well as detritus and mineral particles). Given c_p is wavelength-dependent, γ is found via a power-law fit (Boss et al., 2001; Dall’Olmo et al., 2011):

$$c_p = c_p(\lambda_0) \left(\frac{\lambda}{\lambda_0} \right)^{-\gamma} \quad (1)$$

where $c_p(\lambda_0)$ is c_p at a reference wavelength λ_0 and γ is the spectral slope of c_p . Variability of γ indicates changes in the median particle size and is most sensitive to particles in the range of 0.22 – 20 μm (Boss et al., 2001, 2018), with smaller γ associated with larger median particle size (similar to the Ångström exponent for aerosols).”

Additionally, to clarify by how much changes in c_p and therefore gamma, are attributed to phytoplankton, we rephrase the sentence in line 187 to: “In the Equatorial Pacific, phytoplankton cells were reported to contribute from 50 to 83% of the total particulate organic carbon concentration (Grob et al., 2007), and the changes in c_p and γ throughout the day, were principally driven by phytoplankton growth and division (Binder and DuRand, 2002; Durand and Olson, 1996)”

- Boss, E., Twardowski, M. S. & Herring, S. Shape of the particulate beam attenuation spectrum and its inversion to obtain the shape of the particulate size distribution. *Appl Opt* 40, 4885 (2001).
- Dall’Olmo, G. et al. Inferring phytoplankton carbon and eco-physiological rates from diel cycles of spectral particulate beam-attenuation coefficient. *Biogeosciences* 8, 3423–3439 (2011).
- Boss, E., Haëntjens, N., Westberry, T. K., Karp-Boss, L. & Slade, W. H. Validation of the particle size distribution obtained with the laser in-situ scattering and transmission (LISST) meter in flow-through mode. *Opt Express* 26, 11125–11136 (2018).

- Durand, M. D. & Olson, R. J. Contributions of phytoplankton light scattering and cell concentration changes to diel variations in beam attenuation in the equatorial Pacific from flow cytometric measurements of pico-, ultra- and nanoplankton. *Deep Sea Research Part II: Topical Studies in Oceanography* 43, 891–906 (1996).
- Grob, C. et al. Contribution of picoplankton to the total particulate organic carbon concentration in the eastern South Pacific. *Biogeosciences* 4, 837–852 (2007).
- Binder, B. J. & DuRand, M. D. Diel cycles in surface waters of the equatorial Pacific. *Deep Sea Research Part II: Topical Studies in Oceanography* 49, 2601–2617 (2002).

12) Line 190 – Should read “We found that the daytime increase in γ is associated...”

Answer: Thank you. The text was changed.

13) Line 236 – I miss discussion relating to the studies by Long et al., (2014) and Keene et al. (2017) here.

Answer: We have now added a discussion about these studies.

The following paragraph was added to the revised discussion:

“Biological activity near the ocean surface has been shown to affect SSA abundance and chemical composition (Facchini et al., 2008; Fuentes et al., 2010; Keene et al., 2017; Long et al., 2014; Quinn et al., 2015). In particular, Long et al. (Long et al., 2014) and Keene et al. (Keene et al., 2017) explored differences in SSA production between day and night, generating SSA artificially from seawater on ships at sea. Differently from our study, they reported an increase in the number and mass of SSA during the day in biologically active waters and not in oligotrophic ones. They attributed the increased number concentration of SSA following sunrise to biogenic surfactants emitted either from biota or produced photochemically. Other studies focused on links between biological activity and SSA production have shown contradicting results. In the laboratory, on the one hand, SSA production increased when diatomaceous exudate (Fuentes et al., 2010) or actively growing bacteria and phytoplankton (Alpert et al., 2015) were present in the seawater; and on the other hand, a decrease in production occurred when surfactants were present (Modini et al., 2013), or when there was an increase in algal biomass (Christiansen et al., 2019). A satellite observation study (Dror et al., 2018), reported a negative correlation of chl-a concentration to coarse-mode aerosol optical depth in the south Pacific Gyre on a seasonal time scale and suggested that high biological activity suppresses particle production.”

- Long, M. S. *et al.* Light-enhanced primary marine aerosol production from biologically productive seawater. *Geophys. Res. Lett.* **41**, 2661–2670 (2014).
- Fuentes, E., *et al.* On the impacts of phytoplankton-derived organic matter on the properties of the primary marine aerosol—Part 1: Source fluxes. *Atmos. Chem. Phys.* **10**, 9295–9317 (2010).
- Facchini, M. C. *et al.* Primary submicron marine aerosol dominated by insoluble organic colloids and aggregates. *Geophys. Res. Lett.* **35**, (2008).
- Quinn, P. K., Collins, D. B., Grassian, V. H., Prather, K. A. & Bates, T. S. Chemistry and related properties of freshly emitted sea spray aerosol. *Chem. Rev.* **115**, 4383–4399 (2015).
- Keene, W. C. *et al.* Factors that modulate properties of primary marine aerosol generated from ambient seawater on ships at sea. *J. Geophys. Res. Atmos.* **122**, 11,961–11,990 (2017).
- Alpert, P. A. *et al.* The influence of marine microbial activities on aerosol production: A laboratory mesocosm study. *J. Geophys. Res. Atmos.* **120**, 8841–8860 (2015).
- Modini, R. L., Russell, L. M., Deane, G. B. & Stokes, M. D. Effect of soluble surfactant on bubble persistence and bubble-produced aerosol particles. *J. Geophys. Res. Atmos.* **118**, 1388–1400 (2013).
- Christiansen, S., Salter, M. E., Gorokhova, E., Nguyen, Q. T. & Bilde, M. Sea spray aerosol formation: laboratory results on the role of air entrainment, water temperature, and phytoplankton biomass. *Environ. Sci. Technol.* **53**, 13107–13116 (2019).
- Dror, T., Lehahn, Y., Altaratz, O. & Koren, I. Temporal-Scale Analysis of Environmental Controls on Sea Spray Aerosol Production Over the South Pacific Gyre. *Geophys. Res. Lett.* (2018). doi:10.1029/2018GL078707

14) Line 277 – There is no mention of any field blanks being conducted. These should be mentioned here.
Answer: We added at the end of the paragraph the following sentence: “Blank filters were collected every few days by placing filters on the filter holders, closing the system for a few seconds, reopening the holders, and storing them in a PetriSlide dish.”

15) Line 407 – Incorrect reference form “method described in...44...”

Answer: It’s the correct reference. This reference refers to how we calculated the mass percent in each particle. After this calculation, the method described by Laskin et al. (2012) was applied. To clarify this, we added the following the revised section: “..., the method described in (Salge et al., 2013) was used to calculate the mass percent of each detected element. We excluded C from the mass percent calculation since the filters were made of polycarbonate. Following the mass calculation, **the method described by Laskin et al. (2012) was applied. First**, particles containing sodium above 0.01 mass percent ($[Na] > 0$) were first separated from those without sodium.”

- Laskin, A. *et al.* Tropospheric chemistry of internally mixed sea salt and organic particles: Surprising reactivity of NaCl with weak organic acids. *J. Geophys. Res.* **117**, (2012).
- Salge, T., Neumann, R., Andersson, C. & Patzschke, M. Advanced mineral classification using feature analysis and spectrum imaging with EDS. *Proceedings: International Mining Congress and Exhibition, 23rd, Turkey, UCTEA Chamber of Mining Engineers of Turkey* 357 (2013).

16) Line 419 – Should read “counts per litre”.

Answer: We have changed it.

17) Line 503 – This would read better as “In order to determine the cause of the daytime γ changes, we calculated...”

Answer: Thank you. Changed

18) Line 514 – “...first averaged for one hour periods and then smoothed using a five hour moving average.”

Answer: Thank you. Changed

19) Figure 1 – While the caption correctly uses “Aerosol concentration per litre” the figure does not and the “SSA” in the figure should be replaced with “Aerosol counts”.

Answer: We changed the figure, it is now:

20) Figure 1 caption, line 699 - I would be careful using the term “grow” since it implies processes which you have not quantified (e.g. hygroscopic growth or condensation of matter onto existing particles). Better to say that the particles were consistently larger during the day than during the night.

Answer: Sorry for the confusion, here we are referring to marine particles. To make it clearer, we rephrased the sentence to: “(C) 24-hour signal of marine particle size index γ (vertical axis inverted), where the mean particle size increases during the day and decreases during the night.”

21) Figure 3 - Again, since the measurement technique has not differentiated between SSA and any other aerosol I would say simply “ $N > 0.58 \mu\text{m}$ ” rather than “ $N_{\text{SAA}} > 0.58 \mu\text{m}$ ”.

Answer: Based on our answer given for question 2) and 3) above, we changed the notation at the beginning of the manuscript and kept the older notation after we show we are measuring SSA.

22) Figure 4 caption, line 731 - Here I am confused. In the previous sentence and in the main text you state that there were 143 days where a clear diel cycle was observed. However here you say 94 days. Which is it? Or was there another criteria for these 94 days? If yes, this should be stated here.

Answer: Given we generalized our findings, the analysis is no longer restricted to where an aerosol diel cycle was found. Figure 4 (now it is Figure 6) shows the days where data was available, 167 days in total.

23) Figure 4 caption, line 734 - Average is meaningless. Presumably the authors mean “mean”?

Answer: Thank you. Changed.

Matt Salter

Reviewer #2 (Remarks to the Author):

Review of “Diel cycle of sea spray aerosol concentration over vast areas of the tropical Pacific Ocean and the Caribbean Sea by Flores et al.”

The manuscript by Flores et al. addresses an important and timely problem, namely the lack of knowledge on sea spray aerosol. Sea spray aerosol constitutes a major fraction of the global mass of aerosols and influences cloud formation and cloud properties. Many aspects of sea spray aerosol are however not understood, in particular the relation between properties of the surface water (e.g. chemical composition, biological activity, temperature), properties of the marine atmosphere and the source strength and properties of the sea spray aerosol.

The manuscript by Flores et al. presents an interesting and high quality dataset from ship based measurements in Atlantic Ocean, Caribbean Sea, and the Pacific Ocean over a year in 2016-2017. The authors used an optical instrument to measure time resolved size distributions in the size range 250 nm-32 micrometer. They assign these particles to be of marine origin based on back trajectory analysis supported by filter analysis (limited number of days).

They demonstrate a clear difference between particle number concentrations during night and day over most of the cruise. The key result is that the authors demonstrate the presence of a diurnal cycle in the concentration of sea spray aerosols with diameter larger than 0.58 micrometer in several cases in the Pacific Ocean and the Caribbean Sea. The authors also measured parameters related to the state of the atmosphere (air temperature, RH and wind speed, PAR) and the surface water (temperature, salinity, Chl-a, particulate organic carbon, particle size index). The authors come to the conclusion, that the 24-hour cycle in airborne particle number (>0.58 micrometer) is unrelated to any of the measured parameters except variations in the size of particles (<1 micrometer) in the surface water.

The authors present an impressive set of field data. The findings of the authors in relation to the diurnal cycle in particle number concentration is very interesting and clearly merits publication in Nature Communications. The authors also discuss an interesting hypothesis that the cycle is linked to particle sizes in the surface water from which the sea spray aerosol derive and Figure 4 provides a nice argument for this connection.

I thus find the findings of the authors novel and of interest to others in the community and in the wider field.

Answer: We would like to thank reviewer#2 for a thorough and very helpful review of our manuscript. We highly appreciate all the constructive comments that helped us improve the paper. Below we address all the reviewer's comments point by point (our answers are marked in blue).

I have some points (detailed below) which should be addressed before publication. In particular:

1) it seems that the authors use somewhat different criteria and baselines for deciding whether the different variables are related to the observed diurnal cycle,

Answer: Thank you for pointing this out. This allowed us to generalize our manuscript. As mentioned in the introductory page, we no longer use a specific definition to find and decide what is a diel cycle or not,

we decided to use a simple ratio. We calculated the day to nighttime ratios (DNR) by averaging from 07:00 to 17:00 for the day and from 19:00 to 05:00 for the night for all diameters. The measurements between 05:00 to 07:00 and 17:00 to 19:00 were excluded to avoid strong fluctuations of the ratio.

First, this change surfaced that there are greater amount of aerosols during the day for all the diameters measured ($D > 0.25\mu\text{m}$) on the vast majority of the route and that the DNR is size dependent. We added the following graph to the revised manuscript:

Figure 2. Mean day-to-night aerosol count ratio for each bin of the optical particle counter for different regions across *Tara*'s route. The shaded areas represent 1σ , and only the top parts are shown for clarity. 'N' refers to the number of days analyzed.

Second, we use the DNR vs. the difference between the daytime mean and nighttime mean for the atmospheric (air temperature, relative humidity, and wind speed) and oceanic (Salinity, sea surface temperature, and chlorophyll-a) variables to check if there's a relationship between them. We found no correlation with any of the variables. We used the difference instead of the DNR for the atmospheric and oceanic variables since it gives a more physical intuition of the diurnal changes. However, we did the same analysis using the DNR and found no correlations either (not shown here).

The revised supplementary figures for this analysis:

Fig. S4. Ratio of the day-to-nighttime concentration for aerosols with $D \geq 0.58\mu\text{m}$ vs the mean day-to-nighttime difference of air temperature, relative humidity and wind speed. The DNR for the atmospheric variables were calculated using the same daytime (from 07:00 to 17:00 MST) and nighttime (19:00 to 05:00 MST) periods. The red line is the linear fit to each data set. ‘N’ refers to the number of days analyzed. No correlation was found with any of the three atmospheric variables.

Fig. S8. Ratio of the day-to-nighttime concentration for aerosols with $D > 0.58\mu\text{m}$ vs mean day-to-nighttime difference of salinity, sea surface temperature, and chlorophyll a at a depth of around 0.5 – 3.0 m. The red line is the linear fit to all the available data. ‘N’ refers to the number of days analyzed.

2) the particles are dried to $\text{RH} < 40\%$. This is close to the efflorescence point of NaCl - is there a chance that phase state may affect the classification into size bins?

Answer: Thank you for this important comment that helped us clarify this issue in the manuscript.

We checked with the most recent literature for the efflorescent point of NaCl (Gupta et al., Atmos Chem. Phys., 2015), where they showed it's 46.6% for pure NaCl and ~45.9% for a mixture of NaCl- MgCl_2 with a 0.9 fraction of NaCl, which is similar to the one of seawater. We, therefore, do not think that our current measurements were affected by the phase state of the SSA.

We added the following paragraph to the revised manuscript after the SEM-EDX analysis paragraph: *“This two-week SEM-EDX analysis where clear diel patterns were seen, confirms the primarily type of aerosol exhibiting diel cycles to be sea salt. While the diameters measured in the filters using the SEM are the dry geometrical diameters, the OPC diameters will depend on the refractive index and shape of the particles. Given that the aerosol was dried to an RH < 40%, which is below the efflorescent point of NaCl(Gupta et al., 2015), we can consider the measured sea salt by the OPC to be dry sea salt. The geometrical diameters of dry sea salt are in general 4 – 30 % larger than the dry optical diameter(Flores et al., 2009) (e.g., $D_{geo} = 0.3 \mu\text{m}$ is approximately $D_{op} = 0.28 \mu\text{m}$; see Fig. S1), therefore, the vast majority of the particles of all diameters measured by the OPC showing the diel pattern, can be considered to be dry sea salt.”*

- Gupta, D. et al. Hygroscopic properties of NaCl and NaNO₃ mixture particles as reacted inorganic sea-salt aerosol surrogates. Atmospheric Chemistry and Physics 15, 3379–3393 (2015).
- Flores, J. M., Trainic, M., Borrmann, S. & Rudich, Y. Effective broadband refractive index retrieval by a white light optical particle counter. Phys Chem Chem Phys 11, 7943–7950 (2009).

3) looking at gamma versus time it seems to be correlated with sea surface temperature with a clear delay in the day compared to the diurnal cycle of the sea spray aerosol, perhaps this would be worth looking into.

Answer: Thank you for this comment. Based on this suggestion we further looked into the relationship between gamma and SST. First, we looked in detail into the period we show in Fig. 1b,c,d (Fig. Q3_1):

Figure Q3_1. Gamma (black circles) and the sea surface temperature (blue triangles) during the same period as shown in Fig. 1b,c,d of the main text.

And then, we explored the data for the whole period presented in this manuscript (Fig. Q3_2):

Figure Q3_2. Gamma vs sea surface temperature separated by region and time.

From both figures above, we see no clear relationship between gamma and SST. We also explored this relationship by plotting $\gamma(t+z)$ vs. $SST(t)$, where z is the delay in hours, for z up to 6, and found no relationship either.

We did not add this analysis to the revised manuscript since we believe the interpretation is out of the scope of the manuscript.

However, this lead us to look into the relationship between the $SSA_{0.58\mu m}$ day-to-night ration vs the daily mean sea surface temperature. We found a clear relationship and added the following figure to the SI:

Fig. S10. Ratio of the daytime to nighttime concentration for aerosols with $D > 0.58\mu m$ vs the daily mean sea surface temperature at a depth of around 0.5 – 3.0 m. The grey circles show all the data. The black circles show the mean of the data, binned into equally number of points bins ($N=62$ per bin). The red line is the linear fit to the mean data.

This result is consistent with the recent publication from Liu et al. PNAS, 2021, where they showed that SST modulates the concentration of SSA. Additionally, it supports our hypothesis of the marine origin of the diel cycle.

We also added the following text to the discussion: “Similarly, although SST diurnal variations are known (Kawai and Wada, 2007), the rate of temperature change is much slower and can be observed later in the day ((Kawai and Wada, 2007) and Fig. S9) compared to the observed increase in $N_{SSA \geq 0.58\mu m}$. Additionally, we observed the $N_{SSA \geq 0.58\mu m}$ cycle in conditions of wind speeds above 10 m s^{-1} (Fig. 3A and Fig. S5), where SST diurnal changes are expected to be less than 0.25 K (Kawai and Wada, 2007), and in morning overcast conditions (Fig. S5B), when no significant changes in SST are expected. We also did not find a correlation with the DNR of $N_{SSA \geq 0.58\mu m}$ ($R^2 = 0.02$; Fig. S8). **We did find a positive correlation of the DNR of $N_{SSA \geq 0.58\mu m}$ to the daily mean SST ($R^2 = 0.89$; Fig. S10), consistent with the recent report showing that higher SST enhances SSA generation (Liu et al., 2021). This suggests that SST is not likely to drive the $N_{SSA \geq 0.58\mu m}$ diel cycles, but it does play a role in modulating the intensity of the DNR.**”

- Liu, S. *et al.* Sea spray aerosol concentration modulated by sea surface temperature. *Proc. Natl. Acad. Sci. USA* **118**, (2021).

Below some detailed comments and suggestions, which I hope, the authors will find useful.

Detailed comments:

4) There is a clear criterium defined by the authors for the presence of a diurnal cycle in sea spray number concentration. The same criterium is not applied to the parameters the authors investigate as being correlated to the cycle – what would the result be if the same criterium was applied to the variables investigated (eg. RH, SST etc)?

Answer: Please see the answer to comment 1)

5) The authors use different base lines for the different variables – it is not clear to me why that is? I.e. for sea spray aerosol the criterium is: $\mu_{\text{morning}} > \mu_{\text{dawn}} + \sigma_{\text{dawn}}$ and $\mu_{\text{night}} < \mu_{\text{afternoon}} - \sigma_{\text{afternoon}}$, where dawn is 00-00.05, morning is 07.00-11:59, afternoon is 12.00-17.00 and night is 19.00 – 23.59. For wind anomalies it is concluded that these “showed no clear diurnal pattern (see Fig. S6).” – but as far as I can see it is not defined what a diurnal cycle is in this case, and the baseline used is the average of 00.00-00.05 (dawn). For the rate of change of organic carbon, chlorophyll, sea surface temperature and salinity the baseline used is from mid-night to two in the morning.

Answer: Please see the answer to comment 1).

And about the different baselines, the reviewer is correct. Thank you for pointing this out, we have corrected it. In the revised manuscript we only calculate anomalies for the SST and the rate of change of particulate organic carbon, and all the baselines are from midnight to five in the morning. Due to this Fig. S9 and Fig. S14 are now updated to:

Fig. S9. Diurnal anomaly of sea surface temperature anomaly per day at a depth of around 0.5 – 3.0 m. The baseline was taken to be the mean value between midnight and five in the morning. The shaded areas are 1σ . ‘N’ refers to the number of days analyzed.

Figure S14. Diurnal anomaly of the rate of change of particulate organic carbon at a depth of around 0.5 – 3.0 m during the Keelung to Fiji leg (May 2017). The baseline was taken to be the *mean* value between midnight and five in the morning. The shaded areas are 1σ and ‘N’ refers to the number of days analyzed.

6) Line 36: the abstract says “no correlation with surface winds, atmospheric radiation, pollution nor oceanic physical properties were found”. It would be informative to give correlation coefficients or perhaps this sentence should be rephrased.

Answer: Given the new analysis we rephrase to: “We did not find significant links between the diel cycle of SSA number concentration and diel variations of surface winds, atmospheric physical properties, radiation, pollution, nor oceanic physical properties.

7) The authors use “optical diameter” throughout the manuscript. Were any corrections for shape applied?

Answer: Thank you for pointing out this important aspect. We chose to use optical diameter because our main measurements were done using an OPC, and wanted to let the reader know that we are not referring to the geometrical size of the SSA measured. We decided not to use the geometrical diameters in the main text since the key result of the manuscript is the presence of a diurnal cycle in the concentration of SSA, and the effect of non-sphericity, inhomogeneities, and the refractive index of the particles is secondary to the main message of the manuscript.

Based on this comment, together with the reviewer’s comment about the phase state of the particles, we added a paragraph after the section where we addressed the chemical signature of the aerosols we measured.

The paragraph reads: “This two-week SEM-EDX analysis where clear diel patterns were seen, confirms the primarily type of aerosol exhibiting diel cycles to be sea salt. *While the diameters measured in the filters using the SEM are the dry geometrical diameters, the OPC diameters will depend on the refractive index and shape of the particles. Given that the aerosol was dried to an RH < 40%, which is below the efflorescent point of NaCl(Gupta et al., 2015), we can consider the measured sea salt by the OPC to be dry sea salt. The geometrical diameters of dry sea salt are in general 4 – 30 % larger than the dry optical diameter(Flores et al., 2009) (e.g., $D_{geo} = 0.3 \mu\text{m}$ is approximately $D_{op} = 0.28 \mu\text{m}$; see Fig.*

S1), therefore, the vast majority of the particles of all diameters measured by the OPC showing the diel pattern, can be considered to be dry sea salt.”

- Gupta, D. et al. Hygroscopic properties of NaCl and NaNO₃ mixture particles as reacted inorganic sea-salt aerosol surrogates. *Atmospheric Chemistry and Physics* 15, 3379–3393 (2015).
- Flores, J. M., Trainic, M., Borrmann, S. & Rudich, Y. Effective broadband refractive index retrieval by a white light optical particle counter. *Phys Chem Chem Phys* 11, 7943–7950 (2009).

Additionally, to treat this issue in more details in the revised manuscript, we added the approximated geometrical diameters to Fig. S1. To do this, we revisited a publication of ours where, in laboratory experiments, we selected sizes of dry NaCl particles with a Differential Mobility Analyzer (DMA) and measured the optical diameter (D_{op}) with a white light OPC (Flores et al. PCCP, 2009; see Figure below).

Fig. 4a from Flores et al. 2009. Optical median diameter vs. mobility diameter for (a) non-absorbing substances: ammonium sulfate, glutaric acid, and sodium chloride.

Since OPC’s are calibrated with polystyrene latex sphere, which do not absorb visible light, the optical diameters measured in this manuscript, at a wavelength of 630nm, will have a similar response to the white light OPC used in the Flores et al. 2009 study. We did a linear fit to the data shown in the Fig. 4a from Flores et al. 2009 for dry NaCl, and use it to estimate the D_{geo} .

The expression we used is:

$$D_{Geo} = \frac{D_{op} - 57.35}{0.75}$$

And the updated Figure and legend are:

Fig. S1.

Diurnal box plot analysis of total concentration of aerosol diameters per hour for 16 bins of the OPC. The box plot analysis shows the median, and the 5th, 25th, 75th, and 95th percentiles. Only data from the Pacific Ocean and when Tara was at least 100 km away from land were used for this analysis, **in total 124 days (~2978 hours) were analyzed.** At the top of each panel the lower and upper limit in micrometers of the OPC bins are shown. We also show the approximate geometrical diameter (D_{Geo}) using a linear fit to the data shown in the Fig. 4a from Flores et al. 2009¹ for dry NaCl: $D_{Geo} = \frac{(D_{OP} - 57.35)}{0.75}$. Note that the linear fit is using data only between 0.4 – 0.8 μm .

And finally, we added the following phrase to the concluding paragraph: “We presented *in-situ* evidence that the number concentration of SSA with $D_{op} \geq 0.25 \mu\text{m}$ (**at $RH < 40\%$; for dry NaCl particles it is equivalent to $D_{Geo} \geq 0.26 \mu\text{m}$, see Fig. S1**) have a distinct 24-hour cycle (clearly visible for $N_{SSA \geq 0.58 \mu\text{m}}$), with higher number concentrations during the day than at night over vast areas of the Pacific Ocean, the Caribbean Sea and the Atlantic Ocean.”

8) The optical diameters were measured at $RH < 40\%$ - this is close to the efflorescence RH of NaCl. The authors should comment on the phase state of the particles they measure and the importance for the correct assignment into size bins. I miss a discussion about the efflorescence and deliquescence of sea salt and particle phase state and potential implications if both solid and liquid states are present.

Answer: Please see the answers to comment no. 2 and 7 above. This issue is regarded in the revised paper: “While the diameters measured in the filters using the SEM are the dry geometrical diameters, the OPC diameters will depend on the refractive index and shape of the particles. Given that the aerosol was dried to an $RH < 40\%$, which is below the efflorescent point of NaCl (Gupta et al., 2015), we can consider the measured sea salt by the OPC to be dry sea salt. The geometrical diameters of dry sea salt are in general 4 – 30 % larger than the dry optical diameter (Flores et al., 2009) (e.g., $D_{geo} = 0.3 \mu\text{m}$ is

approximately $D_{op} = 0.28 \mu\text{m}$; see Fig. S1), therefore, the vast majority of the particles of all diameters measured by the OPC showing the diel pattern, can be considered to be dry sea salt..”

9) Line 68: “we explore how atmospheric and oceanic environmental factors” – I would suggest to write “selected atmospheric and oceanic environmental factors” – and perhaps explain how the factors were selected.

Answer: We deleted this phrase, it’s not included in the revised manuscript.

10) Line 69: “show they are not responsible for the diel cycle;” quite a strong statement – see other comments about the criteria and statistical basis. Can it be entirely excluded that these factors are unrelated to the presence of a cycle?

Answer: We also deleted this phrase. In the updated manuscript the last paragraph of the introduction now reads: “Here we report the discovery of a distinct 24-hour pattern in the number concentration of marine aerosols, show that SSA number concentration (N_{SSA}) variations are responsible for this pattern, and suggest a possible mechanism.”

11) Line 80: It would be nice if the authors gave this average ratio for the different sections of the route.

Answer: Thank you for this comment. We added a new Figure where we show the ratios for all the diameters measured for different regions. The new Fig. 2 is now:

Figure 2. Mean day-to-night aerosol count ratio for each bin of the optical particle counter for different regions across Tara’s route. The shaded areas represent 1σ , and only the top parts are shown for clarity. ‘N’ refers to the number of days analyzed.

12) Line 104: we also found an inverse relationship between the day-to-night ratio of the $N_{SSA \geq 0.58 \mu m}$ and the background aerosol concentration (taken as the nighttime concentration; see Fig. S4). – What did this look like in the clean atmosphere? I think the authors should also show the corresponding plot for a case of clean background.

Answer: Thank you for this suggestion. We expanded this analysis to explore the relationship between the DNR of $N_{SSA \geq 0.58 \mu m}$, and the total aerosol concentration (taken as the concentration for $D_{op} \geq 0.25 \mu m$ during daytime) at different wind speed regimes, and we added it as a new Figure to the revised manuscript. The new Figure 5 is:

Figure 5. Dependence of the day-to-night ratio of $N_{SSA \geq 0.58 \mu m}$ on total daytime aerosol count for three different wind speed regimes. For daily mean wind speeds below 4 m s⁻¹ (black circles), between 4 and 8 m s⁻¹ (orange triangles), and above 8 m s⁻¹ (blue squares) for (A) all the data, and (B) a clean atmosphere (i.e., only in the Pacific Ocean at least 100km away from the continents). ‘N’ refers to the number of days analyzed. While there is no clear relationship between the DNR of $N_{SSA \geq 0.58 \mu m}$ and the wind speed, the DNR tending to one as the total aerosol count increases shows that long-range continental transport of aerosols can mask the diel cycle.

13) Line 117: why are there two modes in the chlorine mass distribution – is this an indication of external mixture? (until May 9)?

Answer: The two modes indicate the presence of pure NaCl (Cl mass % around 0.55) and particles with a chlorine present, but depleted. This happens at the same time we see a greater amount of sulfates. The chlorine depletion and the increase in sulfates suggest that anthropogenic pollutants (e.g., H₂SO₄, SO₂) were present in the AML. Laskin et al. 2012 showed that SSA can react with these pollutants.

To clarify this point, we rephrased the sentence to read: “The chlorine depletion (*seen as a second mode in the chlorine mass distribution with Cl mass % < 0.1; Fig. S5b*) together with the increase in sulfates, suggests that anthropogenic pollutants (e.g., H₂SO₄, SO₂) were present in the AML, as SSA are known to react with them¹⁸.”

14) Line 132: “Further, wind speed anomalies, for the days the diel cycles were detected, showed no clear diurnal pattern (see Fig. S6).” What was the criterion for a diurnal cycle used to make this conclusion?

Answer: We no longer use wind speed anomalies. Please see the answer to comment 1).

15) Line 134-135” Other atmospheric and oceanic environmental factors are known to affect the production and number of SSA” – which ones do the authors think of here – could be stated with references.

Answer: We have added this information to the revised text: : “Other atmospheric environmental factors (*e.g., RH, air temperature, and atmospheric stability*¹) are known to affect the production and number of SSA, but we found no evidence that they are the main drivers of the observed diel cycle.”

16) Figure S9 – “No link between the intensity of photosynthetically active radiation and the daytime NSSA_{0.58μm} was found (Fig. S9).” - I agree it does not look like it from the plot. It would however be nice to see the slope of a linear fit to the data and the correlation coefficient. This could be provided as background information, for example in the figure caption.

Answer: Thank you for this comment. We added the linear fit to the graph (which is now Fig. S7) and wrote the correlation coefficient in the legend.

Fig. S7. Ratio of the daytime to nighttime concentration for aerosols with $D > 0.58\mu\text{m}$ vs the average photosynthetically active radiation. The red line is the linear fit to all the available data. Pr refers to the Pearson correlation coefficient. ‘N’ refers to the number of days analyzed.

17) Line 140: “We also discarded secondary organic aerosol production as the cause of the diel cycle since the produced aerosol by this mechanism is much smaller in diameter ($< 0.1 \mu\text{m}$). “ – what about growth of the aerosols due to condensation –can that be ruled out – in particular in polluted regions?”

Answer: We think that aerosol growth by condensation can also be ruled out due to the known slow growth rates ($< 10\text{nm h}^{-1}$) (Paasonen et al. 2018, Burkart et al. 2017). We explored the full size distribution of the OPC for the period we show in Figure 1 (see Fig. Q17 below), and we could not see any evidence of particle growth (i.e., a steady increase in concentration and the maximum concentration; a banana plot). There is the possibility that the coarse resolution from the OPC will not detect it. However, based on studies in other regions (we could not find a study of condensational growth rate in the Pacific Ocean), it seem highly unlikely that condensational growth is responsible for the diel pattern we observe in SSA. For example, in boreal forest Paasonen et al. 2018 found growth rates of $\sim 3.5\text{nm h}^{-1}$ for initial sizes between 190 - 200nm. In the marine arctic, the growth rates for a preexisting mode of $\sim 90\text{nm}$ were less than 6nm h^{-1} (Burkart et al. 2017). Additionally, condensational growth cannot explain the decrease in concentration at sunset.

Based on the reviewers comment, we have rephrased the sentence, it now reads: “While we cannot discard condensational growth after sunrise, the growth rates are too slow ($< 6 \text{nm h}^{-1}$)(Burkart et al., 2017; Paasonen et al., 2018) to account for the increase in N_{MA} within 30 min after sunrise (Fig. 3A). Furthermore, condensational growth cannot explain the decrease in concentration at sunset.”

Figure Q17. Full size distribution of the OPC vs. time during the same period shown in Fig. 1.

- Paasonen, P., Peltola, M., Kontkanen, J., Junninen, H., Kerminen, V.-M., and Kulmala, M.: Comprehensive analysis of particle growth rates from nucleation mode to cloud condensation nuclei in boreal forest, *Atmos. Chem. Phys.*, 18, 12085–12103, <https://doi.org/10.5194/acp-18-12085-2018>, 2018.
- Burkart, J., Hodshire, A. L., Mungall, E. L., Pierce, J. R., Collins, D. B., Ladino, L. A., and Abbatt, J.: Organic condensation and particle growth to CCN sizes in the summertime marine Arctic is driven by materials more semivolatile than at continental sites, *Geophys. Res. Lett.*, 44, 10725–10734, <https://doi.org/10.1002/2017GL075671>, 2017.

18) Line 143: “While there is a diurnal signal in the AMLB height, it has less than 30% variations and it cannot account for the variations we see in $N_{SSA \geq 0.58 \mu m}$. “ - do I understand correctly that this is based on literature and AMLB height was not measured during the cruise? Why is less than 30% variation an argument that it cannot account for the variations seen in $N_{SSA \geq 0.58 \mu m}$? there are other variations discussed which seem to be within 30% (e.g. gamma figure 1C)?

Answer: Yes, the reviewer is correct our argument was based on literature. Based on this comment, we calculated the AMLB height using the new ERA5 reanalysis data. We added the following section to the methods:

“Atmospheric marine boundary layer height

The atmospheric marine boundary layer height (AMBLH), in meters, was derived from ERA5 reanalysis data (Hersbach et al., 2020). The ERA5 reanalysis data is provided by the European Centre for Medium-Range Weather Forecasts (ECMWF) with 30-km horizontal resolution and 137 vertical hybrid levels with the high vertical resolution (10–200 m intervals) in the lowermost 3 km above surface. The ERA5 AMBLH data has a 0.25° spatial and a one-hour temporal resolution. We used the latitude and longitude along *Tara's* route to obtain the closest AMBLH point from ERA5. Figure S6 shows the AMBLH along the orange and blue-shaded transect in the western Pacific between Keelung and Fiji (next to the double ended arrow) shown in Fig. 3. We also did not find a correlation between the AMBLH and the measured $N_{SSA \geq 0.58 \mu m}$.”

And the following Figure to the SI:

Fig. S6. Aerosol concentration per litre ($D_{\text{dry}} > 0.58 \mu\text{m}$) and the atmospheric marine boundary layer (AMBL) height from ERA5 superimposed on the photo-synthetically active radiation (PAR). The data shown here is from the orange and blue-shaded transect in the western Pacific between Keelung and Fiji (next to the double ended arrow) shown in Fig. 3 of the main text.

Based on this analysis we added the following text to the revised manuscript:

“Changes in AMBL height do not fully explain the diel cycles in N_{SSA} either. If height variations within the AMBL were driving the diel pattern in $N_{\text{SSA} \geq 0.58 \mu\text{m}}$, given there are no changes in the production between day and night, the diurnal concentration changes should follow those seen in the AMBL. For the period with clear diel cycles in $N_{\text{SSA} \geq 0.58 \mu\text{m}}$ shown in Fig. 3A, we did not find AMBL height (using ERA5 reanalysis data (Hersbach et al., 2020); see Methods) diurnal changes that follow those we measured in $N_{\text{SSA} \geq 0.58 \mu\text{m}}$, and the changes we found of the AMBL height were less than 40m for a mean height of 3735m (i.e., ~1%; Fig. S6). We cannot exclude a possible influence, but it is unlikely that atmospheric stability or changes in the AMBL height are driving the diel cycle.”

- Hersbach, H. et al. The ERA5 global reanalysis. Q.J Royal Met. Soc. (2020). doi:10.1002/qj.3803

19) Figure 10 in the reference (19) shows a clear diurnal variation in height of the marine boundary layer – just by looking by eye it seems that it would be very consistent with variation in the time intervals defined by the authors for the sea spray particle cycle. How can a link be entirely excluded? See also Davis et al. 2020.

Answer: Please see the answer to the previous comment.

20) Figure S9: I suggest that the corresponding figure for days without diurnal variation is shown as a panel b.

Answer: We no longer use a definition to find a diel cycle (see answer to comment 1 above), therefore we only added the linear fit the reviewer requested on comment 16)

21) It is not clear what the particle index looks like for non-cycle days. The authors could make a figure similar to figure S10 for the particle index showing the diurnal variation in gamma during cycle and non-cycle days for the different parts of the cruise.

Answer: Thank you for this comment. As mentioned in comment 1) we do not separate the data by days where there was a cycle and when there was not. We, therefore, do not use Fig. 10 or similar analysis in the revised manuscript.

22) In general: N (the number of days of which the average is taken) should be given in all figures.

Answer: We added the N to all relevant figures.

23) Line 169: Figure 1C shows that the $N_{SSA \geq 0.58 \mu m}$ diel cycle correlates well with the values of γ . – here $N_{SSA \geq 0.58 \mu m}$ could be plotted versus gamma and the correlation coefficient given to quantify “well”.

Answer: We are not expecting to have a “good” correlation between specific values of gamma and the total concentration for aerosols with $D \geq 0.58 \mu m$, as on the one hand, gamma can change latitudinally (see Fig. S11B, Fig. S12B in the revised SI), and on the other hand, we do not argue that specific values of gamma will drive a specific change in $N_{SSA \geq 0.58 \mu m}$.

We do, however, expect a greater correlation with the rate of change of gamma, which tells us if there is an increase of the mean particle size ($\partial\gamma/\partial t < 0$) within the ocean surface during the day (which has been shown to be driven by photosynthetic growth and secretion of extracellular polymeric substances), or a decrease in mean particle size ($\partial\gamma/\partial t > 0$) (suggesting cell division). Below we show $N_{SSA \geq 0.58 \mu m}$ vs. $\partial\gamma/\partial t$ for the Pacific Ocean data. As expected, we found a negative correlation, which further supports our results. We think that Figure 4 within the manuscript shows quite well this relationship, and the analysis we’ve done here adds support. Therefore, we decided to add this figure to the SI; it’s now Figure S11.

We, rephrased the sentence and added another phrase to the paragraph: “Figure 3C shows *that there is a congruency* between the $N_{SSA \geq 0.58 \mu m}$ diel cycle **and the daily changes of γ** . Note that γ reached a maximum value (i.e., minimum mean diameter) right after sunrise when the $N_{SSA \geq 0.58 \mu m}$ began to increase, and a minimum value (i.e., maximum mean diameter) before sunset when $N_{SSA \geq 0.58 \mu m}$ decreased (Fig. 3C). We, therefore, calculated the rate of change of γ ($\partial\gamma/\partial t$ (hr^{-1}); see Methods) for concurrent days (167 days in total) where the $N_{SSA \geq 0.58 \mu m}$ was also measured (Fig. 6A, B). We found a parallel behavior, with a continuous decrease in the mean particle diameter at nighttime ($\partial\gamma/\partial t$ (hr^{-1}) > 0) when $N_{SSA \geq 0.58 \mu m}$ were lowest, and a continuous increase ($\partial\gamma/\partial t$ (hr^{-1}) < 0) between 07:00 and 17:00, when $N_{SSA \geq 0.58 \mu m}$ were highest. ***We also found an inverse relationship between $N_{SSA \geq 0.58 \mu m}$ and $\partial\gamma/\partial t$ (hr^{-1}) (Fig. S11; $R^2 = 0.51$), with higher $N_{SSA \geq 0.58 \mu m}$ when $\partial\gamma/\partial t$ (hr^{-1}) is negative.***”

We also added in the Methods, at the end of the section “Rate of change of γ ($\partial\gamma/\partial t$): “Finally, Fig. 6B shows a box plot analysis for 167 days across Tara’s route and there was at least 23 hours of the AC-S data, **and Fig. S9 shows a scatter plot of $N_{SSA \geq 0.58 \mu m}$ vs. $\partial\gamma/\partial t$ (hr^{-1}) for the Pacific Ocean data.**”

Fig. S11. $N_{\text{SSA} \geq 0.58 \mu\text{m}}(\text{h}^{-1})$ vs. $\partial\gamma/\partial t$ (h^{-1}) for the Pacific Ocean data. The grey circles show all the data. The red circles show the mean of the data, binned into equally number of points bins ($N=223$ per bin). The red line is the linear fit to the mean data. Pr refers to the Pearson correlation coefficient.

24) By number the majority of sea spray aerosol are smaller than 0.58 micrometer in dry diameter - what are the reflections on that?

Answer: Thank you for this comment. We added the following paragraph about this issue to the revised conclusions:

“Although the SSA number fraction (for $D > \sim 0.5 \mu\text{m}$) in the AML, by number, is less than 20% (Flores et al., 2020; Quinn et al., 2017), small concentration changes of big aerosols can drive relatively large changes in cloud sizes, lifetime, and rain yields (Dagan et al., 2015, 2017), especially over pristine areas where clouds are aerosol-limited (Koren et al., 2014).”

- Quinn, P. K., Coffman, D. J., Johnson, J. E., Upchurch, L. M. & Bates, T. S. Small fraction of marine cloud condensation nuclei made up of sea spray aerosol. *Nat. Geosci.* 10, 674–679 (2017).
- Flores, J. M. et al. Tara Pacific Expedition’s Atmospheric Measurements of Marine Aerosols across the Atlantic and Pacific Oceans: Overview and Preliminary Results. *Bulletin of the American Meteorological Society* 101, E536–E554 (2020).
- Dagan, G., Koren, I. & Altaratz, O. Aerosol effects on the timing of warm rain processes. *Geophys. Res. Lett.* 42, 4590–4598 (2015).
- Dagan, G., Koren, I., Altaratz, O. & Heiblum, R. H. Time-dependent, non-monotonic response of warm convective cloud fields to changes in aerosol loading. *Atmos. Chem. Phys.* 17, 7435–7444 (2017).
- Koren, I., Dagan, G. & Altaratz, O. From aerosol-limited to invigoration of warm convective clouds. *Science* 344, 1143–1146 (2014).

25) Line 449: the authors state that one would expect to see diurnal variations in the smaller particles if RH is an important parameter – but since such variation in the small particles is not observed RH is not important – but elsewhere in the text the authors say that the expect that there is a diurnal variation in the smaller

particle sizes but that it is masked by the long lifetime of these particles (Line 228) – how can the effect of RH then be ruled out?

Answer: We have rephrased this part of the discussion. We no longer use a specific definition for the diel cycle and using the day-to-night ratio we simplified our argument. If RH is the main drivers of the diel cycle, we expect to have a clear relationship between the DNR of $N_{SSA \geq 0.58 \mu m}$ and the mean day-to-night differences of RH, but we do not. We added this new figure to the SI:

Fig. S4. Ratio of the day-to-nighttime concentration for aerosols with $D \geq 0.58 \mu m$ vs the mean day-to-nighttime difference of air temperature, relative humidity and wind speed. The DNR for the atmospheric variables were calculated using the same daytime (from 07:00 to 17:00 MST) and nighttime (19:00 to 05:00 MST) periods. The red line is the linear fit to each data set. ‘N’ refers to the number of days analyzed. No correlation was found with any of the three atmospheric variables.

And in the revised text we rephrased to:

“Second, links between the diel changes of RH and air temperature to the N_{SSA} diel cycle were explored. If RH or the air temperature were the main drivers of the diel cycle, by affecting bubble and droplet evaporation, or SSA sizes, we expect to have a clear relationship between the DNR of $N_{SSA \geq 0.58 \mu m}$ and the mean day-to-night differences of RH and air temperature, but we found no correlation between them (Fig. S4). Therefore, while RH and air temperature can affect bubble evaporation, and in consequence, the production rate of SSA, they are most likely not driving the diel cycles in $N_{SSA \geq 0.58 \mu m}$.”

26) Line 472: “No clear correlation between PAR and $N_{SSA_0.58 \mu m}$ was found” – the authors should show a linear least squares fit to the data and give the correlation coefficient.

Answer: We updated the figure to include this requested information:

Fig. S7. Ratio of the daytime to nighttime concentration for aerosols with $D > 0.58\mu\text{m}$ vs the average photosynthetically active radiation. The red line is the linear fit to all the available data. Pr refers to the Pearson correlation coefficient. ‘N’ refers to the number of days analyzed.

And we amended the text in the methods, the revised text now reads:

“By definition, solar radiation drives diurnal cycles. Therefore, we explored links between the intensity of solar radiation, measured by the average daytime photosynthetically available radiation (PAR), *and the DNR of $N_{SSA \geq 0.58\mu\text{m}}$* to determine if the intensity of solar radiation has a measurable effect on the total amount of $SSA_{0.58\mu\text{m}}$. *Figure S7 shows the scatter plot of the DNR of $N_{SSA \geq 0.58\mu\text{m}}$ vs. the average PAR.* No *significant linear relation* between PAR and $N_{SSA \geq 0.58\mu\text{m}}$ was found. Furthermore, examples of days with similar PAR that showed different daytime $N_{SSA \geq 0.58\mu\text{m}}$ can be seen in Fig. 3B and Fig. S5.”

27) Figure S8 shows that the strength of the cycle is stronger with stronger windspeed – for the three selected days shown on the photos this seems to be a clear trend? Would that be interesting to look further into?

Answer: For checking this idea, we explored the day-to-night concentration ratio (a measure for the strength of the cycle) vs. average wind speed per day (see Figure Q27 below). We did not find a clear relationship between them. While we did find a clear relationship between the daytime $N_{SSA \geq 0.58\mu\text{m}}$ and wind (Fig. 4 in the revised manuscript), it does not translate to the day-to-night ratio. Also, as shown in comment 12) above, we explored the relationship between the DNR of $N_{SSA \geq 0.58\mu\text{m}}$, and the total aerosol concentration (taken as the concentration for $D_{op} \geq 0.25 \mu\text{m}$ during daytime) at different wind speed regimes, and we added it as a new Figure to the revised manuscript (see Fig.5 below). Which further shows there is no correlation between the DNR and the wind speed. On the other hand, the strength of the cycle seems to be, in part, modulated by the SST (see answer to comment 3) above, and the new Fig. S10 of the revised manuscript added below for the reviewers convenience).

Figure Q27. Day-to-nighttime ratio of $N_{SSA \geq 0.58 \mu m}$ vs. the average daytime wind speed. The blue circles show all the data and the different colored markers the different regions we sampled. We also did the analysis taking the nighttime wind speed and found similar results. This analysis shows that the relationship between $N_{SSA \geq 0.58 \mu m}$ and wind speed, does not translate to the day-to-nighttime ratio and should be the subject of future studies.

Figure 5. Dependence of the day-to-nighttime ratio of $N_{SSA \geq 0.58 \mu m}$ on total daytime aerosol count for three different wind speed regimes. For daily mean wind speeds below 4 ms^{-1} (black circles), between 4 and 8 ms^{-1} (orange triangles), and above 8 ms^{-1} (blue squares) for (A) all the data, and (B) a clean atmosphere (i.e., only in the Pacific Ocean at least 100 km away from the continents). ‘N’ refers to the number of days analyzed. While there is no clear relationship between the DNR of $N_{SSA \geq 0.58 \mu m}$ and the wind speed, the DNR tending to one as the total aerosol count increases shows that long-range continental transport of aerosols can mask the diel cycle.

Fig. S10. Ratio of the daytime to nighttime concentration for aerosols with $D > 0.58\mu\text{m}$ vs the daily mean sea surface temperature at a depth of around 0.5 – 3.0 m. The grey circles show all the data. The black circles show the mean of the data, binned into equally number of points bins ($N=62$ per bin). The red line is the linear fit to the mean data.

28) Figure S11: it is not quite clear – are the averages from days where the cycle is present only or do the averages include data where there is not cycle in the airborne particles? How many days were involved in each of the averages shown?

Answer: Since we do not separate the data by the days where there was a cycle or not, we revised Fig. S11 (now Fig. S12 in the revised SI) to show the differences between the Atlantic and Pacific, and the latitudinal dependence within the Pacific ocean.

Fig. S12. Mean particle size index, γ , in different oceanic regions. (A) In the Atlantic (black circles), and in the Pacific Ocean (open circles). (B) During the Keelung – Fiji transect for latitudes above 10°N (red circles; $N=10$), between the Equator and 10°N (orange squares; $N=9$), between the Equator and 10°S (green triangles; $N=8$), and below 10°S (blue diamonds; $N=4$). The average γ values while *Tara* was anchored near Niue Island are also shown (open purple triangles; $N=7$). The shaded areas are 1σ , and ‘ N ’ refers to the number of days analyzed.

29) Were there some days where a cycle in gamma was observed but not in the airborne particles? or was the cycle in gamma always and only occurring when there was a cycle in the airborne particles (larger than 0.58 micrometer)?

Answer: In the revised manuscript we no longer separate the data by the definition of when there was a diel cycle or not

Minor.

30) Line 56: “in high spatial and temporal resolution”: it should be stated what the spatial and temporal resolution were and also the time it took to cover the 42000 km should be given.

It would be nice to state also when the expedition took place even if it is given in the references.

Answer: We added the following phrase: “...in high spatial and temporal (*every minute*) resolution along 42,000 km over the Atlantic Ocean, Caribbean Sea, and Pacific Oceans, measured aboard the schooner *Tara* during the *Tara* Pacific Expedition (Flores et al., 2020; Gorsky et al., 2019; Planes et al., 2019). *The expedition began on 28 May 2016 from Lorient, France and finished the first year of the campaign on June 17, 2017 in Whangarei, New Zealand.*”

- Planes, S. *et al.* The *Tara* Pacific expedition—A pan-ecosystemic approach of the “-omics” complexity of coral reef holobionts across the Pacific Ocean. *PLoS Biol.* **17**, e3000483 (2019).
- Gorsky, G. *et al.* Expanding *Tara* Oceans Protocols for Underway, Ecosystemic Sampling of the Ocean-Atmosphere Interface During *Tara* Pacific Expedition (2016–2018). *Front. Mar. Sci.* **6**, (2019).
- Flores, J. M. *et al.* *Tara* Pacific Expedition’s Atmospheric Measurements of Marine Aerosols across the Atlantic and Pacific Oceans: Overview and Preliminary Results. *Bulletin of the American Meteorological Society* **101**, E536–E554 (2020).

31) Line 60: “AMBL and oceanic (salinity, sea surface temperature, and chlorophyll *a*) variables. Only oceanic variables are listed, the variables measured for AMBL should also be listed.

Answer: We added the variables. The phrase now reads: “as well as AMBL (**temperature, RH, wind speed and direction, and photosynthetic active radiation**) and oceanic (salinity, sea surface temperature, and chlorophyll *a*) variables.”

32) Line 103: I do not see that reference 11 shows that the regions near Japan and New Zealand were heavily polluted during the *Tara* cruise – is it the correct reference?

Answer: We deleted this phrase

33) Figure 1:

I suggest to write the names of the oceans on Figure 1 (they are given in figure 2 but used before that).

Answer: We updated Figure 1 (Figure 3 in the revised manuscript) to include the oceans names (see below).

34) In line 80 it says that the average ratio is 2.3 over the tropical Pacific Ocean. In figure S1 it is averages over the Pacific ocean. Is it the same time periods, that are referred to in figure S1 and in Figure 2?

Answer: No, in Figure 2 is from all the days that we could analyze, and in Figure S1 we took the data that was 100km away from islands or continents. We added in the Figure legend of Fig. S1 the number of days that were analyzed: “..., in total 124 days (~2978 hours) were analyzed”

35) Line 86 it says “pristine Pacific Ocean” – what is the definition of pristine here- is it the 100 km from the coast?

Answer: Thank you for this comment. We deleted the word pristine since it's not consistent. We measured the diel pattern next to habitable islands which, by definition, are not pristine.

36) Line 156: "Furthermore, the presence of the NSSA_0.58 μ m diel cycle in the presence of anthropogenic pollutants and in clean conditions (Fig. 1A,B and Fig. S5), implies anthropogenic and continental sources are also not the cause." On one of the maps it could be shown where the authors consider conditions clean and where they consider them polluted.

Answer: We added this information to Fig.1 (now Fig. 3); we added two colors to distinguish between clean and polluted. See the revised Fig. 3 above in the answer to comment 33). We update the legend to read: "**Figure 3. 24-hour cycle of aerosol concentration and marine particle size index.** Top panel: Map of R/V *Tara*'s route, with dotted arrows along the sailing direction and solid black lines along the 48 hour back-trajectories. Filled circles on the route are colored by the value of day-to-night concentration ratio for $D_{op} \geq 0.58 \mu\text{m}$. The data in panels (a) through (c) are from the *orange and blue-shaded transect* in the western Pacific between Keelung and Fiji (next to the double ended arrow). *The orange-shaded region represents anthropogenic polluted conditions, and the blue-shaded refers to clean ones.*"

37) The authors write "litter" as unit in several places - I think it should be "Liter "

Answer: Thank you, we changed the places where we found this typo to "litre" (given reviewer#1 asked for this same mistake to be written as 'litre')

38) Figure S1: It should be stated for how long time (days/hours) was averaged over (the time more than 100 km in the Pacific ocean) This should be stated in the Figure caption.

Answer: We updated the caption, it now reads: "**Diurnal box plot analysis of total concentration of aerosol diameters per hour for 16 bins of the OPC.** The box plot analysis shows the median, and the 5th, 25th, 75th, and 95th percentiles. Only data from the Pacific Ocean and when *Tara* was at least 100 km away from land were used for this analysis, *in total 124 days (~2978 hours) were analyzed.* At the top of each panel the lower and upper limit in micrometers of the OPC bins are shown..."

39) Figure S2: In the figure caption it says concentration per hour? on the plots the y-axes just give concentration? It should be stated in the Figure caption what "N" is.

Answer: We no longer use this Figure in the revised SI.

40) Figure S10: In the caption it says gamma – but gamma is not shown.

Answer: Thank you, we no longer use this Figure in the revised SI.

41) Figure S10 –What is shown on the y-axis – is it an average of how many days – all days in the Pacific or were some days left out? This should be stated,

Answer: We no longer use this Figure in the revised SI.

42) In general it should say what the shaded areas in the figures are – is it always one standard deviation of the average?

Answer: Yes, it is one standard deviation. We added this to all relevant legends.

References

Effect of Clouds on the Diurnal Evolution of the Atmospheric Boundary-Layer Height Over a Tropical Coastal Station, Edwin V. Davis, K. Rajeev & Manoj Kumar Mishra, *Boundary-Layer Meteorology* volume 175, pages 135–152 (2020)

Reviewer #3 (Remarks to the Author):

This paper reports a field aerosol measurement over Pacific Ocean and the Caribbean Sea. It reports an interesting and novel finding that the number concentrations of large marine aerosols ($D_p > 580$ nm) would show a diel cycle. This paper proposes that the diel cycle might relate to biological activity in seawater. I have some major concerns for this study.

We would like to thank reviewer#3 for the time and effort spent reviewing our manuscript. We are grateful for all the comments that helped us improve the paper. We address all of the comments point-by-point, our answers are marked in blue.

(1) The aerosol number concentrations that this paper reports seem to be very low. A typical marine aerosol concentration is at least several hundred particles/cm³ in a pristine environment [Lewis & Schwartz 2004]. The number concentration for particles larger than 0.25 μm would contribute at least ~30% of the total concentration. However, the OPC measurement (Fig.S1) shows that the total aerosol concentration seems to be fewer than 10 particles/cm³. It does not look quite right to me. The OPC is not a good instrument to measure submicron particles. I think the authors need to provide a comparison of particle size distribution measurement between the OPC and a SMPS.

Answer: We understand the reviewer's concerns about the number concentration and the use of an OPC. One of the first analyses we did to verify our data made sense, was a comparison between the OPC and the SMPS. We did not provide it in this manuscript, but we have done so in two previous publications regarding this expedition.

One in Flores et al., *Bull. Amer. Meteor. Soc.*, 2020; <https://doi.org/10.1175/BAMS-D-18-0224.1>, and the other in Dror et al., *Atmos. Chem. Phys.*, 2020; <https://doi.org/10.5194/acp-20-15297-2020>.

In Flores et al. 2020 we described the expedition and in Dror et al. 2020, we use six different full size distributions we measured in the Atlantic, Caribbean Sea, and Pacific Ocean to study their effect on warm cloud properties.

In Flores et al., 2020 we show that we applied a particle loss calculation to the inlets using the particle loss calculator (von der Weiden et al., 2009), and for the cases where the SMPS was working, we corrected the OPC data following the scheme described in Hand and Kreidenweis, 2002. However, for the majority of the data points the correction was minimal. Below we are attaching Fig. 4 from Flores et al., *BAMS*, 2020.

Figure 4b shows an example of two size distributions after correction. One from the Atlantic Ocean leg and one from the Keelung to Fiji leg. In the case of these size distributions, the bins from the OPC were not changed. It can be seen that the overlap region coincides very well and gives us confidence that the concentrations we are showing in this manuscript are reliable.

Based on the reviewer's comments, we added the following sentence to the section "Continuous aerosol instrumentation and inlet" in the Methods: "***Particle loss corrections to the OPC were applied taking into***

account the inlet's length and the tube's internal diameter using the particle loss calculator (von der Weiden et al., 2009). During the Atlantic crossing and the Keelung to Fiji leg, a scanning mobility particle sizer was functioning and the overlap region with the OPC was used to check and correct it if necessary; see Flores et al. 2020 for details.”

- Flores, J. et al. Tara Pacific Expedition's Atmospheric Measurements of Marine Aerosols across the Atlantic and Pacific Oceans: Overview and Preliminary Results. *Bulletin of the American Meteorological Society* 101, E536–E554 (2020).
- Dror, T., Flores, J. M., Altaratz, O., Dagan, G., Levin, Z., Vardi, A., and Koren, I.: Sensitivity of warm clouds to large particles in measured marine aerosol size distributions – a theoretical study, *Atmos. Chem. Phys.*, 20, 15297–15306, <https://doi.org/10.5194/acp-20-15297-2020>, 2020.
- Von der Weiden, S.-L., F. Drewnick, and S. Borrmann, 2009: Particle loss calculator—A new software tool for the assessment of the performance of aerosol inlet systems. *Atmos. Meas. Tech.*, 2, 479–494, <https://doi.org/10.5194/amt-2-479-2009>.

Fig. 4 from Flores et al., BAMS, 2020. Differences in concentration, size distribution, and composition between the Atlantic and Pacific Oceans. (a) Boxplot analysis of the total aerosol concentration in the open ocean (for the Pacific Ocean refers to the period between 3 and 26 May 2017, and for the Atlantic Ocean for the period between 6 and 27 Jun 2016), and near-continental sources (for the Pacific it refers to the period around Japan and Taiwan, 1 Apr to 1 May 2017, and for the Atlantic Ocean the period between France and the Azores, 28 May to 5 Jun 2016). (b) Aerosol size distribution for periods of ~ 16 h with similar wind speed of 11 m s^{-1} . The areas where the size distributions were measured are marked in yellow on the black sailing path in Fig. 3. (c) SEM image of a representative Pacific Ocean particle (9–10 May 2017) found in the same period as the size distributions from (b). The energy-dispersive X-ray spectrum are shown below the SEM images. (d) As in (c), but for the Atlantic Ocean (7 Jun 2016).

We have also added the total aerosol concentration for $D > 0.25 \mu\text{m}$ to Fig. 3 (Fig. 1 in the revised manuscript).

Figure 1. Day-to-night ratios for individual bins of the optical particle counter, total marine aerosol count for $D \geq 0.25\mu\text{m}$, and chlorophyll a concentration along R/V *Tara*'s route. Top panel: *Tara*'s route, color-coded by month; (A) The ratio (5-day running average) of day-to-night concentration of marine aerosol for each size bin of the OPC; (B) The total aerosol count (daily mean) for $D \geq 0.25\mu\text{m}$, the grey dot are the total counts per minute. The colors as defined in the map; (C) a 5-day running average of Chlorophyll- a concentration measured with the AC-S aboard *Tara* (red circles) and calculated using satellite data (green triangles; see methods), with the shaded area outlining the standard deviation. The day-to-night ratios are greater than one on the vast majority of the route and largest in clean areas with low chl- a concentration, i.e., in oligotrophic ("blue") waters.

(2) Another concern is that most of SSA are smaller than $0.58\mu\text{m}$. $\text{NSSA}_{0.58\mu\text{m}}$ only accounts for a smaller fraction of total SSA number concentration. Thus, this study only investigated a small fraction of SSA, rather than total SSA. The atmospheric implication for this small SSA fraction should be discussed specifically.

Answer: Thank you for this comment. The concern from the reviewer made us explore the full size distribution from the OPC and, due to this, as explained in the opening statement of our answers to the reviewers, we have now expanded our analysis to all the diameters measured by the OPC (i.e., $D_{\text{op}} > 0.25\mu\text{m}$), and show there are consistently more aerosols during the day than at night. We now use the $\text{NSSA}_{\geq 0.58\mu\text{m}}$ only as a proxy for the diel cycle to help understand the impact of atmospheric and oceanic variables.

We also added the following paragraph to the conclusions for discussing the atmospheric implications:

“Although the SSA number fraction (for $D > 0.5\mu\text{m}$) in the AMBL, by number, is less than 20% (Flores et al., 2020; Quinn et al., 2017), small concentration changes of big aerosols can drive relatively large changes in cloud sizes, lifetime, and rain yields (Dagan et al., 2015, 2017), especially over pristine areas where clouds are aerosol-limited (Koren et al., 2014).”

- Quinn, P. K., Coffman, D. J., Johnson, J. E., Upchurch, L. M. & Bates, T. S. Small fraction of marine cloud condensation nuclei made up of sea spray aerosol. *Nat. Geosci.* 10, 674–679 (2017).
- Flores, J. M. et al. Tara Pacific Expedition’s Atmospheric Measurements of Marine Aerosols across the Atlantic and Pacific Oceans: Overview and Preliminary Results. *Bulletin of the American Meteorological Society* 101, E536–E554 (2020).
- Dagan, G., Koren, I. & Altaratz, O. Aerosol effects on the timing of warm rain processes. *Geophys. Res. Lett.* 42, 4590–4598 (2015).
- Dagan, G., Koren, I., Altaratz, O. & Heiblum, R. H. Time-dependent, non-monotonic response of warm convective cloud fields to changes in aerosol loading. *Atmos. Chem. Phys.* 17, 7435–7444 (2017).
- Koren, I., Dagan, G. & Altaratz, O. From aerosol-limited to invigoration of warm convective clouds. *Science* 344, 1143–1146 (2014).

(3) The authors argue that the RH, air temperature, and atmospheric stability can be ruled out as the cause of the diel pattern. However, their explanation is weak, as the RH, air temperature, and atmospheric stability clearly showed a diel pattern. It is possible that they might have a delayed effect on SSA production. For example, air temperature needs some time to change the temperature of air-sea interface, which is known to affect SSA production.

Answer: Thank you for this comments as it allowed us to strengthen our explanation. As explained above, we now use a simple ratio and expanded the analysis to all the data available.

To explore the relationship with the atmospheric variables, we use the day-to-night ratio of $N_{\text{SSA} \geq 0.58\mu\text{m}}$ vs. the difference between the daytime mean and nighttime mean for the atmospheric (air temperature, relative humidity, and wind speed) variables to check if there’s a relationship between them. We found no correlation with any of the variables. We used the difference instead of the day-to-night ratio for the variables since it gives a more physical intuition of their diurnal changes. However, we did the same analysis using the day-to-night ratio and found no correlations either (not shown here).

The revised supplementary figure for this analysis:

Fig. S4. Ratio of the day-to-nighttime concentration for aerosols with $D \geq 0.58\mu\text{m}$ vs the mean day-to-nighttime difference of air temperature, relative humidity and wind speed. The DNR for the atmospheric variables were calculated using the same daytime (from 07:00 to 17:00 MST) and nighttime (19:00 to 05:00 MST) periods. The red line is the linear fit to each data set. ‘N’ refers to the number of days analyzed. No correlation was found with any of the three atmospheric variables.

We also updated the section “Atmospheric and oceanic environmental factors that may control SSA production and transport” (previously ‘Atmospheric and oceanic environmental factors’), the paragraph now reads:

“Second, links between the diel changes of RH and air temperature to the N_{SSA} diel cycle were explored. If RH or the air temperature were the main drivers of the diel cycle, by affecting bubble and droplet evaporation, or SSA sizes, we expect to have a clear relationship between the DNR of $N_{SSA \geq 0.58 \mu m}$ and the mean day-to-night differences of RH and air temperature, but we found no correlation between them (Fig. S4). Therefore, while RH and air temperature can affect bubble evaporation, and in consequence, the production rate of SSA, they are most likely not driving the diel cycles in $N_{SSA \geq 0.58 \mu m}$.

The atmospheric stability that influences the transport of aerosols from the ocean surface upward does not fully explain the diurnal cycle either. First, under most atmospheric conditions, concentrations of SSA with $D_{dry} < 10 \mu m$ are well mixed in the marine boundary layer, showing little variation with height (Lewis and Schwartz, 2004); hence, assuming a constant production of SSA, a change in atmospheric stability will most likely not cause a change in N_{SSA} . In addition, during the Taiwan – Fiji transect we took photos of the sky and we could see that the cycle appeared in three distinct atmospheric states: with clear skies at low wind speeds, with overcast conditions, and with trade cumulus throughout the day (see Fig. S5). Especially, that a cycle is observed even if the morning is overcast (Fig. S5b), and that cycles are evident also when there is no air temperature variability (Fig. S4). These observations suggest that atmospheric stability does not play a significant role driving the diel cycles.”

(4) The authors use Figure S9 to argue that “no link between the intensity of photosynthetically active radiation and the daytime $N_{SSA_0.58 \mu m}$ was found”. However, I think Figure 9 can only suggest that there is no significant linear relation between the intensity of PAR and the daytime $N_{SSA_0.58 \mu m}$.

Answer: Thank you, we corrected this phrase. It now reads: “Finally, we did not find a significant linear relationship between the intensity of photosynthetically active radiation and the DNR of $N_{SSA \geq 0.58 \mu m}$ (Fig. S7).”

(5) The authors states that “we also discarded secondary organic aerosol production as the cause of the diel cycle since the produced aerosol by this mechanism is much smaller in diameter ($< 0.1 \mu m$)”, which is an absolute false statement. Secondary organic aerosol (SOA) matter can form and condense on the existing particles. It is totally possible that smaller SSA were coated by SOA during daytime and their diameters became larger than $0.58 \mu m$, thereby causing the diel pattern.

Answer: Thank you for this comments that allowed us to clarify this point. We think that aerosol growth by condensation can also be ruled out due to the known slow growth rates ($< 7 \text{ nm h}^{-1}$) (Paasonen et al. 2018, Burkart et al. 2017), which are too slow to account for the increase in concentration within 30 min after sunrise. Additionally, condensational growth cannot explain the decrease in concentration we observe at sunset. However, we explored the full size distribution of the OPC for the period we show in Figure 1 (see Fig. Q5 below), and we could not see any evidence of particle growth (i.e., a steady increase in concentration and the maximum concentration; a banana plot). There is the possibility that the coarse resolution from the OPC will not detect it. However, based on studies in other regions (we could not find a study of condensational growth rate in the Pacific Ocean), it seems highly unlikely that condensational growth is responsible for the diel pattern we observe in SSA. For example, in boreal forest Paasonen et al.

2018 found growth rates of $\sim 3.5 \text{ nm h}^{-1}$ for initial sizes between 190 - 200nm. In the marine arctic, the growth rates for a preexisting mode of $\sim 90 \text{ nm}$ were less than 6 nm h^{-1} (Burkart et al. 2017).

Based on the reviewers comment, we have rephrased the sentence, it now reads: *“While we cannot discard condensational growth after sunrise, the growth rates are too slow ($< 6 \text{ nm h}^{-1}$) (Burkart et al., 2017; Paasonen et al., 2018) to account for the increase in N_{MA} within 30 min after sunrise (Fig. 3A). Furthermore, condensational growth cannot explain the decrease in concentration at sunset.”*

Figure Q5. Full size distribution of the OPC vs. time during the same period shown in Fig. 1.

- Paasonen, P., Peltola, M., Kontkanen, J., Junninen, H., Kerminen, V.-M., and Kulmala, M.: Comprehensive analysis of particle growth rates from nucleation mode to cloud condensation nuclei in boreal forest, *Atmos. Chem. Phys.*, 18, 12085–12103, <https://doi.org/10.5194/acp-18-12085-2018>, 2018.
- Burkart, J., Hodshire, A. L., Mungall, E. L., Pierce, J. R., Collins, D. B., Ladino, L. A., and Abbatt, J.: Organic condensation and particle growth to CCN sizes in the summertime marine Arctic is driven by materials more semivolatile than at continental sites, *Geophys. Res. Lett.*, 44, 10725–10734, <https://doi.org/10.1002/2017GL075671>, 2017.

Specific comments:

6) Line 31: “global scale assessment of this micro-scale process is highly challenging”: this sentence is very vague. Please be specific. Which assessment is challenging?

Answer: We rephrased to: “but measurements at a global-scale of this micro-scale process are highly challenging”

7) Line 33: “42,000 km of open ocean waters”: sounds strange. Please revise.

Answer: We changed the phrase to: “...covering over 42,000 km.”

8) Line 36: “No correlation” should be changed to “no significant correlation

Answer: The text was changed accordingly.

9) Line 65: “N_{SSA_0.58μm}” is quite misleading. I would use something like “N_{SSA>0.58μm}”.

Answer: We changed the notation to N_{SSA≥0.58μm}.

10) Line 108: I would never use “prove” in any non-mathematical scientific paper.

Answer: We change the word “prove” to “show”.

11) Line 138: please explain in the main text why the RH, air temperature, and atmospheric stability can be ruled out as the cause of the diel pattern. This is very important.

Answer: Thank you, we added the following paragraphs to the main text.

“Other atmospheric environmental factors (e.g., rain, RH, air temperature, and atmospheric stability¹) are known to affect the production and number of SSA, but we found no evidence that they are the main drivers of the observed diel cycle. First, rain suppressed the cycle (Fig. 3A), as it is a known washout mechanism of aerosols. Second, links between the diel changes of RH and air temperature to the N_{SSA} diel cycle were explored. If RH or the air temperature were the main drivers of the diel cycle, by affecting bubble and droplet evaporation, or SSA sizes, we expect to have a clear relationship between the DNR of N_{SSA≥0.58μm} and the mean day-to-night differences of RH and air temperature, but we found no correlation between them (Fig. S4). Therefore, while RH and air temperature can affect bubble evaporation, and in consequence, the production rate of SSA, they are most likely not driving the diel cycles in N_{SSA≥0.58μm}.”

The atmospheric stability that influences the transport of aerosols from the ocean surface upward does not fully explain the diurnal cycle either. First, under most atmospheric conditions, concentrations of SSA with D_{dry} < 10 μm are well mixed in the marine boundary layer, showing little variation with height (Lewis and Schwartz, 2004); hence, assuming a constant production of SSA, a change in atmospheric stability will most likely not cause a change in N_{SSA}. In addition, during the Taiwan – Fiji transect we took photos of the sky and we could see that the cycle appeared in three distinct atmospheric states: with clear skies at low wind speeds, with overcast conditions, and with trade cumulus throughout the day (see Fig. S5). Especially, that a cycle is observed even if the morning is overcast (Fig. S5b), and that cycles are evident also when there is no air temperature variability (Fig. S4). These observations suggest that atmospheric stability does not play a significant role driving the diel cycles. Changes in ABL height do not fully explain the diel cycles in N_{SSA} either. If height variations within the ABL were driving the diel pattern in N_{SSA≥0.58μm}, given there are no changes in the production between day and night, the diurnal concentration changes should follow those seen in the ABL. For the period with clear diel cycles in N_{SSA≥0.58μm} shown in Fig. 3A, we did not find ABL height (using ERA5 reanalysis data (Hersbach et al., 2020); see Methods) diurnal changes that follow those we measured in N_{SSA≥0.58μm}, and the changes we found of the ABL height were less than 40m for a mean height of 3735m (i.e., ~1%; Fig. S6). We cannot exclude a possible influence, but it is unlikely that atmospheric stability or changes in the ABL height are driving the diel cycle.

Finally, we did not find a significant linear relationship between the intensity of photosynthetically active radiation and the DNR of N_{SSA≥0.58μm} (Fig. S7). While we cannot discard condensational growth after sunrise, the growth rates are too slow (< 6 nm h⁻¹) (Burkart et al., 2017; Paasonen et al., 2018) to account for the increase in N_{MA} within 30 min after sunrise (Fig. 3A). Furthermore, condensational growth cannot explain the decrease in concentration at sunset. Thus, atmospheric variables, while they can affect the production, size, and transport of SSA, do not seem likely to drive the observed N_{SSA≥0.58} diel cycles.”

- Lewis, R. & Schwartz, E. *Sea salt aerosol production: mechanisms, methods, measurements and models—a critical review*. **152**, (American Geophysical Union, 2004).

- Hersbach, H. *et al.* The ERA5 global reanalysis. *Q.J Royal Met. Soc.* (2020). doi:10.1002/qj.3803.
- Paasonen, P. *et al.* Comprehensive analysis of particle growth rates from nucleation mode to cloud condensation nuclei in boreal forest. *Atmos. Chem. Phys.* **18**, 12085–12103 (2018).
- Burkart, J. *et al.* Organic condensation and particle growth to CCN sizes in the summertime marine arctic is driven by materials more semivolatile than at continental sites. *Geophys. Res. Lett.* **44**, 10,725–10,734 (2017).

12) Line 161: what is “seawater particles”?

Answer: We updated the introductory paragraph to this section to clarify this point, the paragraph now reads: “The identified N_{SSA} diel cycles in the lower atmosphere were accompanied by distinct diel cycles of the near-surface ocean light attenuation wavelength dependence of c_p (expressed as particle size index γ ; Fig. 1C). *c_p is an inherent optical property that depends, theoretically, on all the particles present in the water column (i.e., autotrophic and heterotrophic micro-organisms, as well as detritus and mineral particles). Given c_p is wavelength-dependent, γ is found via a power-law fit (Boss et al., 2001; Dall’Olmo et al., 2011):*”

- Boss, E., Twardowski, M. S. & Herring, S. Shape of the particulate beam attenuation spectrum and its inversion to obtain the shape of the particulate size distribution. *Appl Opt* 40, 4885 (2001).
- Dall’Olmo, G. et al. Inferring phytoplankton carbon and eco-physiological rates from diel cycles of spectral particulate beam-attenuation coefficient. *Biogeosciences* 8, 3423–3439 (2011).

13) Line 183: documented in what?

Answer: We deleted the word ‘in’ to make the sentence clear.

14) Line 183: “Changes in γ can be attributed to several factors: cell growth, division, and aggregation, selective changes in particle size or concentration due to a balance between primary production and loss due to grazing and viral pressure, or changes in the refractive index of the cell population, which is related to their carbon content. “: how can the authors contribute the change of aerosol particle size to cell growth, division, and aggregation etc? Or they are actually POC in seawater. It is confusing.

Answer: We rephrase to make this sentence clearer, the phrase now reads: “Changes in γ can be driven by different reasons; for example, cell growth, division, and aggregation, selective changes in particle size or concentration due to a balance between primary production and cell loss due to grazing and viral pressure, or changes in the refractive index of the cell population, which is related to their carbon content.”

15) Line 207: Poulain & Bourouiba (literature 34) does not show any results of bubble bursting aerosol flux. The production rate of film drop is an extremely complicated problem. How organic surfactant affect this rate is still largely unknown.

Answer: We agree with the reviewer that Poulain & Bourouiba did not explicitly measured aerosol flux, however, they do conclude that “The shift in thinning law makes the droplets produced by contaminated [by bacterial secretions] bubbles smaller, faster, and more numerous than those produced by clean bubbles.” Looking at their Fig. 1 it is clear that there are more droplets formed after the bubble burst when bacterial secretions were present. We thus made the possible link between the number and size of the formed droplets and the derived aerosol flux.

To make this point clearer we rephrase this sentence to:

“Recently, in a different context, EPS released by bacteria were found to increase bubble lifetime, thereby dramatically decreasing their film thickness, and yielding more numerous and transportable droplets at burst than those produced by clean bubbles (see Fig. 1 in (Poulain and Bourouiba, 2018)). This observations can be directly linked to SSA formation since it is directly related to film drops formed by the fragmentation of the thin fluid cap film (Wang et al., 2017).”

- Poulain, S. & Bourouiba, L. Biosurfactants Change the Thinning of Contaminated Bubbles at Bacteria-Laden Water Interfaces. *Phys. Rev. Lett.* **121**, 204502 (2018).
- Wang, X. *et al.* The role of jet and film drops in controlling the mixing state of submicron sea spray aerosol particles. *Proc. Natl. Acad. Sci. USA* **114**, 6978–6983 (2017).

16) Line 209: Again, SSA flux is an extremely complicated problem. In our laboratory, we studied the effect of biology in seawater on SSA flux but got inconsistent, sometimes contradictory results. I am not convinced that the hypothesis proposed by this paper is well supported by existing evidence.

Answer: We agree that SSA flux is a complicated problem. However, we do think that, given the limited measurements at hand, we provide evidence that atmospheric variables are not likely to be the main drivers of the cycle observed, leaving cyclical oceanic processes as the main suspects, and that to pinpoint the mechanism more research needs to be done. Based on the reviewer’s comment, we further explored the relationships with some of the oceanic variables we measured.

We found, on the one hand, that the daily mean sea surface temperature (SST) positively correlates with the magnitude of the day-to-night ratio of $N_{SSA \geq 0.58 \mu m}$ (see Fig. S10 below), but the day-to-night ratio of $N_{SSA \geq 0.58 \mu m}$ does not correlate with mean day-to-night differences of the SST (Fig. S8 below). This suggests that SST is not likely to drive the $N_{SSA \geq 0.58 \mu m}$ diel cycles, but it does play a role in modulating the intensity of the DNR, which supports our hypothesis of the diel cycle being driven by ocean processes.

Fig. S10. Ratio of the daytime to nighttime concentration for aerosols with $D > 0.58 \mu m$ vs the daily mean sea surface temperature at a depth of around 0.5 – 3.0 m. The grey circles show all the data. The black circles show the mean of the data, binned into equally number of points bins ($N=62$ per bin). The red line is the linear fit to the mean data.

Fig. S8. Ratio of the day-to-nighttime concentration for aerosols with $D > 0.58\mu\text{m}$ vs mean day-to-nighttime difference of salinity, sea surface temperature, and chlorophyll a at a depth of around 0.5 – 3.0 m. The red line is the linear fit to all the available data. ‘N’ refers to the number of days analyzed.

Moreover, we further explored the relationship with the rate of change of gamma, which tells us if there is an increase of the mean particle size ($\partial\gamma/\partial t < 0$) within the ocean surface during the day (which has been shown to be driven by photosynthetic growth and secretion of extracellular polymeric substances), or a decrease in mean particle size ($\partial\gamma/\partial t > 0$) (suggesting cell division). Below we show $N_{\text{SSA} \geq 0.58\mu\text{m}}$ vs. $\partial\gamma/\partial t$ for the Pacific Ocean data. As expected, we found a negative correlation, which further supports our results. We think that Figure 4 (now Figure 6) within the manuscript shows quite well this relationship, and the analysis we’ve done here adds support. Therefore, we decided to add this figure to the SI; it’s now Figure S11.

Fig. S11. $N_{\text{SSA} \geq 0.58\mu\text{m}}(\text{h}^{-1})$ vs. $\partial\gamma/\partial t (\text{h}^{-1})$ for the Pacific Ocean data. The grey circles show all the data. The red circles show the mean of the data, binned into equally number of points bins ($N=223$ per bin). The red line is the linear fit to the mean data. Pr refers to the Pearson correlation coefficient.

These two new analyses add support to our hypothesis, but in the concluding paragraph we openly address that we do not have the measurements to pinpoint the mechanism, we write: “While *we do not provide (or possess) direct measurements of near-surface water microbial processes*, the parallel increase of the mean particle size within the ocean surface during the day, driven by photosynthetic growth and secretion of extracellular polymeric substances, *points towards a possible link* between microbial processes at the ocean surface and the N_{SSA} cycle.

Although the SSA number fraction (for $D > \sim 0.5 \mu\text{m}$) in the AMBL, by number, is less than 20% (Flores et al., 2020; Quinn et al., 2017), small concentration changes of big aerosols can drive relatively large changes in cloud sizes, lifetime, and rain yields (Dagan et al., 2015, 2017), especially over pristine areas where clouds are aerosol-limited (Koren et al., 2014). Consequently, *the discovery of the diel cycle in N_{SSA} opens many new questions for future studies to elucidate the mechanism underlying this phenomenon and the direct impact of marine biological processes on the physical properties of the surface ocean, and the link to aerosol fluxes and properties*. Moreover, on a larger scale, the connection to cloud properties and consequently to energy fluxes and climate.”

- Quinn, P. K., Coffman, D. J., Johnson, J. E., Upchurch, L. M. & Bates, T. S. Small fraction of marine cloud condensation nuclei made up of sea spray aerosol. *Nat. Geosci.* 10, 674–679 (2017).
- Flores, J. M. et al. Tara Pacific Expedition’s Atmospheric Measurements of Marine Aerosols across the Atlantic and Pacific Oceans: Overview and Preliminary Results. *Bulletin of the American Meteorological Society* 101, E536–E554 (2020).
- Dagan, G., Koren, I. & Altaratz, O. Aerosol effects on the timing of warm rain processes. *Geophys. Res. Lett.* 42, 4590–4598 (2015).
- Dagan, G., Koren, I., Altaratz, O. & Heiblum, R. H. Time-dependent, non-monotonic response of warm convective cloud fields to changes in aerosol loading. *Atmos. Chem. Phys.* 17, 7435–7444 (2017).
- Koren, I., Dagan, G. & Altaratz, O. From aerosol-limited to invigoration of warm convective clouds. *Science* 344, 1143–1146 (2014).

17) Line 228: “the lifetime of aerosols is inversely proportional to their size”: this statement is not quite correct. For submicron marine aerosols, the main way to remove them from the air is through wet deposition, which is insensitive to the aerosol sizes. Only for larger aerosol particles, when dry deposition is their main scavenge mechanism, then their lifetime is inversely proportional to their size.

Answer: Thank you for this comment that helped us clarify this point. We rephrase the sentence to: “the lifetime of aerosols, *excluding rain events*, is inversely proportional to their size”

18) Line 245: how the ship engine exhaust was avoided during the aerosol sampling?

Answer: We described the influence of the boat and instrumentation setup in the SI section “S1. Influence of the boat and instrumentation setup on the measured SSA concentration”. In short:

We only experienced engine contamination in the Atlantic crossing when the inlet was ~15m above mean sea level, and cleaned the data accordingly. When we moved the inlet to the top of the mast, we did not observe any evidence the engine contaminated our measurement.

Here we quote the description we wrote in Appendix B of Flores et al., *Bull. Amer. Meteor. Soc.*, 2020, about how we handle engine contamination in the Atlantic crossing: “For an accurate representation of the marine size distributions, three steps were taken for the SMPS–OPC data. First, engine contamination periods were identified. Taking advantage that the R/V *Tara* is a schooner, the engines were only turned on when necessary. These periods were recorded by the crew and were used as the first filter to detect

possible contamination periods. If a drastic increase was identified in the transitions from sailing to engines on, the data were excluded. Additionally, contamination periods were identified in the SMPS–OPC system as a single-mode size distribution with a mode diameter of less than $0.04 \mu\text{m}$ and total particle concentrations on the order of 10^5 cm^{-3} . Contamination periods were identified only in the first Atlantic crossing and were excluded from the analysis.”

- Flores, J. M. et al. Tara Pacific Expedition’s Atmospheric Measurements of Marine Aerosols across the Atlantic and Pacific Oceans: Overview and Preliminary Results. *Bulletin of the American Meteorological Society* 101, E536–E554 (2020).

19) Line 449: “Hence, if RH variations were the cause, we expect to see the diurnal patterns in all sizes, and especially at smaller diameters”: I do not understand the logic here. The particles were dried before measurement, right? The RH could affect bubble evaporation, and it is possible that the production rate of bubble bursting aerosol is affected.

Answer: We deleted this statements, please see the answer to comment 3) for the full paragraph.

20) Line 465: Again, Figure 9 can only suggest that there is no significant linear relation between the intensity of PAR and the daytime $\text{NSSA}_{0.58\mu\text{m}}$.

Answer: We changed the phrase “No clear correlation between PAR and $\text{N}_{\text{SSA}\geq 0.58\mu\text{m}}$ was found.” To “No significant linear relation between PAR and $\text{N}_{\text{SSA}\geq 0.58\mu\text{m}}$ was found.”

REVIEWER COMMENTS

Reviewer #1 (Remarks to the Author):

I would like to thank the authors for taking the time to address my detailed comments along with those of the other reviewers. While the data cannot fully explain the mechanism(s) driving the differences in particle concentrations between day and night that the authors observe it is my view that enough information is included to suggest which processes are more or less likely. Given this, and the fact that I would like to see this work spur further research around this topic, I would like to see this work published. That said I have a number of comments based on the response to reviewers document that I think deserve attention by the authors prior to publication.

1) The authors point to a comparison between their optical particle size spectrometer and a scanning mobility particle sizing system capable of sizing and counting particles down to less than 50nm, the data of which were presented in an earlier article. Given that this data exists it really begs the question what did the authors see when they looked at the day night ratio for all particle sizes measured? I am sure they have looked and presumably there is no significant difference hence the data are not presented here. However, I think some mention of this is warranted in the manuscript.

Along the same lines, I would argue that the authors should use the full aerosol size distribution to determine whether the condensation of vapours is growing particles into the size range the optical particle size spectrometer is sensitive to. That is, I would also like to see a version of Q5 where the y-axis covers the entire range of the SMPS and OPSS.

2) Figure S10 which presents the ratio of the daytime to nighttime concentration for aerosols with $D > 0.58\mu\text{m}$ vs the daily mean sea surface temperature is certainly intriguing and leads to a number of questions. Firstly, on a very practical level, what is the binning criteria the authors have used? The following is stated "The black circles show the mean of the data, binned into equally number of points bins ($N=62$ per bin)". I suspect the number of bins will have a significant impact on the fit of the linear regression so some reasoning on the number of bins would be useful here (this plot catches the eye hence my question on more detail).

3) The authors now state the following in the manuscript: "Given that the aerosol was dried to an $\text{RH} < 40\%$, which is below the efflorescent point of NaCl(Gupta et al., 2015), we can consider the measured sea salt by the OPC to be dry sea salt. The geometrical diameters of dry sea salt are in general 4 – 30 % larger than the dry optical diameter(Flores et al., 2009) (e.g., $D_{\text{geo}} = 0.3 \mu\text{m}$ is approximately $D_{\text{op}} = 0.28 \mu\text{m}$; see Fig. S1), therefore, the vast majority of the particles of all diameters measured by the OPC showing the diel pattern, can be considered to be dry sea salt. "

As I am sure the authors are aware, seawater and the aerosols that result from bubbles bursting in seawater are a little more complex than purely NaCl. Critically, some of the calcium and magnesium salts present in sea spray aerosol likely start to grow at very low RH (as low as perhaps 10%) [e.g. Guo et al., 2019; Zieger et al., 2017]. Given this I would be careful with stating that sea spray aerosol is dry at an RH of 40%.

Matt Salter

Reviewer #2 (Remarks to the Author):

Re-Review of "Diel cycle of sea spray aerosol concentration over vast areas of the tropical Pacific Ocean and the Caribbean Sea by Flores et al."

I find that the authors have done fine job in answering my comments and that new insight has even come out and been presented.

In particular, the authors have addressed my three major concerns: 1) the inconsistencies in use of baselines and time-periods have been resolved. I find that the use of day to nighttime ratio (DNR) is a good choice and the new figure 2 is nice, 2) the authors have discussed the particle phase state with respect to drying, and 3) the authors have further looked at gamma and added e.g. figure S10 and S11 to the supporting material.

The authors present an impressive set of field data and the topic, findings and discussions are timely and relevant.

I recommend publication of the revised manuscript in Nature Communications. A few minor comments, which the authors can consider, are given below.

Merete Bilde

Minor comments:

Figure 1 caption: It says that the map is color coded by month – but the color coding is not shown – I suggest to show that as a legend. Perhaps also add arrows pointing in the direction the ship was sailing.

Page 4 lines 85-89: the classification into size ranges is typically done for dried particles or particle sizes measured at a certain RH. I suggest to specify this.

Page 7 line 174: It could be relevant to mention that even though the sea salt particles are dried they may still contain water bound in the form of hydrates (see Rosati et al. 2020). In fact, water in this form may constitute a significant fraction of the particle volume of the dried sea salt particles.

Rosati et al. Reconciling atmospheric water uptake by hydrate forming salts. Environmental Science: Processes & Impacts 22, 1759-1767 (2020).

Reviewer #3 (Remarks to the Author):

The authors had addressed all my questions and concerns very well. The paper is publishable now.

REVIEWER COMMENTS

We would like to thank again the reviewers for the time and effort they invested in our work. We do not take for granted the high quality of the reviews. We believe their reviews significantly helped make our manuscript better.

The answers to their comments are marked in blue below.

Reviewer #1 (Remarks to the Author):

I would like to thank the authors for taking the time to address my detailed comments along with those of the other reviewers. While the data cannot fully explain the mechanism(s) driving the differences in particle concentrations between day and night that the authors observe it is my view that enough information is included to suggest which processes are more or less likely. Given this, and the fact that I would like to see this work spur further research around this topic, I would like to see this work published. That said I have a number of comments based on the response to reviewers document that I think deserve attention by the authors prior to publication.

We would like to thank again Dr. Salter for his constructive reviews.

1) The authors point to a comparison between their optical particle size spectrometer and a scanning mobility particle sizing system capable of sizing and counting particles down to less than 50nm, the data of which were presented in an earlier article. Given that this data exists it really begs the question what did the authors see when they looked at the day night ratio for all particle sizes measured? I am sure they have looked and presumably there is no significant difference hence the data are not presented here. However, I think some mention of this is warranted in the manuscript.

Answer: Thank you for bringing this point up to clarify it. We did in fact looked at it, but we decided not show it or mention it for the following reasons:

1) Contrary to the OPC data, we only have a little less than 2 months of data from the SMPS in only two regions, and we do not have any elemental or chemical data for this size range.

2) As we expected and Dr. Salter correctly suspected, we do not see any difference. We added an extended version of Fig. 2 below (Fig. Q1), where it is clear that the DNR ~ 1 for all diameters below 0.25 μm . We added this figure to the SI. In the revised version it is Fig. S1

3) We believe that the manuscript as it is presented now, it is self-contained; with the SEM-EDX analysis we were able to show that the diel cycle in marine aerosol is in fact SSA. Whereas, in the size range of the SMPS we do not have any way to check their elemental or chemical signature, we cannot distinguish between SSA and continental aerosol.

However, based on Dr. Salter's comment we added the following sentence (marked in bold and italic) to the section 'Diel patten in marine aerosol number concentration': "The DNR were calculated across *Tara's* route for each size bin of the OPC and found it to be > 1 for all sizes on the vast majority of the route (Fig. 1). DNR are shown to depend strongly on the aerosol diameter and the region (Fig. 1 and Fig. 2). For smaller aerosol diameters ($0.25 < D_{op} < 0.5 \mu\text{m}$) the DNR is closer to one (***for the Atlantic and along the Keelung to Fiji leg in the Pacific Ocean, we could calculate the DNR for $D < 0.25\mu\text{m}$ and***

found it to be ~1 for all diameters down to ~0.03 μm ; Fig. S1), and it becomes larger as the diameters increase, reaching DNR > 10 for $D_{\text{op}} > 1.0 \mu\text{m}$.”

Fig. Q1. Mean day-to-night aerosol count ratio for each bin for different regions across *Tara's* route. The full markers are the same as in Fig. 2 of the main text. The open markers show two legs where we calculated the DNR for $D < 0.25\mu\text{m}$ using SMPS data. The Pacific leg refers to the leg between Keelung and Fiji in the western Pacific. For clarity, the SMPS data was binned into 25 nm segments and then the DNR was calculated. The shaded areas represent 1σ , and only the top parts are shown for clarity. ‘N’ refers to the number of days analyzed.

Along the same lines, I would argue that the authors should use the full aerosol size distribution to determine whether the condensation of vapours is growing particles into the size range the optical particle size spectrometer is sensitive to. That is, I would also like to see a version of Q5 where the y-axis covers the entire range of the SMPS and OPSS.

Answer: Below, we explored the full size distribution of the SMPS-OPC data combined for the period we show in Figure 1 (see Fig. Q1_2 below). We normalized the data to the maximum concentration to better visualize it.

Also here we could not see any evidence of particle growth (i.e., a steady increase in concentration and the maximum concentration; a banana plot). Instead we see a variation between the Aitken and Accumulation modes typical to marine aerosol size distributions.

Figure Q1_2. Full size distribution normalized to the maximum concentration of the SMPS-OPC vs. time during the same period shown in Fig. 1.

2) Figure S10 which presents the ratio of the daytime to nighttime concentration for aerosols with $D > 0.58\mu\text{m}$ vs the daily mean sea surface temperature is certainly intriguing and leads to a number of questions. Firstly, on a very practical level, what is the binning criteria the authors have used? The following is stated "The black circles show the mean of the data, binned into equally number of points bins ($N=62$ per bin)". I suspect the number of bins will have a significant impact on the fit of the linear regression so some reasoning on the number of bins would be useful here (this plot catches the eye hence my question on more detail).

Answer: We decided to bin the data into equally number of points per bin since we have a small disparity if we binned them into SST intervals. For example, if we binned the data into 5°C intervals, we get: 36 data point for the $10\text{-}15^\circ\text{C}$ bin, 25 for the $15\text{-}20^\circ\text{C}$ bin, then 78, 99, and finally for the $>30^\circ\text{C}$ bin 64. We, nonetheless, performed this analysis, shown in Fig. Q2 below, and see that the result does not vary significantly. Therefore, we don't think it is necessary to add or change this figure.

Figure Q2. Ratio of the daytime to nighttime concentration for aerosols with $D > 0.58\mu\text{m}$ vs the 151 daily mean sea surface temperature at a depth of around 0.5 – 3.0 m. The grey circles show all 152 the data. The black circles show the mean of the data, binned into equally number of points bins (N=62 per bin). The red line is the linear fit to the mean data. The orange squares show the data binned into 5°C intervals, and the dotted-orange-line is the linear fir to this data.

3) The authors now state the following in the manuscript: "Given that the aerosol was dried to an RH < 40%, which is below the efflorescent point of NaCl(Gupta et al., 2015), we can consider the measured sea salt by the OPC to be dry sea salt. The geometrical diameters of dry sea salt are in general 4 – 30 % larger than the dry optical diameter(Flores et al., 2009) (e.g., $D_{geo} = 0.3 \mu\text{m}$ is approximately $D_{op} = 0.28 \mu\text{m}$; see Fig. S1), therefore, the vast majority of the particles of all diameters measured by the OPC showing the diel pattern, can be considered to be dry sea salt."

As I am sure the authors are aware, seawater and the aerosols that result from bubbles bursting in seawater are a little more complex than purely NaCl. Critically, some of the calcium and mangesium salts present in sea spray aerosol likely start to grow at very low RH (as low as perhaps 10%) [e.g. Guo et al., 2019; Zieger et al., 2017). Given this I would be careful with stating that sea spray aerosol is dry at an RH of 40%.

Answer: We rephrased the paragraph to: "This two-week SEM-EDX analysis where clear diel patterns were seen confirms the primarily type of aerosol exhibiting diel cycles to be sea salt. **The diameters measured in the filters using the SEM are the dry geometrical diameters, which differ from the OPC diameters that depend on the particle's shape and refractive index (RI) compared to that of the particles used for the OPC calibration (typically polystyrene latex spheres with an RI = 1.59(Flores et al., 2009)). The difference between the measured particle RI and the RI of the particles used for calibration causes the discrepancy between D_{geo} and D_{op} . For instance, the geometrical diameters of dry sea salt are generally 4 – 30 % larger than the dry optical diameter(Flores et al., 2009) (e.g., $D_{geo} = 0.3 \mu\text{m}$ is approximately $D_{op} = 0.28 \mu\text{m}$; see Fig. S2). The OPC measured the aerosol at an RH < 40%, which is below the efflorescent point of NaCl(Gupta et**

al., 2015), though some SSA may still contain water bound in the form of hydrates(Guo et al., 2019; Rosati et al., 2020; Zieger et al., 2017). Therefore, most aerosols measured by the OPC during the diel cycles will be sized smaller than their geometrical size. Consequently, the vast majority of the particles of all diameters measured by the OPC showing the diel pattern can be considered dry or slightly humidified sea salt.”

Matt Salter

Reviewer #2 (Remarks to the Author):

Re-Review of “Diel cycle of sea spray aerosol concentration over vast areas of the tropical Pacific Ocean and the Caribbean Sea by Flores et al.”

I find that the authors have done fine job in answering my comments and that new insight has even come out and been presented.

In particular, the authors have addressed my three major concerns: 1) the inconsistencies in use of baselines and time-periods have been resolved. I find that the use of day to nighttime ratio (DNR) is a good choice and the new figure 2 is nice, 2) the authors have discussed the particle phase state with respect to drying, and 3) the authors have further looked at gamma and added e.g. figure S10 and S11 to the supporting material.

The authors present an impressive set of field data and the topic, findings and discussions are timely and relevant.

I recommend publication of the revised manuscript in Nature Communications. A few minor comments, which the authors can consider, are given below.

Merete Bilde

We would like to thank Prof. Bilde for her comments and constructive reviews.

Minor comments:

Figure 1 caption: It says that the map is color coded by month – but the color coding is not shown – I suggest to show that as a legend. Perhaps also add arrows pointing in the direction the ship was sailing.

-Answer: Thank you, we added the suggestions. The new Figure 1 is appended below.

Figure 1. Day-to-night ratios for individual bins of the optical particle counter, total marine aerosol count for $D \geq 0.25 \mu\text{m}$, and chlorophyll a concentration along R/V Tara's route. Top panel: Tara's route, color-coded by month *and dotted arrows showing the sailing direction*; (A) The ratio (5-day running average) of day-to-night concentration of marine aerosol for each size bin of the OPC; (B) The total aerosol count (daily mean) for $D \geq 0.25 \mu\text{m}$, the grey dot are the total counts per minute. The colors as defined in the map; (C) a 5-day running average of Chlorophyll-a concentration measured with the AC-S aboard Tara (red circles) and calculated using satellite data (green triangles; see methods), with the shaded area outlining the standard deviation. The day-to-night ratios are greater than one on the vast majority of the route and largest in clean areas with low chl-a concentration, i.e., in oligotrophic ("blue") waters.

Page 4 lines 85-89: the classification into size ranges is typically done for dried particles or particle sizes measured at a certain RH. I suggest to specify this.

Answer: We clarified this, the phrase now reads: “To distinguish between different aerosol sources, it is common to discuss the different aerosol types and properties according to their sizes (***for dried particles or particle sizes measured at a certain RH***), where different regimes of the aerosol size distribution belong to different aerosol types and processes(de Leeuw et al., 2014). For instance, the freshly produced marine secondary organic aerosol (SOA) are small in diameter (< 0.1 μm); whereas, the typical size range of SSA is larger (> 0.1 μm ***at an RH ~80%***)(Lewis and Schwartz, 2004).”

Page 7 line 174: It could be relevant to mention that even though the sea salt particles are dried they may still contain water bound in the form of hydrates (see Rosati et al. 2020). In fact, water in this form may constitute a significant fraction of the particle volume of the dried sea salt particles.

Answer: Thank you for this comment. Together with the comment from reviewer #1, we rephrased the paragraph to: “This two-week SEM-EDX analysis where clear diel patterns were seen confirms the primarily type of aerosol exhibiting diel cycles to be sea salt. ***The diameters measured in the filters using the SEM are the dry geometrical diameters, which differ from the OPC diameters that depend on the particle’s shape and refractive index (RI) compared to that of the particles used for the OPC calibration (typically polystyrene latex spheres with an RI = 1.59(Flores et al., 2009)). The difference between the measured particle RI and the RI of the particles used for calibration causes the discrepancy between D_{geo} and D_{op} . For instance, the geometrical diameters of dry sea salt are generally 4 – 30 % larger than the dry optical diameter(Flores et al., 2009) (e.g., $D_{\text{geo}} = 0.3 \mu\text{m}$ is approximately $D_{\text{op}} = 0.28 \mu\text{m}$; see Fig. S2). The OPC measured the aerosol at an RH < 40%, which is below the efflorescent point of NaCl(Gupta et al., 2015), though some SSA may still contain water bound in the form of hydrates(Guo et al., 2019; Rosati et al., 2020; Zieger et al., 2017). Therefore, most aerosols measured by the OPC during the diel cycles will be sized smaller than their geometrical size. Consequently, the vast majority of the particles of all diameters measured by the OPC showing the diel pattern can be considered dry or slightly humidified sea salt.***”

Rosati et al. Reconciling atmospheric water uptake by hydrate forming salts. Environmental Science: Processes & Impacts 22, 1759-1767 (2020).

Reviewer #3 (Remarks to the Author):

The authors had addressed all my questions and concerns very well. The paper is publishable now

We thank the reviewer for the valuable comments made in the first review.

REVIEWERS' COMMENTS

Reviewer #1 (Remarks to the Author):

Once again I would like to thank the authors for taking the time to address my comments along with those of the other reviewers. I congratulate the authors for an interesting and timely piece of work and feel that the manuscript is now in a publishable form.

Matt Salter